# Evaluation and interpretation of convolutional long short-term memory networks for regional hydrological modelling

Sam Anderson[1], Valentina Radić[1]

[1]Department of Earth, Ocean, and Atmospheric Sciences, University of British Columbia, Vancouver, V6T 1Z4, Canada

*Correspondence to*: Sam Anderson (sanderson@eoas.ubc.ca)

**Abstract.** Deep learning has emerged as a useful tool across geoscience disciplines; however, there remain outstanding questions regarding the suitability of unexplored model architectures and how to interpret model learning for regional scale hydrological modelling. Here we use a convolutional long short-term memory network, a deep learning approach for learning
both spatial and temporal patterns, to predict streamflow at 226 stream gauges across southwestern Canada. The model is forced by gridded climate reanalysis data and trained to predict observed daily streamflow between 1980 and 2015. To interpret the model's learning of both spatial and temporal patterns, we introduce a set of experiments with evaluation metrics to track the model's response to perturbations in the input data. The model performs well in simulating daily streamflow over the testing period, with a median Nash-Sutcliffe Efficiency (NSE) of 0.68 and 35% of stations having NSE > 0.8. When
predicting streamflow, the model is most sensitive to perturbations in the input data prescribed near and within the basins being predicted, demonstrating that the model is automatically learning to focus on physically realistic areas. When uniformly perturbing input temperature timeseries to obtain relatively warmer and colder input data, the modelled indicator of freshet timing and peak flow changes in accordance with the transition timing from below- to above-freezing temperatures. We also demonstrate that modelled August streamflow in partially glacierized basins is sensitive to perturbations in August
temperature, and that this sensitivity increases with glacier cover. The results demonstrate the suitability of a convolutional long short-term memory network architecture for spatiotemporal hydrological modelling, making progress towards interpretable deep learning hydrological models.

## 1 Introduction

The use of deep learning (DL) has gained traction in geophysical disciplines as an active field of exploration in efforts to
maximize the use of growing in situ and remote sensing datasets (Bergen et al., 2019; Reichstein et al., 2019; Shen, 2018). In hydrology, DL can provide alternative or complementary approaches to supplement traditional process-based modelling (Shen et al., 2018; Hussain et al., 2020; Van et al., 2020; Marçais and de Dreuzy, 2017; Shen, 2018). Particularly notable are DL models which have been found to outperform traditional hydrological models applied at regional scale, including those for

streamflow prediction at daily temporal scale (Kratzert et al., 2018, 2019b), at hourly temporal scale (Gauch et al., 2021), and at ungauged basins (Kratzert et al., 2019a). These recent DL-based studies have emphasized the development of lumped hydrological models with inputs that are aggregated to the basin-level. However, fewer DL-based studies have explored the use of spatially discretized forcing and geophysical data (Gauch and Lin, 2020). In contrast, traditional process-based approaches have made substantial progress towards distributed hydrological models which are driven by spatially discretized inputs (Freeze and Harlan, 1969; Marsh et al., 2020; Pomeroy et al., 2007). Nevertheless, as input and target data are becoming available at increasingly finer spatiotemporal resolution, process-based modellers are having to address the rising computational requirements and human labour required to represent the relevant hydrological processes across larger spatial scales (e.g. Marsh et al., 2020). A key opportunity exists, then, to develop a DL hydrological model which can utilize spatially discretized forcing data at regional scale.

Early applications of machine learning in hydrology date back to the 1990s, with artificial neural network (ANN) models used for rainfall-runoff modelling (Hsu et al., 1995; Maier and Dandy, 2000, 1996; Maier et al., 2010; Zealand et al., 1999) and a range of other hydrometeorological analysis such as flood forecasting (Fleming et al., 2015), improving gridded snow-water equivalent data products (Snauffer et al., 2018), and predicting seasonal water supply (Hsieh et al., 2003). ANN models aim to approximate functions that connect input data (e.g. weather data), represented by input neurons, to output or target data (e.g. streamflow data), represented by output neurons, through a series of hidden layers, each containing hidden neurons. The training of these models, i.e. the tuning of model parameters in the functions interconnecting each layer, aims to minimize the distance between model output and observed target data. In particular, numerous types of machine learning applications have been developed for hydrometeorological analyses and applications in Western Canada. For example: Bayesian neural networks, support vector regression, and Gaussian processes have been used for streamflow prediction at a single basin (Rasouli et al., 2012); quantile regression neural networks have been used for precipitation downscaling in British Columbia (Cannon, 2011) and estimation of rainfall intensity-duration-frequency curves across Canada (Cannon, 2018); online sequential extreme learning machines have been used for streamflow prediction in two basins (Lima et al., 2016, 2017); and random forest models have been used to identify temperature controls on maximum snow-water equivalence in Western North America (Shrestha et al., 2021). While such machine learning architectures have a long history and continue to find useful applications in hydrology, DL has more recently become a promising area of investigation due to several key characteristics (Shen, 2018): DL models can automatically extract abstract features from large, raw datasets (Bengio et al., 2013); and the existence of DL model architectures which are explicitly designed to learn complex spatial and/or temporal information, in particular convolutional neural networks (LeCun et al., 1990) and long short-term memory neural networks (Hochreiter and Schmidhuber, 1997).

Long short-term memory (LSTM) neural networks are designed to learn sequential relationships on a range of scales (Hochreiter and Schmidhuber, 1997). LSTMs are a type of recurrent neural network (RNN). Traditional RNNs include a feedback loop between the network output and input in order to learn temporal dependency within the data (Rumelhart et al., 1985); however, they struggle to learn long-term dependencies greater than around 10 time steps (Bengio et al., 1994). LSTMs

overcome this limitation through the inclusion of an internal memory state or cell state which can store information, and learning is achieved by including internal gates through which information can flow and interact with the cell state. LSTMs have had particular success in natural language processing (NLP), including applications of text prediction (Karpathy et al., 2015), language translation (Sutskever et al., 2014), image captioning (Kiros et al., 2014), and video-to-text conversion (Venugopalan et al., 2015). In hydrology, Kratzert et al. (2018) demonstrated the effectiveness of LSTMs for rainfall-runoff modelling, using the previous 365 days of basin-averaged weather variables to predict the next day of streamflow at a stream gauge station. They have shown that a single LSTM can be trained on hundreds of basins, and then fine-tuned for either each region or each basin, oftentimes outperforming standard hydrological models. LSTM models trained on many basins have been shown to outperform standard hydrological models for prediction at ungauged basins, and the inclusion of physical basin characteristics as predictors further improved the LSTM model performance, demonstrating the potential for LSTM models to be used as regional hydrological models (Kratzert et al., 2019a). However, while addressing the need to learn complex sequential information, the LSTM approach does not explicitly learn from spatially discretized information, and as such has been primarily used for lumped hydrological modelling.

Another type of DL architecture, convolutional neural networks (CNNs) are designed to learn spatial information. Learning is achieved through convolving an input with a layer of filters made up of trainable parameters. The development of CNNs was largely driven by image classification applications (Krizhevsky et al., 2012). In the geosciences, CNNs have gained popularity more recently with applications including long-term El-Nino forecasting (Ham et al., 2019), precipitation downscaling (Vandal et al., 2017), hail prediction (Gagne II et al., 2019), urban water flow forecasting (Assem et al., 2017), and beach state classification (Ellenson et al., 2020). Importantly, CNNs have been combined with LSTMs to encode both spatial and temporal information. Sequential CNN-LSTM models have been used to map input videos to class labels or text captions, where frames in the video are passed first through a CNN, the output of which is then passed through an LSTM (Donahue et al., 2017). Alternatively, LSTM models with convolutional rather than fully connected (or 'dense') layers have also been used to encode spatiotemporal information for applications including precipitation nowcasting (Shi et al., 2015). In hydrology, CNN (and particularly combined CNN-LSTM) models have seen fewer applications to date as compared to the LSTM approach, with recent work developing 1D CNNs for rainfall-runoff modelling (Hussain et al., 2020; Van et al., 2020). Notably, the CNN-LSTM architecture has been identified as being an architecture of potential or emergent interest for geoscientific applications involving spatiotemporal phenomena (Reichstein et al., 2019).

Historically, hydrological model development has emphasized understanding and incorporating physical processes in order to improve model performance (Freeze and Harlan, 1969; Pomeroy and Gray, 1990; Hedstrom and Pomeroy, 1998; Painter et al., 2016; Marsh et al., 2020). Considering the emphasis on process-based modelling within the hydrological community (Bahremand, 2016) and the multifaceted challenges surrounding water management (Milly et al., 2008; Wheater and Gober, 2013), it is important that DL-based hydrological models are interpretable and trustworthy in addition to being successful in simulating accurate streamflow. Fleming et al. (2021b) discuss the importance of model interpretability in the context of operational hydrological forecasting where model predictions may be used for potentially high-stakes decision

making; for example, the end user may need to communicate why models make a certain prediction in order to answer clients' questions or to satisfy legal requirements. We may begin to build trust in a model's ability to forecast in the near-term by evaluating model performance on a testing dataset that is separate in time from the training and validation datasets. This approach, however, does not offer much insight into the physical relationships that the models are relying on for decision making. Practical methods are beginning to appear that allow users to easily identify and geophysically interpret, in detail, spatiotemporal patterns or input-output relationships identified by, respectively, new unsupervised learning (e.g., Fleming et al. (2021b)) and supervised learning (e.g., Fleming et al. (2021a)) algorithms designed for applied operational hydrological modelling environments where interpretability is key. However, there is still much work to be done on developing new and better ways to further the goal of explainable machine learning for hydrology, in both deep and non-deep contexts and both operations and research settings. Additionally, without an understanding of what models have learned, it is challenging to trust a DL model for predictions in periods or places where observational datasets do not exist (e.g. for reconstructing missing historical streamflow, for predicting streamflow at ungauged basins, or for long-term forecasting of streamflow under climate change scenarios). By interpreting what a DL model has learned, we can better understand where and when a DL model can be trusted and the tasks for which it can be applied.

How a model is interpreted, and what it means to interpret a DL model, may depend on the model architecture (e.g. ANN, CNN, LSTM), the task the model is performing (e.g. regression, classification), and the research or practical questions being asked with the model. A review of methods used for DL interpretation in a geoscientific context is provided in McGovern et al. (2019), and here we summarize select concepts and methods. One approach to interpret CNN models is to visualize the regions in the input that are most important for decision making, which can be done for both classification and regression problems. Techniques such as class activation mapping (CAM) and gradient class activation mapping (Grad-CAM) utilize deep feature maps to determine the regions most important for classification (Selvaraju et al., 2016). Another technique, layerwise relevance propagation (LRP), backpropagates from a single output neuron through the trained network to identify the input region which is most relevant for determining the value of the output neuron (Bach et al., 2015). For LRP, the propagation rules used depend on model architecture (Arras et al., 2019; Bach et al., 2015; Toms et al., 2020). In contrast to the above approaches which interpret the model through explicit use of the model parameters, alternative methods exist which do not use internal network states for interpretation. For example, techniques such as occlusion (Zeiler and Fergus, 2014) and randomized image sampling explanation (RISE) (Petsiuk et al., 2018) iteratively gray- or zero-out subsets of an input image and measure how a model's predictions change due to this perturbation. Occlusion and RISE can identify the area in the input where the model's predictions are most sensitive to perturbation, which can be interpreted as being the most important information for the model to have in order to make its prediction.

Recurrent networks can be challenging to interpret as the relevance of any feature in the network depends on the processing of previous features. LSTMs have often been interpreted by analysing their internal states (Shen, 2018). For example, Karpathy et al (2015) visually inspect cell states of an LSTM trained for natural language processing applications to identify states which track various recognizable text features, such as quotations and line length. Most states, however, were found to

be uninterpretable (Karpathy et al., 2015). A similar approach has been taken for interpreting LSTMs in hydrology; for example, Kratzert et al. (2018) discuss cell states as being comparable to storages in traditional hydrological models. They show that the evolution of one cell state closely resembles the dynamics of a snowpack, increasing when temperatures are below freezing and quickly depleting when temperatures rise above freezing (Kratzert et al., 2018). More recently, LRP has been adapted for the LSTM architecture (Arras et al., 2019); however, to our knowledge there are no examples of its use in the geoscientific literature.

Deep learning in hydrology has shown promise for streamflow prediction tasks, but knowledge gaps exist surrounding the development of architectures which explicitly incorporate both space and time, the interpretation of model learning, and the limitations of such modelling approaches. We aim to address some of these knowledge gaps by creating a relatively simple and interpretable DL model which maps spatiotemporal weather fields, represented by gridded climate data at a relatively coarse (~75 km) spatial resolution, to streamflow at multiple stream gauge stations across a region. By explicitly encoding spatial information, we aim to develop a DL analogue of a distributed hydrological model which predicts streamflow on a regional scale without the need for climate downscaling. Our specific objectives for this paper are to: 1) evaluate how well the sequential convolutional long short-term memory model performs when predicting daily streamflow simultaneously at multiple stream gauge stations across a region, 2) investigate if the model has learned to focus on the areas of the spatially distributed input data that are within or near the watersheds where streamflow is being predicted, 3) investigate if the model has learned physically interpretable temporal links which drive the timing and peak flow of the spring freshet in snowmelt dominated basins, and 4) investigate if the model has learned to link summer temperature with summer streamflow in glacierized basins. The first objective is related to evaluating the accuracy of the model's predictions, while the latter three objectives relate to model interpretability. We do not undergo an exhaustive parameter search to create the best or most complex model; rather, we develop a model with relatively few predictor variables which is sufficient for achieving these objectives. We explore several ways that perturbations to the input temperature and precipitation fields result in streamflow responses that are expected on the basis of physical hydrologic knowledge. While this is not necessarily a unique property of DL and may be found when using non-deep machine learning or other empirical models applied to the same task, our findings are encouraging given the recent use of DL for streamflow prediction tasks.

The paper is structured in the following way: in Sect. 2, we discuss the study region. In Sect. 3, we outline the datasets used, and detail our decision making for choosing the input and output variables. In Sect. 4, we outline our methods, including the architecture, training, and evaluation of the model, and describe the experiments developed for interpreting the model's learning. In Sect. 5, we present and discuss the results of our analysis, and we present a summary and conclusions in Sect. 6.

## 2 Study region

We use southwestern Canada as our study region, containing large sections of the provinces of British Columbia (BC) and Alberta (AB) (Fig. 1). This region contains a range of hydroclimate regimes, allowing for our modelling approach to be

evaluated across a range of conditions. In winter, relatively warm and moist Pacific air is advected into the study region, leading to frequent rainfall events at low elevations along the west coast of British Columbia where the maritime climate is

wetter and milder as compared to much of the rest of the study region. While most precipitation typically falls as rain at lower elevations, substantial winter snowfall leads to seasonally continuous snow and substantial glacier coverage at higher elevations in the coastal region (Trubilowicz et al., 2016). Cooler winter conditions through much of the rest of the province allow for the accumulation of a seasonal snowpack (Moore et al., 2010). In contrast, winters in Alberta are colder and drier given the influence of Arctic air masses. Substantial snowfall can occur in Alberta when comparably moist Pacific air crosses

the Rockies and interacts with cold and dry Arctic air (Vickers et al., 2001; Sinclair and Marshall, 2009), but most precipitation in Alberta falls as rain in the spring and summer. The seasonal streamflow characteristics are described in Sect. 3.1.

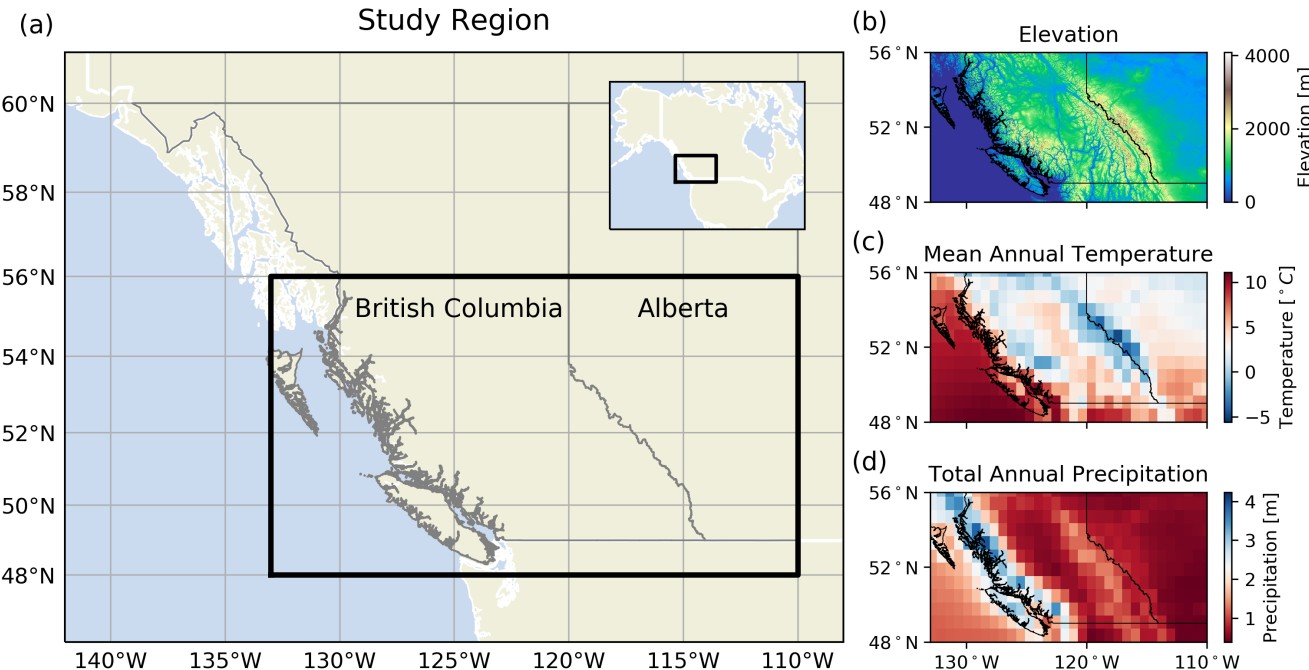

**Figure 1: The study region in Western Canada.** a) The black box outlines the study region in both the main figure and the inset. The provincial borders of British Columbia and Alberta are shown in grey. The inset shows the broader context of the study region in North America. Made with the Python library Cartopy (Met Office, 2018) with data from Natural Earth. b)

Elevation data of the study region from the Shuttle Radar Topography Mission (Farr et al., 2007). c) Mean annual temperature

and d) mean annual precipitation as calculated over the study period (1979 – 2015) from ERA5 climate reanalysis (Hersbach et al., 2020).

## 3 Data

### 3.1 Streamflow data

We use daily streamflow data (in $[m^3\ s^{-1}]$) extracted from Environment and Climate Change Canada's Historical Hydrometric Data website (HYDAT) (Water Survey of Canada HYDAT data). HYDAT classifies stream gauge stations as either "regulated" (downstream of regulating structures such as a dam) or "natural" (upstream of regulating features). We use stations which are classified as natural and which are currently active. Many stream gauges do not record data every day of the year, so we only select stream gauges which have no more than 40% of daily data missing between 1980 and 2015 and for which no more than 1 year is missing more than 40% of data.

A temporally complete dataset is needed to train the model. For all missing data, we fill the daily value with the average value of that day between 1980 – 2015. If all years are missing that day (which is true for some stations which record data seasonally rather than continuously), we fill the missing day with the minimum seasonal streamflow. The threshold of 40% is chosen to allow for relatively dense spatial coverage of stations across the study region and is acceptable for the purposes of this study considering that most missing data are during low-flow seasons when rainfall and snowmelt are not strongly driving streamflow dynamics. It is acceptable to allow for one year with greater than 40% missing data because it substantially increases the station density. There are 279 stream gauge stations in Alberta which are active and measure natural flow. Of these, 120 meet the aforementioned criteria; however, only 66 meet the stricter criteria of having all years with less than 40% missing data. In BC, there are 288 active and natural stream gauges; of these, 145 meet the less strict criteria and 108 meet the stricter criteria. We further restrict the study region to stations south of 56° N because stream gauge density is greater below this latitude. Missing data are common features in geoscientific datasets, the presence of which pose challenges for the use of machine learning models (Karpatne et al., 2019), and so creating a suitably large training dataset may require pre-processing steps like those outlined above.

The Reference Hydrometric Basin Network (RHBN) is a subset of the national stream gauge network which have long records and minimal human impacts that have been identified for use in climate change studies. Of the 226 stations used in our study, 213 are within the RHBN. The remaining 13 stations have long observational records and are not modified by regulating structures but may have more than minimal human impacts through other disturbances to the natural system such as land use. We provide station names, station numbers, and if they are a part of the RHBN network in Table S1.

### 3.2  Streamflow clusters

Streamflow throughout the study region varies strongly in space and time and reflects the varied topographic and climatic conditions in British Columbia and Alberta. Here we provide a brief, high-level overview of streamflow characteristics, and while it is not a complete summary of the full range of hydrologic conditions throughout the study region, we aim to highlight that streamflow through the region is heterogeneous in space and time. Streamflow at low-elevation coastal stations is primarily driven by rainfall, with highest monthly discharge in November or December. In contrast, streamflow at stations

that are at higher elevation, further north, or further inland transition to a snowmelt-dominated regime, with highest monthly

discharge in spring or early summer. Numerous glaciers exist in high elevation alpine areas throughout both the Coast Mountains along the west coast of British Columbia and the Rocky Mountains along the border between British Columbia and Alberta, and glacier runoff contributes to streamflow through late summer once the seasonal snowpack has melted (Eaton and Moore, 2010). East of the Rocky Mountains, the Prairie region in eastern Alberta is characterized by relatively flat topography with small surface depressions (LaBaugh et al., 1998). Water can pond and be stored in these depressions, leading to

intermittent connectivity throughout many basins and drainage areas which may vary in time (Shook and Pomeroy, 2011).

Previous studies have used a range of techniques to cluster or summarize the diversity of spatiotemporal streamflow characteristics in the study region (e.g. Halverson and Fleming (2015) use complex networks to represent similarity between streamflow timeseries in the Coastal Mountains, while Anderson and Radić (2020) use principal component analysis and Self-Organizing Maps to characterize summer streamflow through Alberta). In this study we use a relatively simple clustering

approach, only considering seasonal streamflow, station latitude, and station longitude. The clustering input (observation) for each station is a vector where the first 365 dimensions are the daily values of the climatological seasonal streamflow, the next 182 dimensions are repeated values of latitude, and the final 182 dimensions of repeated values of latitude (Fig. A1 in Appendix A). This clustering input is designed to give the daily values of the climatological seasonal streamflow and geographic location similar weight in the clustering algorithm, where approximately one half of the input is seasonal streamflow, one quarter is

latitude, and one quarter is longitude. By clustering in this way, the stream gauges that belong to the same cluster are likely to have similar streamflow and experience similar climatic conditions. Seasonal streamflow is normalized at each station to have zero mean and unit variance, while latitude and longitude are each normalized across all stations to have a mean of zero and unit variance. We use agglomerative hierarchical clustering with Ward's method (Hastie et al., 2009) to identify six subdomains or clusters of stream gauge stations (Fig. 2). The number of clusters chosen (six in this case) is determined from

the dendrogram (Fig. A2). We refer to the clusters as north-western, north-eastern, central, southern, eastern, and coastal, as labelled in Fig. 2. The elevation and drainage area of stations for each cluster is shown in Fig. A3.

There are key differences between the streamflow regimes identified by the clustering (Fig. 2). Only the lower-elevation coastal stream gauges are characterized by low summer flows and high winter flows which are driven by winter rainfall events; all other clusters differ from one another largely in the timing and peak flow of both the spring freshet (the first streamflow

peak in a year) and a second rainfall-driven peak occurring in spring, summer, or fall. The eastern and north-eastern clusters are characterized by relatively early spring freshet, followed by rainfall-driven streamflow peaks in early summer. The southern, central, and north-western stations are characterized by a later and more sustained spring runoff, in part due to a longer-lasting snowpack which accumulates from the relatively higher rates of winter precipitation in British Columbia.

Our clustering approach does not explicitly consider input features such as land use, glacier coverage, drainage area,

or elevation, but rather implicitly considers the expressions of these features in the seasonal hydrograph. The goal of this type of clustering is to define subsets of stream gauge stations that are nearby in space and share similar hydrographs. We prioritize proximity in space over an explicit representation of other important features (e.g. drainage area, elevation, glacier coverage)

because a key goal of the study is to interpret where in space the DL models have learned to focus when predicting streamflow. As discussed in Sect. 4.3.1 and Sect. 4.5.1, having clusters of stream gauge stations which are nearby in space allows us to visualize if the trained models are learning to focus on the subregion of the input domain which overlaps with the watersheds where streamflow is being predicted.

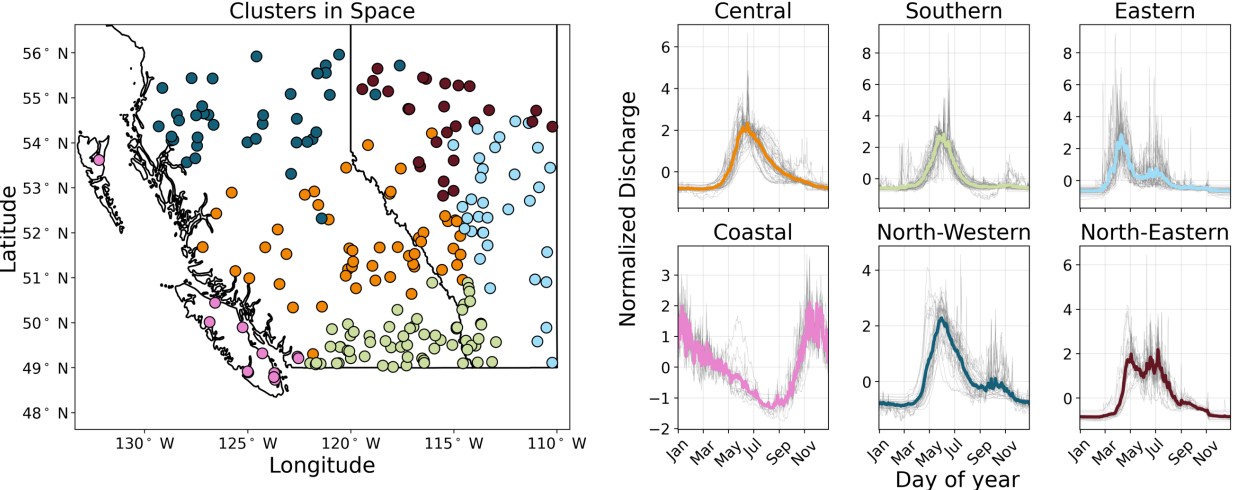

**Figure 2: The seasonal streamflow cluster patterns and their locations in space.** The colour of the stream gauge in the left panel corresponds to the climatological seasonal streamflow cluster pattern on the right. The background grey curves in the cluster pattern panels are the daily values of climatological seasonal streamflow at the stream gauges in each cluster. Seasonal discharge of each station is normalized to have a mean of zero and unit variance.

### 3.3 Weather data

As input weather variables to the model, we select daily fields of precipitation, maximum temperature, and minimum temperature, extracted from ERA5 reanalysis (Hersbach et al., 2020) from the European Centre for Medium-Range Weather Forecasts (ECMWF). Data are aggregated to daily temporal resolution and 0.75° x 0.75° spatial resolution for the time period 1979 – 2015. Our selection of variables is based on the assumption that the combination of precipitation and temperature is sufficient for estimating both how precipitation contributes to streamflow as rainfall, and for estimating the onset, peak flow, and longevity of the spring freshet through the seasonal accumulation and ablation of a snowpack. We recognize that the underlying physics which governs streamflow throughout the year is more complex than these comparably simple assumptions (e.g. interactions between surface water and ground water (Hayashi et al., 2016), evapotranspiration (Penman and Keen, 1948), snow redistribution from wind (Pomeroy and Li, 2000)); however, we are assuming that temperature and precipitation from reanalysis data can act as proxies from which most of the information can be inferred (e.g. Essou et al., 2016). While additional

variables could be used as climatic drivers of streamflow (e.g. solar radiation, evaporation, wind), we opt to use a simpler model with fewer input variables as a proof of concept and to achieve our goals stated in Sect. 1.

ERA5 reanalysis is globally available from 1979 through the present (once complete it will be available from 1950 onwards), and has been shown to compare well against other reanalysis products (Hersbach et al., 2020). ERA5 reanalysis was preceded by the ERA-Interim reanalysis, which has been evaluated for use across British Columbia. It was found that daily minimum and daily maximum temperatures are well represented across British Columbia (Odon et al., 2018). Additionally, daily precipitation was found to be well represented, with the caveat that extreme precipitation is less successfully represented (Odon et al., 2019). ERA5 reanalysis better represents precipitation as compared to ERA-Interim reanalysis at the global scale (Hersbach et al., 2020). Importantly, the precipitation output from ERA5 has been found to typically outperform the earlier ERA-Interim reanalysis in the northern Great Plains region, which experiences a similar climate to the Prairie region in our study area (Xu et al., 2019). For these reasons we consider the ERA5 reanalysis to be suitable for our study. ERA5 data are available as a preliminary product 5 days behind real time, and as a final product 2 – 3 months behind real time (Hersbach et al., 2020). This latency has implications for model applications, as it may not be possible to use ERA5 data for real-time forecasting with the model in this study. We downloaded total precipitation (variable name: 'tp'; parameter ID: 228) and near-surface air temperature (variable name: '2m Temperature'; parameter ID: 500011), from which daily total precipitation, daily maximum temperature, and daily minimum temperature are calculated.

## 4 Methods

Here we summarize our methods before providing details of each key step. We use a sequential CNN-LSTM model to map weather predictors to streamflow at multiple stream gauge stations simultaneously. As input data, we use the past 365 days of weather, covering the whole study region, in order to predict streamflow of the next day at $N$ stream gauge stations (Fig. 3 and Table 1). The CNN learns the spatial features in each day of the input, while the LSTM model learns the temporal relationship between these features in order to predict streamflow. After the model is trained, we evaluate its performance against the observed streamflow over a testing period, which is independent of the training and validation periods. Finally, we introduce three experiments to investigate the model's learning. The first experiment is focused on interpreting the learning of spatial links between the predictors and streamflow, the second experiment is focused on the learning of links between temperature and the snowmelt-driven freshet, and the third experiment is focused on the learning of links between August temperature and August streamflow in glacierized basins. This section will provide a brief overview of both the CNN and LSTM architectures, followed by a description of our CNN-LSTM model design and training, and finally with a description of metrics and experiments developed for the model evaluation and interpretation.

**Table 1: Details of the model layers.** The model input has shape $365 \times 12 \times 32 \times 3$, with the final dimension corresponding to the 3 input variables (daily maximum temperature, minimum temperature, and precipitation). Note that each of the 365 daily weather images is passed independently through the convolutional and pooling layers (e.g. these layers are time-distributed), and so the output shape of these layers has a first dimension of 365.

| Layer Type | Description | Output Shape | Number of Parameters |
|---|---|---|---|
| Convolutional | 32 filters, 1x1 size | 365 x 12 x 32 x 32 | 128 |
| Convolutional | 16 filters, 3x3 size | 365 x 12 x 32 x 16 | 4624 |
| Convolutional | 16 filters, 3x3 size | 365 x 12 x 32 x 16 | 2320 |
| Max Pooling | 2x2 pool size | 365 x 6 x 16 x 16 | 0 |
| Convolutional | 32 filters, 3x3 size | 365 x 6 x 16 x 32 | 4640 |
| Convolutional | 32 filters, 3x3 size | 365 x 6 x 16 x 32 | 9248 |
| Max Pooling | Global | 365 x 32 | 0 |
| Dropout | 0.1 dropout rate | 365 x 32 | 0 |
| LSTM | 80 units | 80 x 1 | 36160 |
| Dense | As many neurons as stream gauges | N x 1 | N*81 |

## 4.1 CNN overview

The CNN is constructed using two main types of layers: convolutional and pooling. Convolutional layers are made up of multiple filters, which are constructed by trainable weights. Each filter convolves across an input layer to produce an output image which is then passed through a nonlinear activation function. Mathematically, a single output neuron can be calculated from a single filter as:

$$y_{CNN} = g\left( \sum_{i,j,k} W_{CNN}^{i,j,k} x_{CNN}^{i,j,k} + b_{CNN} \right) \tag{1}$$

where $y_{CNN}$ is the value of one neuron in the output layer, $g$ is the nonlinear activation function, $W_{CNN}$ are the weights of the filter, $x_{CNN}$ is the region of the input layer, $b_{CNN}$ is the bias value of the output neuron, and $i$, $j$, and $k$ correspond to width (e.g. number of pixels along the $x$-direction of the image), height (e.g. number of pixels along the $y$-direction of the image), and depth (e.g. number of channels of the image) of the input, respectively. Pooling layers reduce image resolution, which reduces memory requirements of the network; for example, a 2x2 max-pooling layer will reduce the number of pixels by a factor of 4 by outputting only the maximum value of each 2x2 region of the input. CNN architectures often have a repeating structure of several convolutional layers followed by pooling layer. Through training, the convolutional layers learn the spatial

features present with more abstract features being learned at deeper layers, and the pooling layers reduce images to smaller and smaller sizes. The output feature vector is encoded with the learned spatial information from the input.

## 4.2 LSTM overview

The LSTM network output is determined by the interaction between two internal states: the cell state $c(t)$ which acts as the memory of the network, and the hidden state $h(t)$ which is an intermediate output of the network. Both states are updated at each time step $t$ ($1 \leq t \leq n$) by a series of gates through which information can flow: the forget gate $f_t$, the input gate $i_t$, the potential cell update $\widetilde{c}_t$, and the output gate $o_t$. Each time step of the input is concatenated with the hidden state as calculated in the prior time step before being passed through the network; in this way, learned information from previous time steps is used to calculate the next output. In the following equations, weights ($\mathbf{W}$) and biases ($\boldsymbol{b}$) are the learnable parameters in the network:

$$f_t = \sigma\big(\mathbf{W}_f[\boldsymbol{x}_t, \boldsymbol{h}_{t-1}] + \boldsymbol{b}_f\big) \tag{2}$$
$$i_t = \sigma(\mathbf{W}_i[\boldsymbol{x}_t, \boldsymbol{h}_{t-1}] + \boldsymbol{b}_i) \tag{3}$$
$$\widetilde{c}_t = \tanh(\mathbf{W}_c[\boldsymbol{x}_t, \boldsymbol{h}_{t-1}] + \boldsymbol{b}_c) \tag{4}$$
$$o_t = \sigma(\mathbf{W}_o[\boldsymbol{x}_t, \boldsymbol{h}_{t-1}] + \boldsymbol{b}_o) \tag{5}$$

where $x_t$ is the input vector to the LSTM at time $t$, tanh is a hyperbolic tangent function, $\sigma$ is a sigmoid function, and square brackets indicate concatenation. The cell state at time $t$ is determined by the prior cell state and the interactions with the outputs of the forget, input, and potential cell update, while the hidden state at time $t$ is determined by the new cell state and the output gate:

$$c_t = c_{t-1} \odot f_t + i_t \odot \widetilde{c}_t \tag{6}$$
$$h_t = \tanh(c_t) \odot o_t \tag{7}$$

where $\odot$ denotes elementwise multiplication. The final hidden state, $\boldsymbol{h}_n$, is passed through a dense layer constructed of fully connected neurons. The activation of this dense layer is linear, and so the final output is a linear transformation of the final hidden state:

$$y_{flow} = \mathbf{W}_d h_n + b_d \tag{8}$$

This output, $\boldsymbol{y}_{flow}$, is a vector of normalized streamflow for a single day at multiple stream gauge stations.

## 4.3 Sequential CNN-LSTM architecture

An overview of the model architecture is shown in Fig. 3, and information of each layer is presented in Table 1. To ensure consistency between terminology in both image processing (from which CNN technologies primarily originated) and this study, a 'weather video' refers to 365 days of the three weather predictors, a 'frame' or 'image' in a weather video refers to one day of the three weather predictors at all grid cells, a 'channel' in a 'frame' or 'image' refers to one day of one weather predictor, and a 'pixel' refers to one grid cell. We use a sequential CNN-LSTM model in order to simultaneously map the previous 365 days of temperature and precipitation to the next day of streamflow at multiple stream gauges throughout the

study region (i.e. days 1 through 365 of weather are used to predict day 366 of streamflow). Daily weather images are constructed where the height and width of the image correspond to the number of grid cells along latitude and longitude, respectively, and with three channels corresponding to normalized maximum temperature ($T_{max}$), minimum temperature

($T_{min}$), and precipitation ($P$). Yearly weather videos are constructed from the past 365 days of weather images, where each frame in the video is a weather image. One year-long weather video is used as an input to predict the next day of streamflow at the 226 stream gauge stations; in other words, all grid cells of temperature and precipitation are mapped to streamflow at all stream gauge stations. Each frame in the video is passed independently through the CNN, which converts each of the 365 frames into a feature vector of length 32. This feature vector is a representation of the learned spatial features found in that

frame of weather. There are 365 feature vectors generated from one year-long weather video, since there are 365 days in the input video. Then, this series of feature vectors is passed through an LSTM, which learns the sequential relationship between the learned spatial features and outputs a final hidden vector, $h_n$ (Equation 7), with length 80. This hidden vector contains information of the sequential relationships between the spatial features and is next passed through a dense layer with linear activation to connect to the final output neurons ($y_{flow}$, Equation 8). In other words, the 80 values in the hidden vector are

linearly combined to predict a single day of streamflow at each individual station.

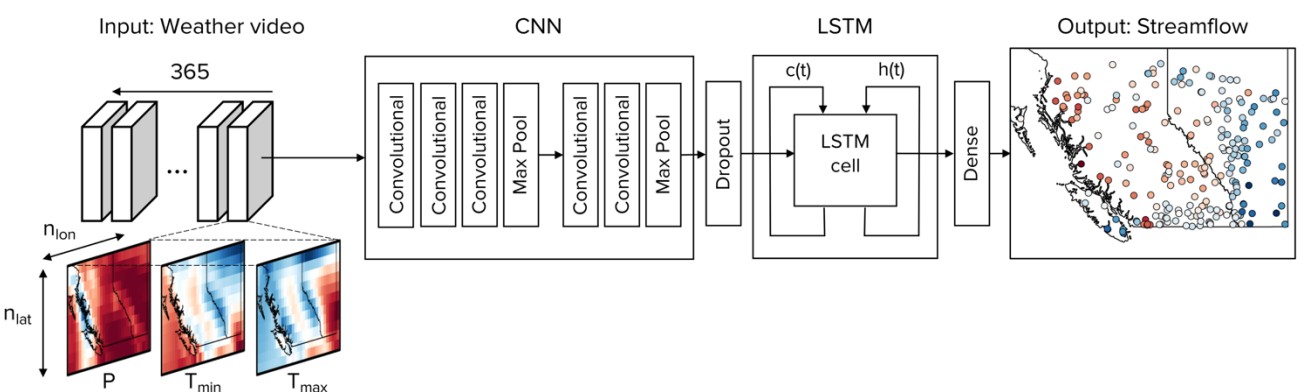

**Figure 3: Overview of the model architecture.** The model input is a weather video with 365 frames/images, each corresponding to one day of weather from ERA5 reanalysis in the past year. Each frame in the video has three channels

corresponding to precipitation (P), maximum temperature ($T_{max}$), and minimum temperature ($T_{min}$). Each channel has dimensions of $n_{lat} \times n_{lon}$, where $n_{lat}$ is the number of grid cells in the vertical direction (latitude) and $n_{lon}$ is the number of grid cells along the horizontal direction (longitude). Each frame in the weather video is passed through a CNN, and each weather video generates a sequence of 365 feature vectors. A dropout layer is used between the CNN and LSTM for regularization. The sequence of 365 feature vectors is then passed through an LSTM and a dense linear transformation to

output the next day of modelled streamflow at N stations (i.e. streamflow at day 366 at N stations). Within the LSTM cell, $c(t)$ is the cell state and $h(t)$ is the hidden state.

We divide our data into three subsets referred to as training, validation, and testing datasets, as is common practice in DL model development (e.g. Goodfellow et al. (2016)). The training data are used to iteratively update the model parameters such
that the error between the model's predictions and known observations is reduced across the training set; the validation data are used to determine when to stop updating the model parameters to prevent the model from overfitting to the training data; and the testing data are used to evaluate the final model's performance.

Since 365 days of previous temperature and precipitation are used to predict streamflow, and since the ERA5 data begin on January 1, 1979, the first day of streamflow predicted is January 1, 1980. For all models, we use 1980 – 2000 for training,
2001 – 2010 for validation, and 2011 – 2015 for testing. In other words, the training period is defined by daily streamflow from January 1, 1980 to December 31, 2000, with forcing data ranging from January 1, 1979 to December 30, 2000. The validation period uses streamflow data from January 1, 2001 to December 31, 2010, with forcing data ranging from January 1, 2000 to December 30, 2010. The testing period uses streamflow data from January 1, 2010 to December 31, 2015, with forcing data ranging from January 1, 2009 to December 30, 2015. We choose to separate the training, validation, and testing
datasets into non-overlapping time periods of streamflow so that model performance can be evaluated on out-of-sample streamflow examples. We choose to use a full decade for validation because we want to encourage the model to perform well across a range of conditions and not for one particular year or climate state, since oscillations in the climate system such as the El-Nino Southern Oscillation and the Pacific-North American atmospheric teleconnection influence streamflow through modifications to temperature, precipitation, and snow accumulation through the study region (Hsieh and Tang, 2001; Whitfield
et al., 2010; Fleming and Whitfield, 2010; Hsieh et al., 2003). We also choose to use multiple years for testing so as to not bias our conclusions towards the conditions of a single year. Furthermore, we partition the training, validation, and testing data by year rather than by percentage of observations (i.e. the testing subset is chosen as 5 years, not 10% of observations) so that we do not bias our results by including or excluding parts of the year when the model performs better or worse than average. Overall, the training-validation-testing data split is approximately 59% - 27% - 14% of the total streamflow dataset.
The input data are normalized so that each variable (maximum temperature, minimum temperature, precipitation) has a mean of zero and unit variance over the training period. The target data from each of the 226 stations are normalized so that each station's streamflow has a mean of zero and unit variance over the training period.

Training the DL model requires a balance of having sufficient complexity to learn the mapping from weather to streamflow, but without being so complex that the model overfits to the training set and performs poorly on the validation or
testing datasets. With that in mind, we designed this architecture (Table 1) considering the following: 1) the number of pooling layers is limited by the relatively small input images (spatial size of 12 x 32); 2) the number of filters in the deepest layer of the CNN determines the length of the spatial feature vector (input to LSTM); and 3) the number of parameters in a single LSTM layer goes linearly with the length of the spatial feature vector and quadratically with the number of LSTM units. In addition to these general guiding principles, we found that a single LSTM layer with more units performed better than multiple
LSTM layers with fewer units, as had previously been used when predicting streamflow at a single station (Kratzert et al.,

2018).  The success of a single LSTM layer with more units is likely because we map the LSTM hidden state to multiple stream gauges (a higher-dimensional space) rather than a single neuron, and so more units are required for this mapping to work well.  Additionally, we found that including 32 filters of size 1x1 as the first layer improved model performance.

### 4.3.1 Training

We use fine-tuning (Razavian et al., 2014; Yosinski et al., 2014) to train our model in two steps:

1.  Bulk training: a CNN-LSTM model is initialized with random weights and is then trained on all 226 stream gauge stations in the region.

2.  Fine-tuning: the bulk model is further trained at stations from each of the six clusters (Fig. 1) in the following way.  The bulk model is copied six times, with one copy used for each cluster, but the last dense layer in the bulk
model is removed and replaced with a new dense layer which has as many neurons as stations in that cluster. Weights in the new dense layer are randomly initialized.  Each fine-tuned model is then trained further on only the stations in that cluster.

For both bulk and fine-tuned models, early stopping is used to reduce overfitting.  We use a dropout layer with a dropout rate of 0.1 between the CNN and LSTM layers for regularization (Srivastava et al., 2014).  We use batch sizes of 64, a learning
rate of $10^{-4}$, mean squared error loss, and Adam optimization (Kingma and Ba, 2017).  We use the Keras (Chollet, 2015) and Tensorflow (Abadi et al., 2016) libraries in Python (Van Rossum and Drake, 2009), and Google Colab for access to a cloud GPU.  We initialize 10 bulk models with 10 different sets of random weights.  Each bulk model is trained and then fine-tuned on each cluster of stream gauge stations, creating 10 fine-tuned CNN-LSTM models per each of the six clusters of stream gauge stations.  We use this ensemble of 10 bulk models and 10 fine-tuned models (per cluster) for our analysis.  Training a
single bulk model on a single cloud GPU in Colab takes on the order of tens of minutes.  It is possible that a better performing architecture or training scheme could be constructed by optimizing hyperparameters with an out-of-sample subset; however, we show our model setup and design is sufficient for achieving the goals of this study.

### 4.4 Evaluation of model performance

We evaluate how well streamflow is simulated by the bulk and fine-tuned models with the Nash-Sutcliffe Efficiency
(NSE) (Nash and Sutcliffe, 1970).  For each station, we calculate NSE over the test period for both the bulk and fine-tuned models, using the ensemble mean as the final model output.  NSE is defined as:

$$NSE = 1 - \frac{\sum_{t=1}^{t=T}(Q_m^t - Q_o^t)^2}{\sum_{t=1}^{t=T}(Q_o^t - \overline{Q_o})^2} \qquad (9)$$

where $T$ is the total number of time steps in the test series, $Q_m^t$ is the modelled streamflow for that station at that time, $Q_o^t$ is the observed streamflow for that station at that time, and $\overline{Q_o}$ is the mean observed streamflow for that station over the whole
test period.  The overall performance of both the bulk and fine-tuned models is evaluated by the median NSE of all stations as evaluated over the test period.  When $NSE = 1$, the modelled streamflow is exactly equal to the observed streamflow, while

$NSE < 0$ indicates very poor model performance as more variability would be captured if the streamflow was represented with its mean value than with the modelled streamflow.

We compute NSE using the mean predictions across the ensemble members, and we quantify an uncertainty in the streamflow prediction as being twice the standard deviation across ensemble members. This uncertainty is due to randomness from the initialized parameters and through training. It is a measure of how different streamflow predictions may be even when using the same architecture and data, and it is not a measure of uncertainty in meteorological forcing. When and where this uncertainty is small (large) indicates that the models in the ensemble predict similar (different) streamflow values for that day. We evaluate performance from an ensemble mean rather than a single model's prediction, and so this uncertainty gives an indication of the magnitude of scatter around the ensemble mean.

## 4.5 Interpretation of model learning

### 4.5.1 Spatial perturbations

We interpret the model's learning of spatial links by testing the following hypothesis: if the model is learning physical processes that drive streamflow at a given stream gauge station, then the modelled streamflow at that station should be most sensitive to perturbations in the watershed or vicinity of that station, and less sensitive to perturbations further from that station. To test this hypothesis, we perturb small spatial regions of the input weather video and determine how sensitive the predicted streamflow of each cluster of stream gauge stations (Fig. 1) is to the areas which are perturbed. To evaluate the regions of the input space which are most important for streamflow predictions at each stream gauge, we take the following steps, each of which will be discussed in more detail:

1. Perturb the input video
2. Evaluate how much the modelled streamflow prediction changes at each station
3. Define a sensitivity map for this perturbation for each station
4. Iterate through steps 1 – 3 for each day in the test series until the sensitivity map no longer substantially changes from further perturbations
5. Evaluate if the sensitive areas are representative of physically realistic learning for each streamflow cluster

Steps 1 – 4 are similar to the occlusion (Zeiler and Fergus, 2014) and RISE (Petsiuk et al., 2018) algorithms in that we iteratively perturb the input and generate sensitivity maps based on how the output changes. The RISE approach zeroes out portions of the input image, which here would be equivalent to setting a portion of the input to be the mean weather values since the input variables are normalized to have zero mean; therefore, the difference between the perturbed and unperturbed input would depend on how close the input variables are to their mean values in each day. Instead, here we perturb the input by adding or subtracting a 2D Gaussian curve from the input video, which alters each day in the input no matter if it is near the mean or not. We developed this method, as opposed to using already established methods such as occlusion, RISE, or LRP, because our method is both agnostic of model architecture, and is grounded in a physical understanding of the key

processes taking place (i.e. the perturbations are adding in synthetic warm/wet or cold/dry areas and we determine if and how
this changes streamflow in the perturbed basins).

In step 1, we define a Gaussian perturbation ($p$) and perturbed daily temperature and precipitation fields ($T_{max,p}$, $T_{min,p}$, and $P_p$) as:

$$
\begin{aligned}
p(x,y) &= \beta * e^{-\frac{1}{2}\left[\frac{(x-x_p)^2}{\sigma_x^2} + \frac{(y-y_p)^2}{\sigma_x^2}\right]} \\
T_{max,p}(x,y,t) &= T_{max}(x,y,t) + p(x,y) \\
T_{min,p}(x,y,t) &= T_{min}(x,y,t) + p(x,y) \\
P_p(x,y,t) &= P(x,y,t) + p(x,y)
\end{aligned}
\tag{10}
$$

where $x$ and $y$ are longitude and latitude (in degrees), $t$ is time (in days), $x_p$ and $y_p$ are the longitude and latitude of a randomly selected point within the study domain (in degrees), $\sigma_x$ and $\sigma_y$ are the standard deviations of the Gaussian distribution in the x- and y-directions (in degrees), $\beta$ is a multiplicative factor which has equal probability of being either 1 or -1 for each perturbation, and $T_{max}$, $T_{min}$, and $P$ are the unperturbed normalized daily maximum temperature, minimum temperature, and precipitation, respectively. The Gaussian distribution has an amplitude of 1 and standard deviations are 1.5 pixels in both the x- and y-direction. The amplitude determines the strength of the perturbation and the standard deviations determine the extent. The amplitude of the perturbation was chosen to be 1 since the climate variables are normalized to have unit variance across the training period. This way each climate variable is perturbed by a maximum of a single standard deviation. $\sigma_x$ and $\sigma_y$ were chosen so that the Gaussian perturbation is small relative to both the height and width of the input weather frame, but larger than a majority of basins. This perturbation is added to every channel (predictor variable) and frame in the input video. An example of a perturbation, a perturbed maximum temperature field, and the perturbed streamflow response is shown in Fig. A4.

In step 2, we pass the perturbed video through the trained model, and calculate the absolute value of the difference between the unperturbed and the perturbed modelled streamflow for each stream gauge. From this difference at each station, we can quantify how important the perturbed area is for the model's decision making. We quantify the importance in step 3 by defining a sensitivity map for each stream gauge station:

$$
s^i(x,y) = \left|Q_m^i - Q_{m,p}^i\right| * p(x,y)
\tag{11}
$$

where $s^i(x,y)$ is the sensitivity map of stream gauge $i$, $Q_m^i$ is the unperturbed modelled streamflow at the stream gauge $i$, $Q_{m,p}^i$ is the perturbed modelled streamflow at the stream gauge $i$, and $p(x,y)$ is the perturbation. Each perturbation produces 226 different sensitivity maps, corresponding to one sensitivity map for each stream gauge station.

In step 4, we iterate through the first three steps for each day in the test series until the mean sensitivity maps converge, here taken as when the relative error between sensitivity maps of subsequent perturbations falls below 0.5%. Then, for each streamflow cluster, we calculate the mean sensitivity map across all stream gauges in the cluster, all iterations, and all days in the test series as:

$$S(x, y) = \frac{1}{N} * \frac{1}{m} * \frac{1}{q} \sum_{k=1}^{q} \sum_{j=1}^{m} \sum_{i=1}^{N} s^{i,j,k}(x, y) \tag{12}$$

where $S(x, y)$ is the mean sensitivity map, $q$ is the number of stream gauges in this cluster, $m$ is the number of days in the test set, $N$ is the number of iterations, and $s^{i,j,k}(x, y)$ is the sensitivity map corresponding to iteration $i$ of day $j$ at stream gauge $k$.

Finally, in step 5, we identify the values in the cluster sensitivity map which are either: 1) within the watershed or within 1 pixel of distance from the watershed boundaries, or 2) further than 1 pixel in distance from the watershed boundaries (where 1 pixel has size 0.75° x 0.75°). If the model is focusing on the areas which are within or near the cluster's basins, then we expect the sensitivity within or near the basins to have a higher mean sensitivity and a substantially different distribution of sensitivity than the distribution of sensitivity outside or far from the basins. To evaluate how different the distributions of sensitivity are, we calculate the Kolmogorov-Smirnov D-statistic (Chakravarti et al., 1967) to compare the distribution of sensitivity map pixels which are within or near the cluster's watershed boundaries, with the distribution of pixels which are not within or near the cluster's watershed boundaries (i.e. all other pixels in the domain). The D-statistic, $D$, is a measure of how different two distributions are, where a value of 0 indicates perfectly overlapping distributions, while a value of 1 indicates entirely non-overlapping distributions. $D$ is calculated as:

$$D = \max|F_{in}(S) - F_{out}(S)| \tag{13}$$

where $F_{in}(S)$ is the cumulative density function (CDF) of sensitivity within/near the cluster's watersheds, and $F_{out}(S)$ is the CDF of sensitivity outside/far from the cluster's watersheds. Watershed boundaries are accessed through the Water Survey of Canada (Environment and Climate Change Canada, 2016). $D$ is calculated for the mean sensitivity map of each cluster for each ensemble member.

Additionally, we characterize the sensitivity maps by the value $A$, here defined as the area fraction of the sensitivity map which is more than the half-maximum sensitivity. Smaller values of $A$ (closer to 0) indicate that the model is focused on a smaller portion of the input area, while large values of $A$ (closer to 1) indicate that larger portions of the input video are important for the model's prediction at that station. $A$ is calculated for the ensemble mean sensitivity map of each station (e.g. there are 226 values of $A$ for the bulk models and 226 values of $A$ for the fine-tuned models).

### 4.5.2 Temperature perturbations

We assume that the transition from below- to above-freezing temperatures is strongly related to the onset of snowmelt and thus the timing of the freshet. While the assumption is a simplification of processes dictated by the surface energy balance, the use of positive temperatures as successful indicators for the warming and melting of snow is a common assumption of positive-degree-day models in simulating snow and glacier melt and was first used by (Finsterwalder and Schunk, 1887). Such positive-degree-day models have since been widely applied for modelling snow and glacier melt across multiple spatial scales (e.g. (Hoinkesand and Steinacker, 1975; Braithwaite, 1995; Hock, 2003; Radic et al., 2014), and have been used in watershed

hydrology models such as the UBC watershed model (Quick and Pipes, 1977) and the HBV-model (Bergström, 1976). Therefore, for interpreting the model's learning, we introduce the following hypothesis: if the model is learning physical processes which are driving streamflow over the course of one year, and since snowmelt is a key contributor to streamflow, then the modelled freshet should occur once temperatures in the forcing data have transitioned from below- to above-freezing. To test the hypothesis, we add a spatially and temporally uniform temperature perturbation, $\Delta T$, to both the maximum and minimum temperature channels, i.e. the same temperature change as measured in degrees Celsius is added to every pixel and every day in the test period. With this perturbation we create a new test set which has either warmer or colder temperature channels than the original, but the same precipitation channel. We pass this new test set through the model and compute the mean seasonal flow for each cluster, where the mean is derived across all years in the test set and all stations in the cluster. We perform these steps for the range $-5°C \leq \Delta T \leq 5°C$ with an increment of 1°C to test how the modelled streamflow responds under a range of warmer or cooler conditions.

Then, for each cluster region and for each temperature perturbation, we identify when the 30-day running mean of daily minimum temperature and maximum temperature transition from being below- to above-freezing temperatures:

$$T_{max}(t_{0,max}) = 0°C \tag{14}$$

$$T_{min}(t_{0,min}) = 0°C \tag{15}$$

where $t_{0,max}$ and $t_{0,min}$ indicate the day when maximum and minimum temperatures warm above freezing, respectively. The timing of a freshet has been previously defined in different ways, each with the goal of indicating when the spring snowmelt is strongly contributing to streamflow (Zhang et al., 2001; Vincent et al., 2015; Woo and Thorne, 2003; Burn et al., 2004). For each cluster and temperature perturbation, we define an indicator of freshet timing ($t_{freshet}$) as the day when the 30-day running mean of modelled streamflow rises to be halfway between the winter minimum flow ($Q_{min}$) and spring maximum flow ($Q_{max}$):

$$Q(t_{freshet}) = \frac{Q_{max} - Q_{min}}{2} \tag{16}$$

For each cluster and temperature perturbation, we also define the peak flow of the freshet to be the spring maximum flow, $Q_{max}$. By perturbing temperatures in the range of $-5°C \leq \Delta T \leq 5°C$, we can track how well the model is learning the links between the temperature transitions and the peak flow and timing of the snowmelt-driven freshets.

Glacier runoff is a key contributor to streamflow in many watersheds in the study region, and compared to non-glacier-fed rivers, glacier-fed rivers have enhanced streamflow in late summer due to glacier runoff contributions after much of the seasonal snowpack has melted (Jost et al., 2012; Moore et al., 2009; Naz et al., 2014; Comeau et al., 2009). Additionally, glacier runoff counteracts variability in precipitation as enhanced (suppressed) glacier melt compensates for less (more) precipitation during hot and dry (cold and wet) years, leading to reduced interannual variability of total summer streamflow (Fountain and Tangborn, 1985; Meier and Tangborn, 1961). These effects lead to spatiotemporal patterns of summer streamflow in glacier-fed rivers which are markedly different than those in non-glacier-fed rivers (Anderson and Radić, 2020). Therefore, the model should learn a unique mapping of late summer climatic drivers to streamflow for glacier-fed rivers as

compared to non-glacier-fed rivers, and the difference in these mappings can be exploited to interpret model learning. In particular, since temperature is a strong control of melt, we assume that mean August streamflow ($Q_{Aug}$) is positively related to mean August temperature ($T_{Aug}$) in basins with partial glacier coverage. Again, while this is a simplification of the actual glacier melt processes, it is a key assumption in widely used temperature index melt models and is supported by empirical evidence in the study region (Stahl and Moore, 2006; Moore et al., 2009). We introduce the following hypothesis: if the model is learning to represent physical processes which drive streamflow in August, then modelled $Q_{Aug}$ in glacier-fed rivers should increase with increasing $T_{Aug}$, while modelled $Q_{Aug}$ in non-glacier-fed rivers should not increase with increasing $T_{Aug}$. To test this hypothesis, we introduce a spatially uniform temperature perturbation to only days in August, $\Delta T_{Aug}$, and add it to the maximum and minimum temperature channels. We then compute $Q_{Aug}$ for each station. We perturb August temperatures from $-5°C \leq \Delta T_{Aug} \leq 5°C$ with an increment of 1°C and use linear regression to estimate the sensitivity $\frac{\partial Q_{Aug}}{\partial T_{Aug}}$ for each station as:

$$Q_{Aug} = \frac{\partial Q_{Aug}}{\partial T_{Aug}} T_{Aug} + c + \varepsilon(T_{Aug}) \tag{17}$$

where $\frac{\partial Q_{Aug}}{\partial T_{Aug}}$ is calculated as the slope of the linear regression, $c$ is a constant coefficient (intercept), and $\varepsilon(T_{Aug})$ is the error. We compute basin glacier cover, $G$, for each stream gauge station as:

$$G = \frac{A_{glaciers}}{A_{basin}} \tag{18}$$

where $A_{glaciers}$ is the total area of glaciers within the watershed boundaries and $A_{basin}$ is the basin drainage area as reported in HYDAT (Water Survey of Canada HYDAT data). To calculate $A_{glaciers}$, we determine which glacier outlines fall within the watershed boundaries and then sum their areas, where glacier locations and areas are taken from the Randolph Glacier Inventory Version 6 (RGI Consortium, 2017).

# 5 Results

## 5.1 Evaluation of NSE

For each station, we derive ensemble-mean streamflow for the bulk model runs and fine-tuned model runs. The median fine-tuned NSE calculated over the test period is 0.68, and 35% of stream gauges have NSE > 0.8 (Fig. 4a). We compare the performance of the bulk versus fine-tuned models by looking at the difference in NSE between the bulk and fine-tuned models ($\Delta NSE$), evaluated across stations for each cluster (Fig. 5a) and in space (Fig. A5a). We find that overall, there is a small increase in NSE, with a median $\Delta NSE = 0.02$. The best performing stations are those in the central, southern, and north-western clusters, all of which have snowmelt dominated streamflow regimes throughout BC (Fig. 2). For these clusters, which represent a majority of stations, there is relatively little change in NSE between the bulk and fine-tuned models (Fig.

5a). The eastern cluster, which is made up of stations in the Prairie region, has the worst overall performance and shows slight improvements after fine-tuning. The coastal cluster, which is made up of rainfall dominated stations along the west coast, has a relatively narrow range of NSE and shows the largest improvement from fine-tuning. The north-eastern cluster, which is characterized as having comparable snowmelt- and rainfall-driven peaks in spring and summer, respectively, also shows a notable improvement from fine-tuning. Importantly, the median NSE is consistent across model runs in the fine-tuned

ensemble, with a range of only 0.05 across all 10 fine-tuned model runs. This result indicates that in terms of NSE, the fine-tuned model runs perform similarly as evaluated across the whole region.

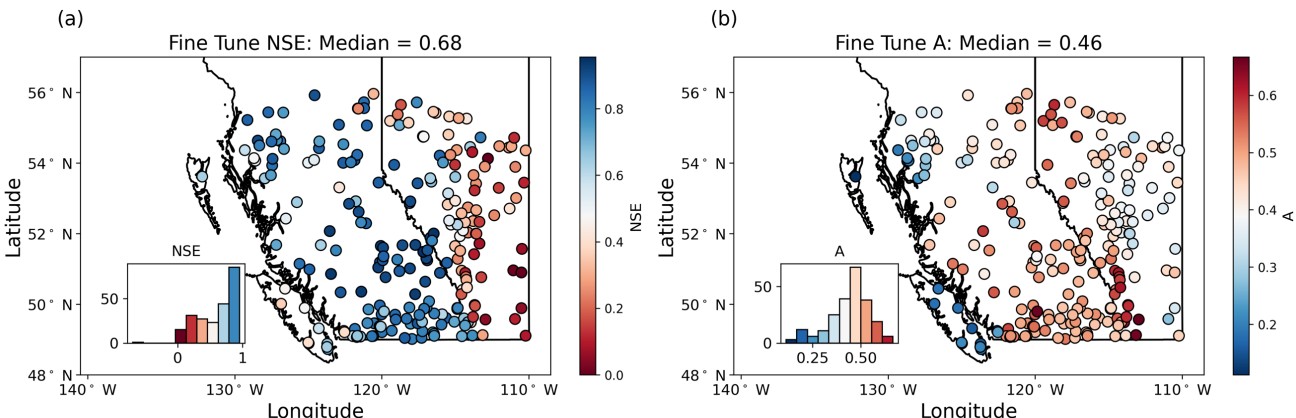

**Figure 4: NSE (a) and sensitive area *A* (b) as calculated over the test period for the fine-tuned model ensemble mean.** The insets show histograms of NSE and *A* across all stations. NSE values are greatest (indicating the best model performance) throughout mainland British Columbia, and are smallest (indicating the worst model performance) in south-eastern Alberta. *A* is smallest (indicating small sensitive areas) in the south-west and north-west coastal regions in British Columbia, and is largest (indicating large sensitive areas) throughout the rest of British Columbia and near the Alberta border. The colourmap

in (a) is clipped at NSE = 0 for better visualization and is justified since only two stations have $NSE < 0$.

To illustrate the model's performance in simulating different streamflow regimes, we compare the fine-tuned model output between a station with a snowmelt dominated regime and a station with a rainfall dominated regime (Fig. 6). The snowmelt dominated station is well simulated by the ensemble-mean ($NSE = 0.87$), capturing the timing and magnitude of many daily or weekly scale streamflow peaks; however, the $2\sigma$ interval is not consistently narrow throughout the year (Fig.

6a). Rather, it is smallest in the low-flow periods and freshet, and then larger during the recession over spring and summer. The modelled streamflow for the rainfall dominated station yields a lower NSE than the snowmelt dominated station ($NSE = 0.59$), despite the ensemble-mean being able to correctly model the timing of most rainfall-induced onset, peaks, and decay.

However, the peak magnitude in streamflow is often under- or over-estimated, particularly for the largest observed peaks (Fig. 6b). The $2\sigma$ interval is relatively narrow throughout the year, indicating that the 10 fine-tuned models output relatively similar streamflow.

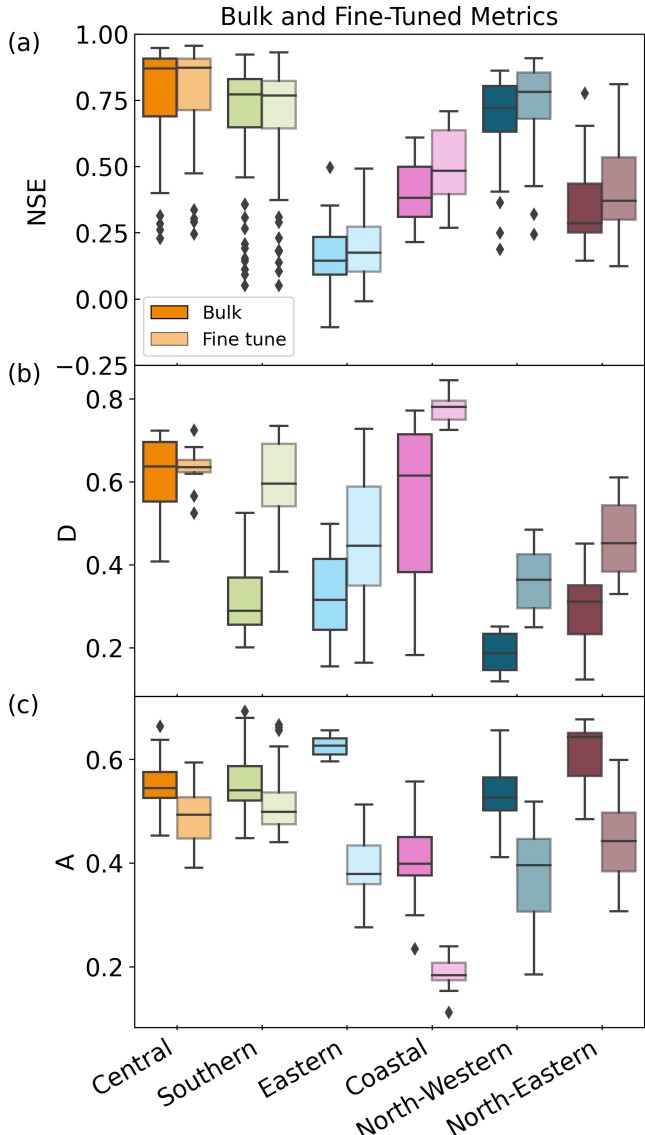

**Figure 5: Comparison of metrics for both the bulk and fine-tuned models for each cluster of stream gauge stations, coloured according to the clusters shown in Figure 2.** a) NSE of modelled streamflow of each stream gauge station ($n = 226$ across all six clusters). The central and southern clusters show the least amount of change between the bulk and fine-tuned models, while all other clusters increase NSE through fine-tuning (indicating improved model performance). For

readability, the y-axis was clipped at $NSE = -0.25$; however, one station in the Eastern cluster is below this threshold for both bulk and fine-tuned models ($NSE = -2.04$ in the bulk model and $NSE = -0.70$ in the fine-tuned model). This station is still included in all analyses and is not shown here for readability. b) The D-statistic for each model run for both bulk and fine-tuned model types ($n = 10$ for each cluster). In the central cluster, the variance of $D$ across model runs decreases through fine-tuning, indicating improved consistency between the fine-tuned central models. In all other clusters, $D$ increases, indicating improved separation between information which is near/within basins as compared to information which is further away. c) Sensitive area of the input as evaluated for each stream gauge station ($n = 226$ across all six clusters). In all clusters, $A$ decreases through fine-tuning, indicating that fine-tuned models are sensitive to smaller areas of the input as compared to the bulk models.

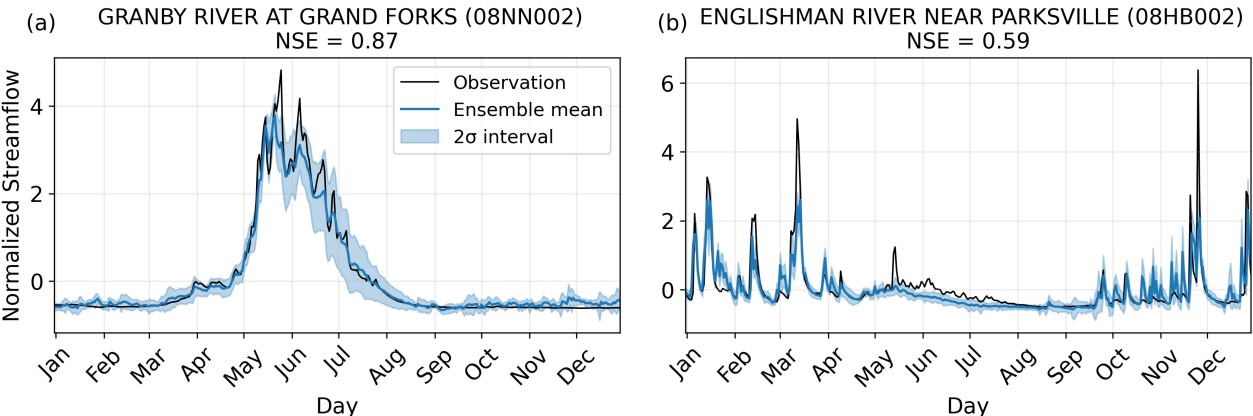

**Figure 6: Examples of observed and modelled streamflow for one year at two stations of different streamflow regimes.** The ensemble mean is the mean across the 10 model runs, and the shaded area is plus/minus two standard deviations across the 10 model runs. The station with the fifth-highest NSE is chosen in each of the southern and coastal clusters, which are snowmelt- and rain-dominated clusters, respectively. We chose the fifth-best performing station as to show more typical model performance for these clusters. An arbitrary year in the test set is chosen.

## 5.2 Evaluation of interpretability

### 5.2.1 Spatial perturbations

Stream gauge stations, watershed boundaries, and sensitivity maps for each cluster are displayed in Fig. 7. The central, southern, and coastal regions are most sensitive in the areas near and within the watersheds of the cluster, which means that information nearby the watersheds is most important for the model to predict streamflow. The models' predictions are less sensitive to perturbations further from the watersheds as indicated by the low values of $S(x, y)$ in Alberta for the coastal cluster and in northern British Columbia for the southern and central clusters. This result indicates that information far from the

650 watersheds is less important for the models' decision making. In contrast, the eastern and north-eastern clusters are sensitive both within the watersheds of the cluster and at a second sensitive area along the west coast of British Columbia. These findings indicate that models for these latter clusters are relying on links across space and time which may be present between the input and output datasets, but which may not be physically driving streamflow; consequently, long-term forecasting may not be appropriate as these links may not hold in the future. Another possible explanation is that there could be temporal

patterns of sensitivity. For example, the eastern and north-eastern regions may be sensitive to coastal conditions when storms travel from west to east. Alternatively, the sensitivity maps may be most sensitive to coastal conditions during winter, when the model could be tracking above-freezing temperatures. Future work should investigate these links further to evaluate their meaning and implications for CNN-LSTM performance.

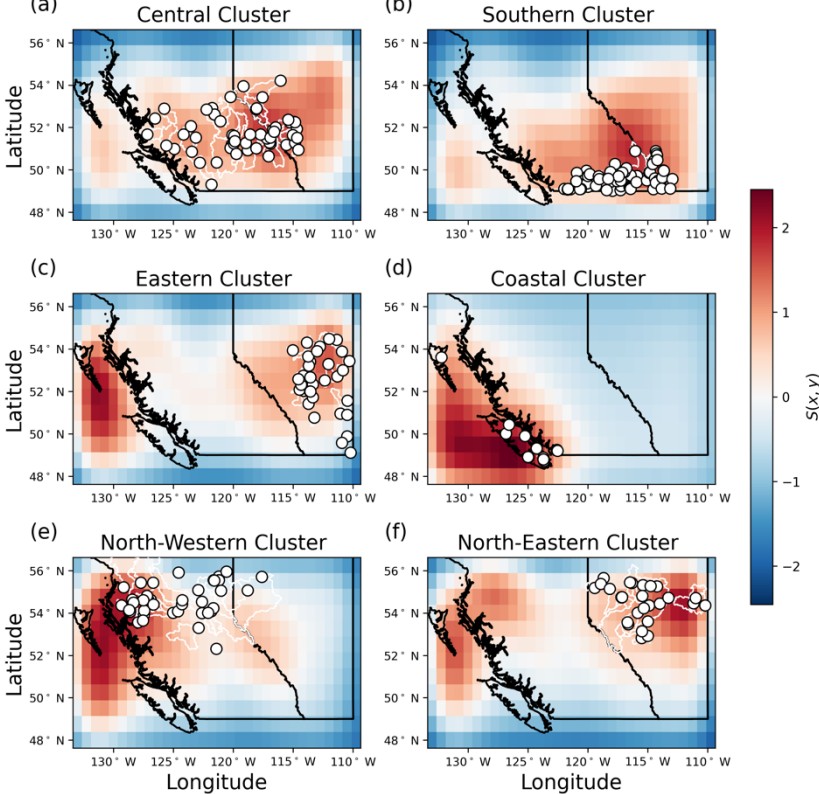

**Figure 7: Ensemble mean sensitivity maps.** For each cluster, the mean sensitivity map $S(x, y)$ is calculated across all stations and all days in the test period. The (a) central, (b) southern, (d) coastal, and (e) north-western clusters are generally most sensitive in the areas nearest the basins where streamflow is being predicted. The (c) eastern and (f) north-eastern clusters

are most sensitive to perturbations both near the stations being predicted and further away along the west coast.

The comparison of sensitivity between regions which are within/near the watersheds versus areas which are outside/far from the watersheds are summarized for each cluster (Fig. 8). The steps to calculate the D-statistic for one cluster is shown in Fig. A6. The difference between the within/near and outside/far sensitivity distributions are relatively large for the snowmelt-dominated stations in the central, southern, eastern, and north-eastern clusters (D values of 0.69, 0.66, 0.54, and 0.52, respectively), with the mean sensitivity being higher within/near than outside the watersheds (Fig. 8). The sensitivity distributions are also different for the coastal cluster (D = 0.77), where regions within/near the coastal watersheds are substantially more sensitive to perturbations than the regions outside the watersheds. Stations in the north-western cluster have the lowest D value relative to other clusters (D = 0.40), with the sensitivity near/within the watersheds not being substantially different from the sensitivity outside the watersheds.

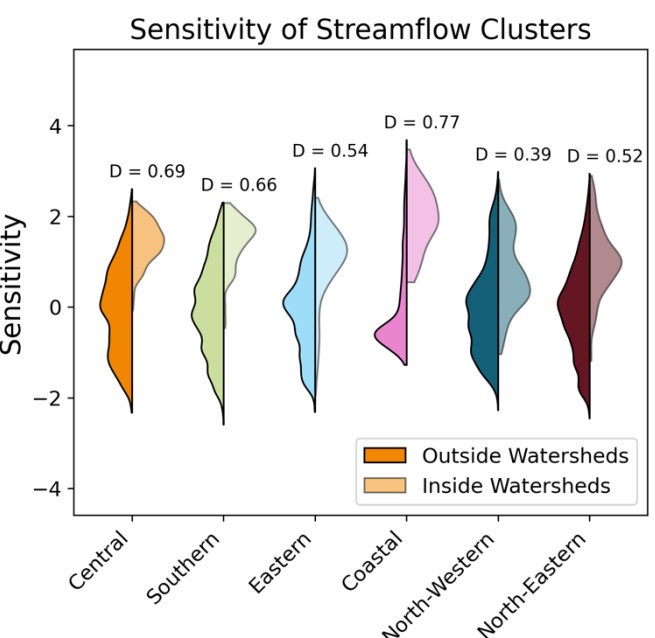

**Figure 8: The sensitivity distributions for inside/near and outside/far from the cluster watersheds.** Distribution pairs are labelled by their corresponding D-statistic from the Kolmogorov-Smirnov test. Clusters are coloured according to Fig. 2, and the cluster watershed regions are shown in Fig. A7. While all clusters are more sensitive to perturbations within/near their watershed regions, the coastal cluster demonstrates the greatest difference in sensitivity between within/near the watersheds and the rest of the domain.

The D-statistic is further evaluated by comparing both the bulk and fine-tuned models (Fig. 5b). All clusters except for the central cluster show an increase in D through fine-tuning, and in particular, southern stations show the largest increase in

D. Because D is a metric which indicates how different the inside/outside sensitivity distributions are from one another, the widespread increase in D through fine-tuning indicates that fine-tuning helps the model separate more relevant information (within/near watershed regions having higher sensitivity) from less relevant information (outside/far from watershed regions having lower sensitivity).

The eastern, coastal, and north-western stations have smaller sensitive areas, while the southern, central, and north-eastern stations have larger sensitive areas (Fig. 4b and Fig. 5c). We also compare the values of $A$ between the bulk and fine-tuned models per cluster (Fig. 5c) and in space (Fig. A5b). Here, all clusters have their mean $A$ decrease with fine-tuning and the median difference in $A$ between the bulk and fine-tuned models is $\Delta A = -0.09$, meaning that the fine-tuned models are sensitive to smaller areas of the input relative to the bulk model. Because $A$ is a metric which indicates the area that is most sensitive to perturbation, the widespread decrease in $A$ indicates that the process of fine-tuning helps the model to focus on smaller areas of the input space. Notably, the clusters which are sensitive to the smallest areas of the input (eastern, coastal, and north-western, Fig. 4b) all experience a substantial decrease in A through fine-tuning (Fig. A5b and Fig. 5c). This indicates that fine-tuning may be necessary for the model to focus on small areas of the input space.

It has been shown that fine-tuning an LSTM-based hydrological model can lead to a moderate improvement in performance which is heterogeneous in space (Kratzert et al., 2018). Here we build on this understanding to show also that fine-tuning substantially influences what the model is learning. Comparing the results between clusters in terms of NSE, $D$, and $A$, we find that the process of fine-tuning does not impact model performance equally across all clusters (Fig. 5). Specifically, the central cluster is the least impacted by fine-tuning, as indicated by the small differences in NSE, $D$, and $A$ between the bulk and fine-tuned models. The bulk model focuses on large areas of the input, and since the central cluster spans a large area of the input space (the largest area of all the clusters), the bulk model is already effective in learning the weather-to-streamflow mapping and further fine-tuning does not substantially change its learning. On the other hand, the coastal cluster is more impacted by fine-tuning, with simulated streamflow better matching observations (increase in NSE) with more realistic learning (increase in $D$ and a substantial decrease in $A$). In other words, fine-tuning has made the model focus on smaller regions nearby the watersheds, which has led to better performance. Considering the processes which drive streamflow, fine-tuning has minimal impact on NSE at southern and central clusters (which are melt dominated flow), and most improves NSE at the coastal, north-western, north-eastern, and eastern clusters (where a rain-driven flow peak is present in the seasonal hydrograph) (Fig. 2 and Fig. 5a). Seeing as fine-tuning also more substantially decreases $A$ at these latter clusters, and that rainfall occurs over smaller spatial scales as compared to snowmelt, we suggest that the process of fine-tuning allows the model to better learn rainfall-runoff processes.

Interestingly, it is not necessarily true that the model run which performs best according to NSE is the model which has best learned to focus within the watersheds being predicted. All models in the fine-tuned ensemble achieve similar performance as evaluated by the median NSE (range of 0.05 across the 10 models) while there is a much larger range of median $D$ (range of 0.25 across the 10 models). Furthermore, NSE and $D$ are not significantly correlated (correlation coefficient $R = 0.05$ with p-value $> 0.1$). In fact, the model with the highest median NSE has the lowest median $D$ ($NSE = 0.69$ and $D =$

0.41).  The range in $D$ across the models in the ensemble and the lack of significant correlation between NSE and $D$ indicate that while all model runs can output streamflow to a similar degree of accuracy, the internal learning processes are different among model runs.

### 5.2.2 Annual temperature perturbations

The peak flow and timing of the modelled freshet, as well as the timing of transition from below- to above-freezing temperature, reveals characteristic patterns of snowmelt for the snowmelt dominated clusters (Fig. 9).  When no temperature perturbation is added (e.g. $\Delta T = 0$), the snowmelt driven streamflow in southern, central, north-western, north-eastern, and eastern clusters experience a large increase in modelled streamflow after temperatures increase above freezing.  For these snowmelt driven streamflow regimes, a positive temperature perturbation ($\Delta T > 0°C$) advances the freshet timing indicator while the peak flow of the freshet decreases (Fig. 10).  A possible physical interpretation of this result is that a warmer climate would lead to both a smaller fraction of precipitation falling as snow rather than as rain and a shorter cold season, leading to a thinner seasonal snowpack, an earlier onset to melt, and less water to drive streamflow in spring.  Similarly, a decrease in temperature ($\Delta T < 0°C$) delays the freshet timing indicator while the peak flow of the freshet increases (Fig. 10).  Notably, fall and winter streamflow is suppressed when temperatures are lowered, and enhanced when temperatures are raised in the more rainfall driven coastal and north-western clusters (approximately before April and after December; Fig. 9d and 9e).  These results are consistent with the rationale that a colder climate would lead to both a larger fraction of precipitation falling as snow rather than rain and a longer cold season, building a deeper snowpack which can deliver a larger volume of water to streamflow in spring.  Similarly, when temperatures are raised ($\Delta T > 0°C$), winter and fall streamflow increases, which can physically be explained by more precipitation falling as rain which leads to a faster streamflow response.  Importantly, while the freshet timing indicator relative to the timing of above-freezing maximum/minimum temperatures may not be the same for all clusters (Fig. 9 and Fig. 10), all clusters respond similarly to a change in the timing of this temperature transition, and this response is consistent with a physical understanding of the drivers of streamflow.

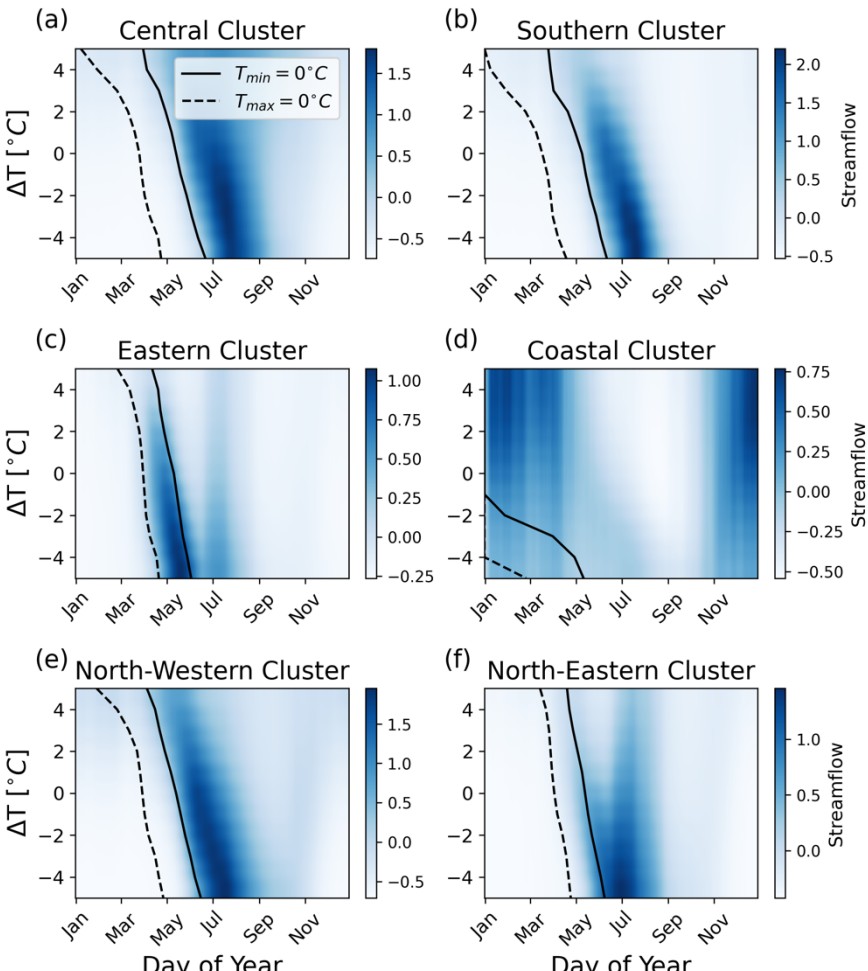

**Figure 9: Modelled streamflow for a range of temperature perturbations, averaged across all stream gauges in each cluster and all years in the test set.** Black lines indicate the transition from below-freezing to above-freezing maximum (dashed) and minimum (solid) daily temperatures. In the (a) central, (b) southern, (c) eastern, (e) north-western, and (f) north-eastern clusters, $\Delta T > 0°C$ leads to an earlier freshet with a smaller peak flow, while $\Delta T < 0°C$ leads to a later freshet with a larger peak flow. In the (d) coastal cluster, $\Delta T > 0°C$ leads to enhanced streamflow in winter and fall and suppressed streamflow in summer, while $\Delta T < 0°C$ leads to suppressed streamflow in winter and fall and enhanced streamflow in summer.

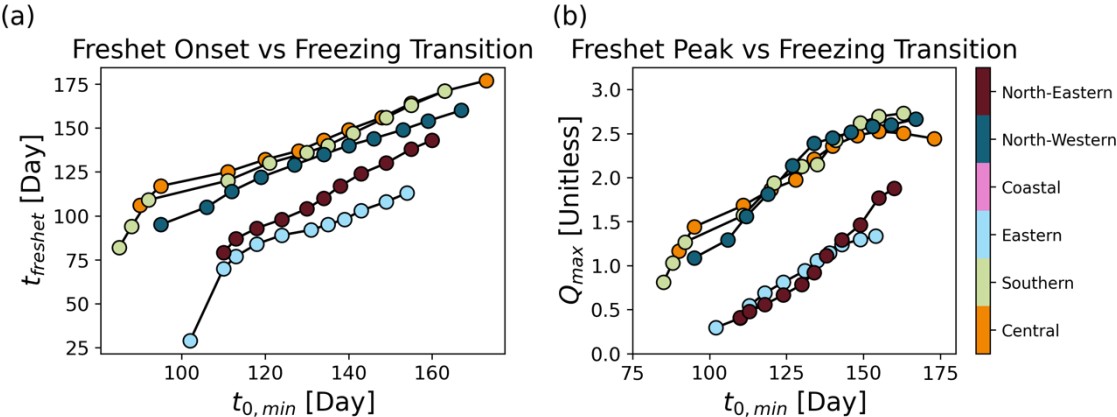

**Figure 10: Modelled a) freshet timing indicator and b) peak magnitude are both positively correlated with the day that minimum temperatures rise above freezing.** The coastal cluster is not shown as it is dominated by winter rainfall rather than a spring freshet.

### 5.2.3 August temperature perturbations

When August temperatures are perturbed with $\Delta T_{Aug} > 0$, modelled mean August streamflow in partially glacierized watersheds increases, while when August temperatures are perturbed with $\Delta T_{Aug} < 0$, modelled mean August streamflow in partially glacierized watersheds decreases. This is indicated by $\frac{\partial Q_{Aug}}{\partial T_{Aug}} > 0$ for stations where watershed glacier cover is non-zero (Figure 11). In contrast, perturbations of mean August temperature (positive or negative) do not (or negligibly) influence modelled $Q_{Aug}$ for stations where watersheds have no glacier coverage, which is indicated by $\frac{\partial Q_{Aug}}{\partial T_{Aug}}$ being narrowly distributed around zero for these stations (Figure 11). Additionally, we investigate how $\frac{\partial Q_{Aug}}{\partial T_{Aug}}$ varies for three ranges of watershed glacier cover, $G$, here defined as light glacier cover ($0\% < G \leq 1\%$), moderate glacier cover ($1\% < G \leq 10\%$), and substantial glacier cover ($10\% < G \leq 100\%$). We find that the median $\frac{\partial Q_{Aug}}{\partial T_{Aug}}$ increases as $G$ increases from light, to moderate, to substantial glacier cover (Figure 11b), indicating that mean August streamflow is more sensitive to August temperature perturbations at higher glacier coverage.

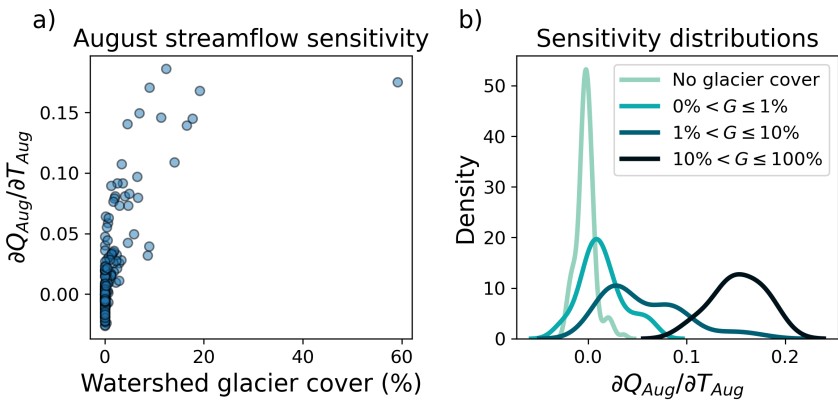

**Figure 11: Modelled sensitivity of mean August streamflow to mean August temperature.** a) $\frac{\partial Q_{Aug}}{\partial T_{Aug}}$ increases non-linearly with watershed glacier cover, $G$, indicating that greater watershed glacier coverage is related to more positive $\frac{\partial Q_{Aug}}{\partial T_{Aug}}$. b) Probability distributions of $\frac{\partial Q_{Aug}}{\partial T_{Aug}}$ for different ranges of watershed glacier coverage, indicating that $\frac{\partial Q_{Aug}}{\partial T_{Aug}}$ for glacier-fed rivers is both greater than non-glacier-fed rivers, and greater at increasing glacier coverage. All probability distributions are normalized to have unit area.

## 6 Discussion

Our model achieves comparable performance to previous studies which have used deep learning for modelling streamflow across a region using meteorological inputs; for example, LSTMs have been used to achieve median NSE of 0.72 across 241 catchments in the United States (Kratzert et al., 2018) with the worst performance in the more arid regions (similar to our model's poor performance in the eastern cluster). Additionally, Kratzert et al. (2019) achieved a median NSE of 0.63 across 531 catchments in the United States and found that model performance was improved (to median NSE of 0.74 across 531 catchments) when catchment characteristics were included in the input to incorporate information related to the climate, topography, vegetation, and subsurface. The CNN-LSTM modelling framework introduced here could be extended to include spatially discretized variables which are constant in time (such as topography and permeability) by using these variables as additional channels in each frame of the input video, for example.

One of our key findings is that the model performs well (high NSE values) in all clusters except for the eastern cluster. We compare our results with findings from process-based models used to predict streamflow at 45 of the same stations as in our study (Table 2). We identify studies which modelled streamflow at daily temporal resolution, and evaluated performance using NSE or correlation between observed and modelled streamflow over at least a one-year period. In total we selected 45 stations for this comparison, keeping in mind that this is not an exhaustive comparison of all studies across the region, nor do

we claim that the identified studies are necessarily directly comparable with our results as each process-based model defines calibration and validation periods differently. We note a difference in terminology between the process-based model results and our CNN-LSTM results. Both evaluate models on 'unseen data' that were not used to determine the model parameters; however, the process-based models refer to this dataset as 'validation data' while we refer to this dataset as 'testing data'. As our goals are to explore the CNN-LSTM model architecture and interpret its decision making, and not necessarily to outperform existing process-based models, we do not need to compare every individual station to process-based models. The intercomparison shows that the CNN-LSTM model performance is at least similar to, and often outperforms, many process-based models existing in the literature for the southern, central, coastal, north-western, and north-eastern clusters (Table 2). Our model achieves higher values of NSE or correlation at 37 of the 45 identified stream gauge stations, indicating generally good performance in all clusters except for the eastern cluster.

Prior studies have modelled daily streamflow at the Englishman River near Parksville (08HB002), one of the locations in our study; for example, Fleming et al. (2015) use an ensemble of ANNs to forecast streamflow and achieve NSE values in the range of 0.7 – 0.8, while Lima et al. (2016) use nonlinear extreme learning machines and achieve NSE > 0.8. These examples outperform the NSE value of 0.59 achieved by our CNN-LSTM. Their success could be in part due to the inclusion of more locally-specific input data (e.g. Fleming et al. (2015) include snow pillow and antecedent streamflow data, while Lima et al. (2016) include predictors such as sea level pressure, wind speed, and humidity, among others), a decision which can be more easily implemented for modelling at a single stream gauge station as compared to a regional scale model. These examples highlight what may be a trade-off between scale and detail in the modelling approach, where the advantage of simultaneous streamflow modelling at multiple stream gauge stations across a region as done by the CNN-LSTM network may be met by the disadvantage of weaker performance on one particular river of interest.

We compare our fine-tuned CNN-LSTM models against linear models to evaluate the extent to which the nonlinearities introduced by the CNN-LSTM approach improve streamflow predictions. We create an ensemble of 10 linear models for each cluster of stream gauge stations. Each linear model is a fully-connected ANN with an input layer, an output layer, and linear activation functions (essentially reducing to multiple linear regression). We use the same training, validation, and testing data as in the CNN-LSTM approach. The CNN-LSTM is designed to receive an input structured as a weather video, while in comparison, ANNs are designed to receive an input structured as a single vector. The input neurons in the ANN correspond to each variable at each grid point and each day in a single weather video, meaning that there are 420,480 input neurons. For example, the input to predict flow on September 30, 2011 is daily maximum temperature, minimum temperature, and precipitation from September 30, 2010 through September 29, 2011, at each grid point in the study region. For the CNN-LSTM, these data are structured as a weather video with shape $365 \times 12 \times 32 \times 3$ (e.g. day × latitude × longitude × variable), but for the ANN, these data are structured as a vector with length 420,480. The target output is the next day of streamflow at all stations in the cluster. Therefore, for each model for cluster $i$, there are 420,480 input neurons and $N$ output neurons (where $N$ is the number of stations in cluster $i$). This approach was chosen to keep as much similarity as possible between the CNN-LSTM and linear model setup. The two approaches use the same input data, the same target data,

and the same number of ensemble members, while the key difference is the nonlinearity and architecture of the CNN-LSTM model. We find that the CNN-LSTM model outperforms this simple linear benchmark, achieving a greater NSE at 222 out of 226 stations. The linear model has a minimum NSE of -13.33, a median NSE of 0.35, and a maximum NSE of 0.76, while the CNN-LSTM model has a minimum NSE of -0.7, a median NSE of 0.68, and a maximum NSE of 0.96.

The eastern cluster is unique among our six clusters in terms of our model's poor performance, and also in terms of
835 the region's hydrology and a lack of studies in the literature at the same locations and with the same metrics as our study for direct comparison. One possible contribution to our model's inability to successfully simulate streamflow in the eastern cluster (Fig. 5a) could be the effect of non-contributing areas within Prairie basins. Prairie topography is characterized by small surface depressions which result in intermittent water connectivity and variable sized drainage basins (Shaw et al., 2012). When the depressions are not full, rainfall and snowmelt can be stored in upstream depressions rather than contributing to
840 streamflow (i.e. non-contributing areas), and so there may not be a substantial streamflow response even if there is rainfall or snowmelt within the basin. In the eastern cluster, 30 out of 34 basins have non-contributing areas, ranging from 1% to 79% of the total basin area with a mean of 20% of the total basin area not contributing to streamflow on average (Government of Canada, 2020). Additionally, there is hysteresis between contributing area and water storage within these depressions, meaning that the area which contributes to streamflow is determined by both the presence of depressional storage and if the storage is
845 increasing (wetting) or decreasing (drying) (Shook and Pomeroy, 2011). Storage in ponds can vary on both seasonal and decadal timescales (Shaw et al., 2012; Hayashi et al., 2016), but only a single year of daily temperature and precipitation is used to predict the next day of streamflow. It could be that the CNN-LSTM model cannot accurately predict the streamflow response in eastern basins because one year of temperature and precipitation is insufficient information to know the state of depressional storage (e.g. seasonal and decadal fluctuations in wetting or drying). Several studies have developed process-
850 based models for application to different stations than ours in the Prairie region, which have outperformed our DL-based approach. While our model had a median $NSE = 0.17$, process-based models achieve higher values, for example, $NSE > 0.4$ (Unduche et al., 2018; Mengistu and Spence, 2016) and $NSE > 0.7$ (Muhammad et al., 2019).

Considering the complexity of hydrology in real catchments and the dependence of streamflow on locally resolved processes, it is possible that the model's performance is limited by an inability to learn processes beyond those which could
be better inferred from streamflow, temperature, and precipitation, such as advective fluxes (e.g. wind transport of snow), evaporative fluxes (e.g. sublimation of the snowpack, evapotranspiration), and interactions between ground and surface water. Nevertheless, the model could still be successful in regions where the importance of such processes is less than those which can be inferred from coarse resolution temperature and precipitation alone. For example, winters are generally warmer and wetter in British Columbia (where the model performs better) as compared to Alberta (where the model performs worse),
which may limit the importance of processes such as blowing snow transport and sublimation by increasing cohesion of the snowpack (Pomeroy and Gray, 1990).

When the input temperature series is made warmer (cooler), the indicator of freshet timing and peak flow advances (delays) and decreases (increases) respectively (Fig. 9 and Fig. 10). This finding is consistent with previous studies of climate

change impacts in the region. For example, Shrestha et al. (2012) used the macro-scale Variable Infiltration Capacity (VIC) hydrological model forced by a suite of global climate models in the Fraser River Basin (which spans the central cluster in our study), finding that spring peak flows occur earlier in the year and with lower magnitude under a warmer future climate (Shrestha et al., 2012). Schnorbus et al. (2014) used the VIC model to project streamflow in the Peace, Campbell, and Columbia River watershed in British Columbia (primarily in the north-western, coastal, central, and southern clusters in our study) under a range of climate change scenarios (Schnorbus et al., 2014). They found greater spring flows and lower summer flows in the snowmelt dominated locations, while the coastal location was projected to experience enhanced winter flows and depressed summer flows. It is promising that not only does the CNN-LSTM model perform well in the historical period (e.g. the test period of 2011 - 2015), but produces conceptually similar projections for a warmer climate as compared to existing physically-based models.

When August temperatures are made warmer (cooler), modelled streamflow in partially glacierized watersheds increases (decreases) and the sensitivity of modelled August streamflow to these temperature perturbations is greater in more glacierized watersheds as compared to less glacierized watersheds (Figure 11). The positive relationship between $Q_{Aug}$ and $T_{Aug}$ in glacierized watersheds indicates that the model has learned that glacierized watersheds have an input to streamflow which is positively related to temperature in August, while non-glacierized watersheds do not. We interpret this result as the model learning to represent glacier runoff as a temperature-dependent source. Interestingly, the relationship between $\frac{\partial Q_{Aug}}{\partial T_{Aug}}$ and watershed glacier cover as derived from the sensitivity test of the CNN-LSTM model (Figure 11a), is similar in form to an empirically derived relationship between $\frac{\partial Q_{Aug}}{\partial T_{Aug}}$ and watershed glacier cover in British Columbia (Figure 5 in Moore et al. (2009), from analysis in Stahl and Moore (2006)). Both analyses identify a positive non-linear relationship between $\frac{\partial Q_{Aug}}{\partial T_{Aug}}$ and $G$ when $G > 0$, while $\frac{\partial Q_{Aug}}{\partial T_{Aug}}$ is distributed around zero when $G = 0$. Note however that the raw values of $\frac{\partial Q_{Aug}}{\partial T_{Aug}}$ differ between our approach and that in Figure 5 of Moore et al. (2009) due to differing normalization schemes. While it is interesting that the model has learned the unique characteristics of temperature-driven August flow of glacierized watersheds, it also highlights a challenge when applying the CNN-LSTM model in its current realization for applications such as long-term forecasting under climate change. Under warmer future climate forcing, the model would associate higher temperatures with greater flow. However, projections of future glacier volume indicate that 70-90% of glacier ice volume will be lost by 2100 in Western Canada (Clarke et al., 2015; Marshall et al., 2011), and so it is expected that the learned temperature-flow relationship from the past will no longer hold under such conditions.

It is notable that the CNN-LSTM model achieves good streamflow simulation with only coarse resolution climate forcing data and localized streamflow data, with no knowledge of features such as basin characteristics, topography, or land use, and no explicit climate downscaling steps. Our model uses forcing data at relatively coarse spatial resolution (0.75° x 0.75°, or ~75 km resolution) as compared to studies identified in Table 2 (e.g. 0.0625° x 0.0.0625° in (Shrestha et al., 2012);

10 km resolution in (Eum et al., 2017)).  Studies that employ a climate downscaling step first map coarse resolution climate data to fine resolution climate data, and then map the downscaled fine resolution climate data to streamflow.  Here, the CNN-LSTM is effectively representing a single transfer function that maps coarse resolution climate data directly to streamflow, and it is possible that an effective downscaling of climate data is learned by the model.  This indirect downscaling is plausible since statistical methods are often used for climate downscaling, including CNNs (Vandal et al., 2017).

## 7 Summary and conclusions

This study investigated the applicability of a sequential CNN-LSTM model for regional scale hydrological modelling, where the model was forced by gridded climate data to predict streamflow at multiple stream gauge stations simultaneously. We focused on using a relatively simple deep learning model, with the input data represented by temperature and precipitation reanalysis given on relatively coarse spatial resolution (0.75° x 0.75°).  The deep learning model is used to predict daily streamflow between 1980 – 2015 at 226 stream gauge stations.  We investigated how well the model learns different streamflow regimes and how physically realistic the model's learning was for each streamflow regime.  To reach these goals, the model was trained, validated, and tested on a set of stream gauge stations across Western Canada, initially partitioned into six clusters based on the similarity in both seasonal streamflow and proximity in space.  A set of metrics was introduced and developed to evaluate the model performance and to investigate the model's learning.  We summarize the major findings as follows:

1) The model successfully simulated streamflow at multiple stations simultaneously, with a median NSE of 0.68 and 35% of stations having NSE > 0.8.  The best model performance was for stations with snowmelt dominated streamflow in British Columbia, and the worst performance was for the eastern cluster of stations in the Prairie region. The poor performance in the Prairie region may be due to the importance of processes which are underrepresented or not represented in the training data, such as processes occurring over longer than annual timescales, or at smaller spatial scales, or which are not able to be described from temperature and precipitation alone.

2) For a majority of stations, the model was most sensitive to perturbations in the input data prescribed near and within the basins being predicted, demonstrating that the model's spatial learning focused on areas where the physical drivers of streamflow are occurring.  For the eastern and north-eastern clusters, the model was sensitive to perturbations that are far from the watersheds where streamflow was being predicted, thus linking the streamflow to weather fields far away (> 500 km apart).  In these cases, the model may be more appropriate for short term rather than long term prediction, as the learned links over far distances may not hold in the future.

3) Fine-tuning by streamflow regime led to modest improvements in model performance as evaluated by NSE, but allowed the model to focus on smaller areas which are near and within the watersheds where streamflow is being

predicted. We conclude that fine-tuning is beneficial for directing the model to focus on the areas where streamflow-driving processes are taking place.

4) To investigate the learning of temporal patterns, we focused on the timing and peak flow of the spring freshet. By uniformly perturbing temperature input to drive the model with warmer and colder climates relative to present, the model responded by changing the peak flow and timing of the freshet in accordance with the timing of the transition from below- to above-freezing temperatures.

5) To investigate the learning of unique processes in partially glacierized basins, we focused on the sensitivity of August flow to August temperature. By increasing August temperature input to drive the model, the model responded by increasing August flow in partially glacierized basins while not increasing August flow in non-glacierized basins. The sensitivity of flow to temperature was found to be greater in more glacierized basins as compared to less glacierized basins.

The CNN-LSTM model presented has been able to explicitly incorporate both spatial and temporal information for predicting streamflow across a region. In addition to successfully simulating streamflow across a range of streamflow regimes, we are able to interpret key aspects of the model's learning. Interpretability of model learning builds trust in the model's predictions, leading to further applications whether for prediction or as a compliment to process based or empirical models. Considering that ERA5 climate reanalysis has global spatial coverage and is temporally complete to 1979, there are many opportunities to investigate the transferability of this approach to different regions, the use of different predictor variables, and the use of different spatial and temporal resolution of both input and target data.

**Table 2: Comparison to select process-based models.** Metrics for comparison are Nash-Sutcliffe Efficiency (NSE) or correlation coefficient (R). For our CNN-LSTM model, the performance metrics are calculated from the test set (between 2011 – 2015). For the reference models, performance metrics are as reported for the validation set (various validation periods). In bold is the better performing (higher) value between the two models.

| Station Name | Station ID | Cluster | Reference NSE | CNN-LSTM NSE | Reference R | CNN-LSTM R | Reference |
|---|---|---|---|---|---|---|---|
| Bridge River below Bridge Glacier | 08ME023 | Central | **0.95** | 0.94 | | | (Stahl et al., 2008) |
| Lillooet River near Pemberton | 08MG005 | Central | 0.70 | **0.88** | | | (Whitfield et al., 2002) |
| Quesnel River near Quesnel | 08KH006 | Central | 0.83 | **0.90** | | | (Shrestha et al., 2012) |

| River | Station | Region | | | | | Reference |
|---|---|---|---|---|---|---|---|
| North Thompson River at McLure | 08LB064 | Central | 0.85 | **0.92** | | | (Shrestha et al., 2012) |
| South Thompson River at Chase | 08LE031 | Central | 0.87 | **0.91** | | | (Shrestha et al., 2012) |
| Thompson River near Spences Bridge | 08LF051 | Central | 0.89 | **0.93** | | | (Shrestha et al., 2012) |
| Harrison River near Harrison Hot Springs | 08MG013 | Central | 0.66 | **0.83** | | | (Shrestha et al., 2012) |
| Columbia River at Nicholson | 08NA002 | Central | | | 0.899 | **0.980** | (Bingeman et al., 2006) |
| Kicking Horse River at Golden | 08NA006 | Central | 0.77 | **0.91** | 0.884 | **0.961** | (Bingeman et al., 2006; Schnorbus et al., 2011) |
| Columbia River at Donald | 08NB005 | Central | 0.91 | **0.96** | 0.924 | **0.984** | (Bingeman et al., 2006; Schnorbus et al., 2011) |
| Goldstream River below Old Camp Creek | 08ND012 | Central | | | 0.689 | **0.961** | (Bingeman et al., 2006) |
| Duncan River below B.B. Creek | 08NH119 | Central | | | 0.863 | **0.959** | (Bingeman et al., 2006) |
| Illecillewaet River at Greeley | 08ND013 | Central | | | 0.906 | **0.959** | (Bingeman et al., 2006) |
| Gold River above Palmer Creek | 08NB014 | Central | | | 0.813 | **0.957** | (Bingeman et al., 2006) |
| Split Creek at the Mouth | 08NB016 | Central | | | 0.744 | **0.954** | (Bingeman et al., 2006) |
| Miette River near Jasper | 07AA001 | Central | **0.86** | 0.84 | | | (Chernos et al., 2020) |
| Athabasca River near Jasper | 07AA002 | Central | **0.93** | 0.90 | | | (Chernos et al., 2020) |
| Athabasca River at Hinton | 07AD002 | Central | **0.91** | 0.88 | | | (Chernos et al., 2020) |
| Athabasca River near Windfall | 07AE001 | Central | 0.80 | **0.86** | | | (Eum et al., 2017) |
| Englishman River near Parksville | 08HB002 | Coastal | **0.65** | 0.59 | | | (Whitfield et al., 2002) |

| | | | | | | |
|---|---|---|---|---|---|---|
| Athabasca River at Athabasca | 07BE001 | North-Eastern | 0.75 | **0.81** | | (Eum et al., 2017) |
| Pembina River at Jarvie | 07BC002 | North-Eastern | 0.48 | **0.63** | | (Eum et al., 2017) |
| Stuart River near Fort St. James | 08JE001 | North-Western | 0.82 | **0.86** | | (Shrestha et al., 2012) |
| Fraser River at Shelley | 08KB001 | North-Western | 0.75 | **0.87** | | (Shrestha et al., 2012) |
| Omineca River near the Mouth | 07EC002 | North-Western | 0.81 | **0.89** | | (Schnorbus et al., 2011) |
| Parsnip River above Misinchinka River | 07EE007 | North-Western | 0.81 | **0.87** | | (Schnorbus et al., 2011) |
| Pine River at East Pine | 07FB001 | North-Western | 0.71 | **0.84** | | (Schnorbus et al., 2011) |
| Murray River above Wolverine River | 07FB006 | North-Western | 0.67 | **0.86** | | (Schnorbus et al., 2011) |
| Murray River near the Mouth | 07FB002 | North-Western | 0.58 | **0.86** | | (Schnorbus et al., 2011) |
| Sukunka River near the Mouth | 07FB003 | North-Western | 0.78 | **0.80** | | (Schnorbus et al., 2011) |
| Kuskanax Creek near Nakusp | 08NE006 | Southern | | | 0.819 | **0.968** | (Bingeman et al., 2006) |
| Kaslo River below Kemp Creek | 08NH005 | Southern | | | 0.864 | **0.938** | (Bingeman et al., 2006) |
| Barnes Creek near Needles | 08NE077 | Southern | | | 0.797 | **0.911** | (Bingeman et al., 2006) |
| Mather Creek below Houle Creek | 08NG076 | Southern | | | 0.795 | **0.868** | (Bingeman et al., 2006) |
| St. Mary River below Morris Creek | 08NG077 | Southern | | | 0.871 | **0.926** | (Bingeman et al., 2006) |
| Fry Creek below Carney Creek | 08NG130 | Southern | | | 0.816 | **0.936** | (Bingeman et al., 2006) |
| Keen Creek below Kyawats Creek | 08NH132 | Southern | | | 0.774 | **0.931** | (Bingeman et al., 2006) |

| | | | | | | | |
|---|---|---|---|---|---|---|---|
| Lemon Creek above South Lemon Creek | 08NJ160 | Southern | | | 0.784 | **0.945** | (Bingeman et al., 2006) |
| Kootenay River at Kootenay Crossing | 08NF001 | Southern | 0.75 | **0.89** | | | (Schnorbus et al., 2011) |
| Kootenay River at Fort Steele | 08NG065 | Southern | 0.85 | **0.86** | | | (Schnorbus et al., 2011) |
| Elk River at Fernie | 08NK002 | Southern | 0.81 **0.92** | 0.73 | | | (Schnorbus et al., 2011) (Chernos et al., 2017) |
| Elk River near Natal | 08NK016 | Southern | 0.75 **0.91** | 0.73 | | | (Schnorbus et al., 2011) (Chernos et al., 2017) |
| Fording River at the Mouth | 08NK018 | Southern | 0.72 **0.84** | 0.71 | | | (Schnorbus et al., 2011) (Chernos et al., 2017) |
| Slocan River near Crescent Valley | 08NJ013 | Southern | 0.78 | **0.88** | | | (Schnorbus et al., 2011) |
| Salmo River near Salmo | 08NE074 | Southern | 0.73 | **0.83** | | | (Schnorbus et al., 2011) |

**Data availability**

All data used in this study are publicly available. ERA reanalysis data are available from the European Centre for Medium-Range Weather Forecasts (ECMWF) (Hersbach et al., 2020). Streamflow data are available from the Environment Canada HYDAT database (Water Survey of Canada HYDAT data). Basin outlines are available from the Water Survey of Canada (Environment and Climate Change Canada, 2016). Provincial borders used in mapping are available from Statistics Canada (Statistics Canada, 2016).

**Code availability**

Code used in this study is available on Github (https://github.com/andersonsam/cnn_lstm_era) (Anderson, 2021).

## Author contributions

SA designed and conducted the study. SA and VR analysed the results and wrote the paper.

## Competing interests

The authors declare that they have no competing interests.

## Acknowledgements

This research was funded through the National Science and Engineering Research Council of Canada (NSERC).

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

1265

**Appendix A**

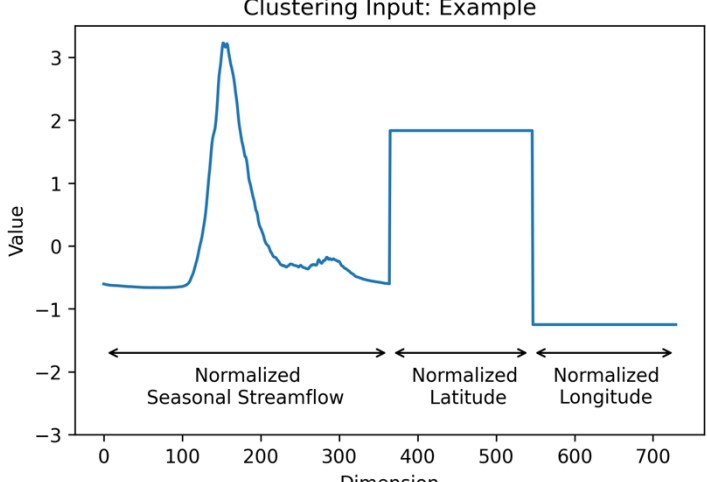

**Figure A1: Example of one observation input into clustering algorithm.** The first half of the input vector is the seasonal streamflow (normalized at each station), the third quarter is latitude (normalized across all stations), and the fourth quarter is longitude (normalized across all stations).

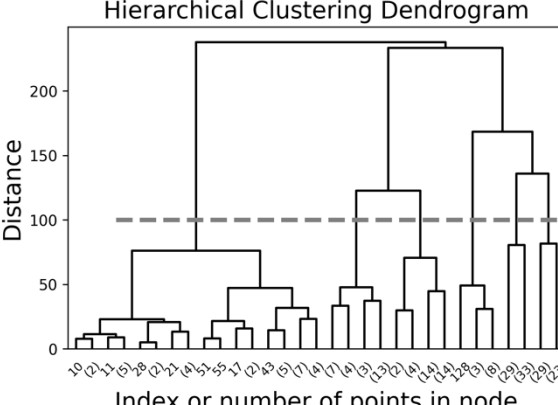

 **Figure A2: Dendrogram of stream gauge clustering.** The dashed grey line indicates the level at which the cluster members were grouped.

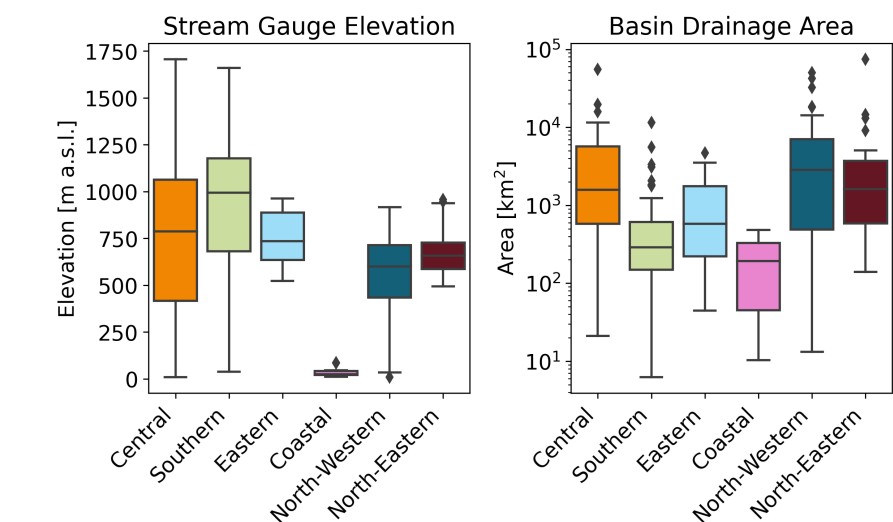

**Figure A3: Elevation and drainage area of stations within each of the identified clusters.** Station elevation is calculated from a digital elevation model from the Shuttle Radar Topography Mission (SRTM) at 90 m resolution (Farr et al., 2007). Drainage area is taken from the Environment Canada HYDAT database (Water Survey of Canada HYDAT data). The coastal cluster is at the lowest elevation and with the smallest drainage areas. Clusters in mainland British Columbia (central, southern, and north-western) span wide ranges of elevation and drainage area, while clusters in Alberta (eastern and north-eastern) span narrower ranges of elevation and drainage area.

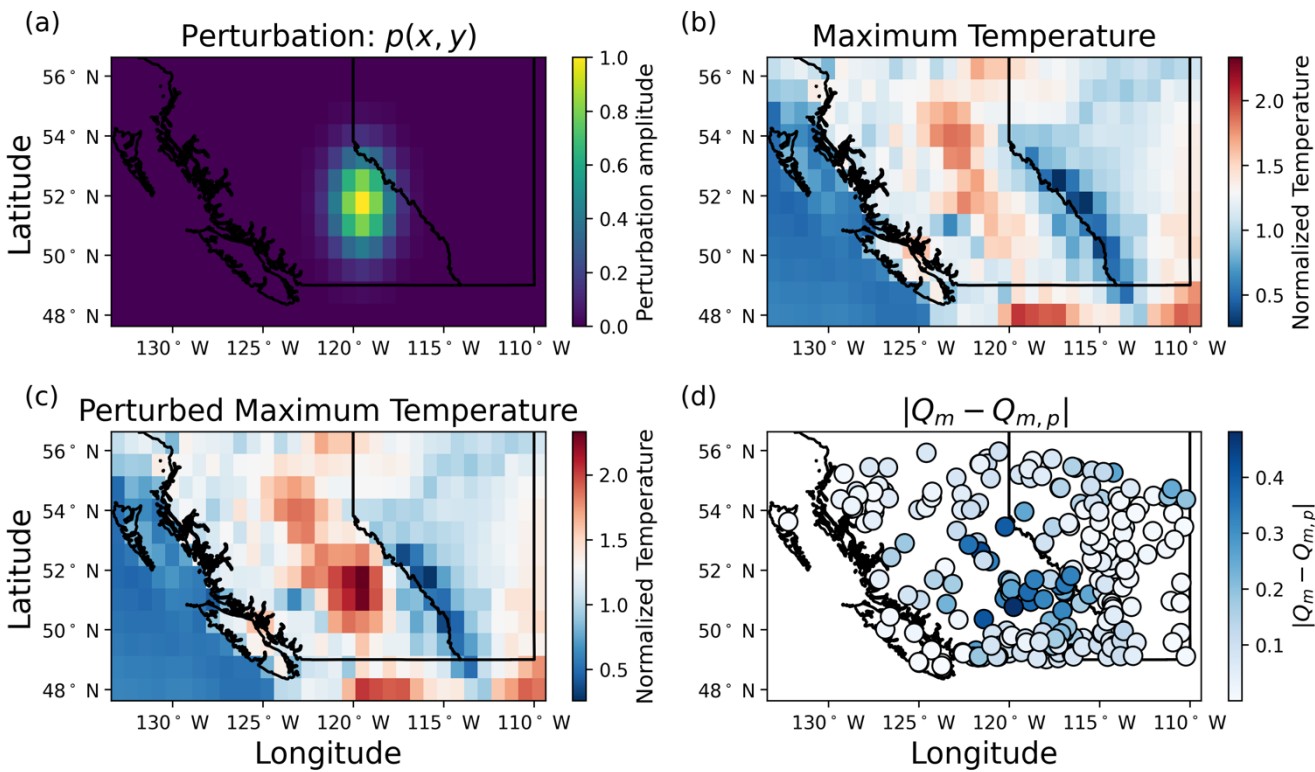

**Figure A4: Perturbing one day's maximum temperature field.** a) The perturbation to be added to the test weather data. b) The unperturbed maximum temperature field for 3 August 2011. c) The perturbed maximum temperature field for 3 August 2011. d) The magnitude of the difference between the unperturbed streamflow and the perturbed streamflow, averaged across all model runs.

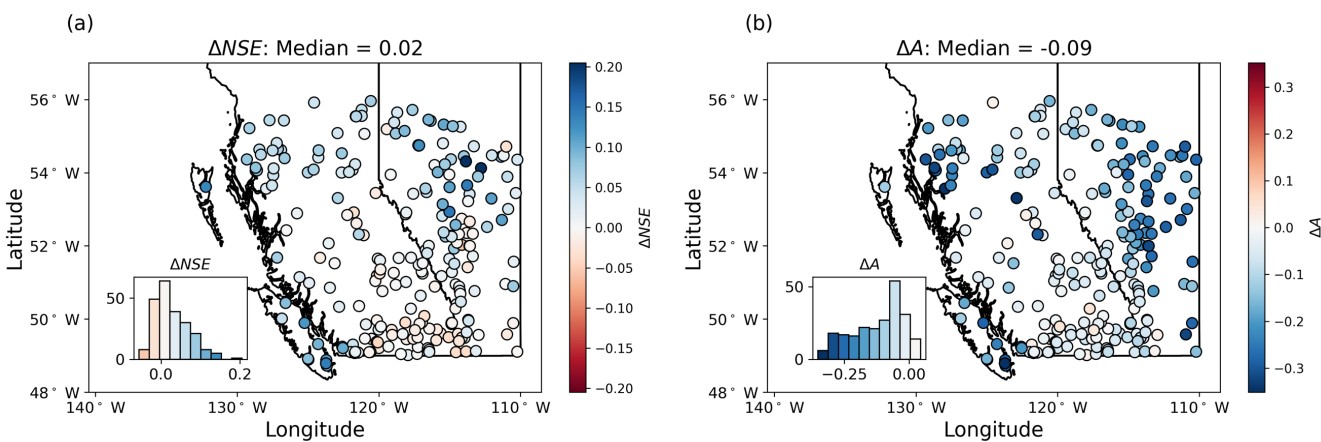

**Figure A5: The difference in model performance between the bulk model and fine-tuned model for both a) NSE, and b) sensitive area $A$.** The insets show histograms of $\Delta NSE$ and $\Delta A$ across all stations. Negative values indicate that fine-tuning reduced NSE or $A$, while positive values indicate that fine-tuning increased NSE or $A$. $\Delta NSE$ is most positive (indicating the greatest improvement through fine-tuning) along the west coast and northern regions of both British Columbia and Alberta. $\Delta A$ is most negative (indicating that fine-tuning reduces the size of the sensitive areas) along the west coast, in northern British Columbia, and throughout Alberta. One stream gauge station experienced a substantial increase in NSE from fine tuning ($\Delta NSE > 1.0$), but the histogram and colour bar are clipped for readability.

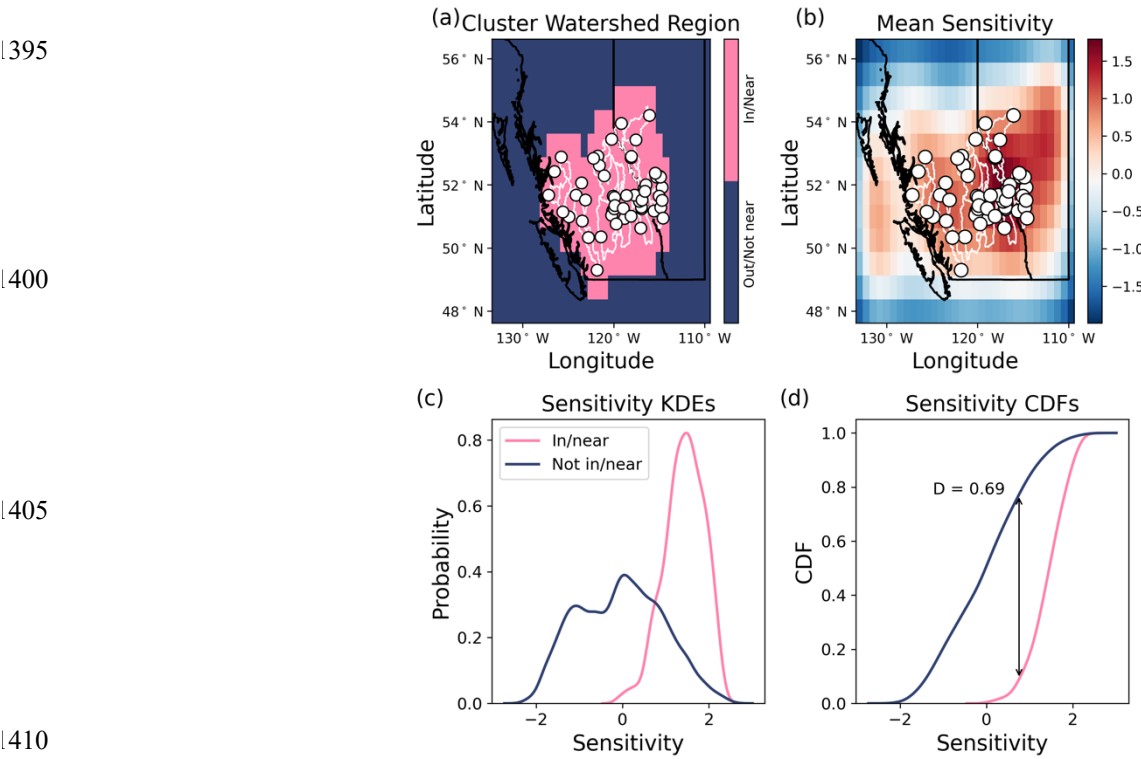

**Figure A6: Steps of calculating the D-statistic from the Kolmogorov-Smirnov test.** A) The mask of pixels which are either within/near the cluster watersheds (pink), or are outside/far from the watershed boundaries (blue). B) The mean sensitivity evaluated over the test period for the ensemble of models, with red indicating more sensitive and blue indicating less sensitive. C) The sensitivity distributions (within/near in pink and not within/near in blue), calculated by kernel density estimation (KDE) using Gaussian kernels and Scott's rule for kernel bandwidth calculation. D) The Kolmogorov-Smirnov D-statistic is calculated from the sensitivity cumulative density functions (CDFs).

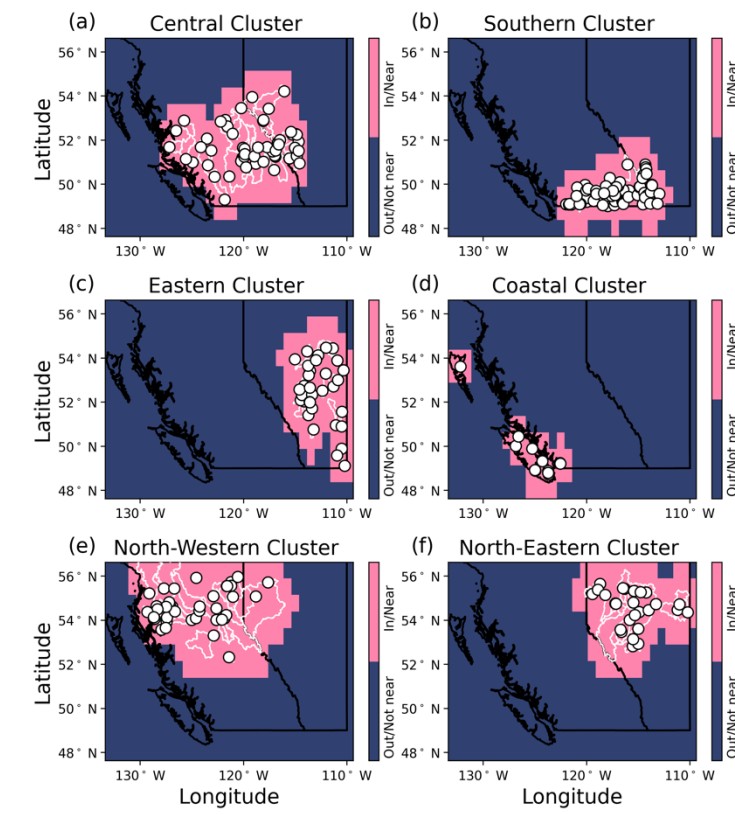

**Figure A7: The areas of the study region which are within/near watersheds (pink) of each cluster, and those which are outside/far from the watersheds (blue).** Watershed boundaries are shown in white and are accessed through the Water Survey of Canada (Environment and Climate Change Canada, 2016).