# Peer review of "Evaluation and interpretation of convolutional long short-term memory networks for regional hydrological modelling"

_Hydrology and Earth System Sciences, 2021_

## Author Comment (AC1)

**Referee Comment #1:**

**This manuscript presents an interesting application of deep learning approach for modelling streamflow responses across 226 streamflow gauges in southwestern Canada. This paper is well written and mostly easy to follow. I find the application of DL approach for streamflow simulation to be quite innovative and worthy addition to the growing body of literature in this field. The paper will be of widespread interest to the community.**

We thank the referee for their comments and are glad they find our study to be innovative and interesting.

**Overall, the paper offers plenty of interesting work, however, some effort is needed to communicate the results more effectively and highlighting key findings and novelty in view of the journal audience, e.g., better describe temperature and spatial perturbations results by linking with previous studies. Additional discussion is also needed on what the DL method brings to the table in comparison to the traditional process based models. Furthermore, while the application of DL methods for streamflow simulation is interesting, it is not entirely clear how this approach could be used for real world applications. There are also questionable choices on some of the methods and data used.**

We thank the referee for the constructive criticism and have now addressed these key comments and revised the manuscript accordingly. The specific points are responded to where they are elaborated on in "Major Comments".

**I also find the most figure captions lacking in details and could be expanded to provide more details. This will avoid readers having to scroll up and down the paper to understand the details in figures.**

We now add more detail and expand most Figure captions. In particular, the revised captions read as following:

*Line 640*:

Figure 4:
"NSE values are greatest (indicating the best model performance) throughout mainland British Columbia, and are smallest (indicating the worst model performance) in south-eastern Alberta. $A$ is smallest (indicating small sensitive areas) in the south-west and north-west coastal regions in British Columbia, and is largest (indicating large sensitive areas) throughout the rest of British Columbia and near the Alberta border."

*Line 670*:

Figure 5:

"The central and southern clusters show the least amount of change between the bulk and fine-tuned models, while all other clusters increase NSE through fine-tuning (indicating improved model performance)."

"In the central cluster, the variance of $D$ across model runs decreases through fine-tuning, indicating improved consistency between the fine-tuned central models. In all other clusters, $D$ increases, indicating improved separation between information which is near/within basins as compared to information which is further away."

"In all clusters, $A$ decreases through fine-tuning, indicating that fine-tuned models are sensitive to smaller areas of the input as compared to the bulk models."

*Line 706*:

Figure 7:
"The (a) central, (b) southern, (d) coastal, and (e) north-western clusters are generally most sensitive in the areas nearest the basins where streamflow is being predicted. The (c) eastern and (f) north-eastern clusters are most sensitive to perturbations both near the stations being predicted and further away along the west coast."

*Line 726*:

Figure 8:
"… and the cluster watershed regions are shown in Fig. A7. While all clusters are more sensitive to perturbations within/near their watershed regions, the coastal cluster demonstrates the greatest difference in sensitivity between within/near the watersheds and the rest of the domain."

*Line 794*:

Figure 9:
"In the (a) central, (b) southern, (c) eastern, (e) north-western, and (f) north-eastern clusters, $\Delta T > 0°C$ leads to an earlier freshet with a smaller peak flow, while $\Delta T < 0°C$ leads to a later freshet with a larger peak flow. In the (d) coastal cluster, $\Delta T > 0°C$ leads to enhanced streamflow in winter and fall and suppressed streamflow in summer, while $\Delta T < 0°C$ leads to suppressed streamflow in winter and fall and enhanced streamflow in summer."

*Line 1451*:

Figure A5:
"$\Delta NSE$ is most positive (indicating the greatest improvement through fine-tuning) along the west coast and northern regions of both British Columbia and Alberta. $\Delta A$ is most negative (indicating that fine-tuning reduces the size of the sensitive areas) along the west coast, in northern British Columbia, and throughout Alberta."

*Line 1523*:

Figure A7:
"Watershed boundaries are shown in white and are accessed through the Water Survey of Canada (Environment and Climate Change Canada, 2016)."

**I hope these comments are helpful and I look forward to reading the revised manuscript. My detailed comments are given below.**

We thank the referee for the numerous constructive comments which have helped to improve this study.

**Major comments**

**It is not clear what the application of DL method bring to the table in comparison to the traditional process based hydrologic models.**

We recognize that we could be clearer on this point in Sect. 1.  As such, we have revised the first paragraph as follows (italicized are new, rather than rewritten/clarified points from the initial submission):

*Line 24*: "The use of deep learning (DL) has gained traction in geophysical disciplines as an active field of exploration in efforts to maximize the use of growing in situ and remote sensing datasets (Bergen et al., 2019; Reichstein et al., 2019; Shen, 2018).  In hydrology, DL can provide alternative or complementary approaches to supplement traditional process-based modelling (Hussain et al., 2020; Marçais and de Dreuzy, 2017; Shen, 2018; Shen et al., 2018; Van et al., 2020).  *Particularly notable are DL models which have been found to outperform traditional hydrological models applied at regional scale, including those for streamflow prediction at daily temporal scale (Kratzert et al., 2018, 2019b), at hourly temporal scale (Gauch et al., 2021), and at ungauged basins (Kratzert et al., 2019a).  These recent DL-based studies have emphasized the development of lumped hydrological models; however, progress has not yet been made toward distributed DL hydrological models (Gauch and Lin, 2020).*  In contrast, traditional process-based approaches have made substantial towards distributed hydrological models (Freeze and Harlan, 1969; Marsh et al., 2020; Pomeroy et al., 2007).  Nevertheless, as input and target data are becoming available at increasingly finer spatiotemporal resolution, process-based modellers are having to address the rising computational requirements and human labour required to represent the relevant hydrological processes across larger spatial scales (Marsh et al., 2020).  *A key opportunity exists, then, to develop a DL hydrological model which can utilize spatially discretized forcing data at regional scale.*"

We also note key benefits of DL as opposed to non-deep machine learning:

*Line 53*: "While ANNs and other non-deep machine learning architectures have a long history and continue to find useful applications in hydrology, DL has more recently become a promising area of investigation due to several key characteristics (Shen, 2018): DL models can automatically extract abstract features from large, raw datasets (Bengio et al., 2013), in contrast to labour-intensive manual feature extraction often required for non-deep models; and the existence of DL model architectures which are explicitly designed to learn complex spatial and/or temporal information, in particular convolutional neural networks (LeCun et al., 1990) and long short-term memory neural networks (Hochreiter and Schmidhuber, 1997)."

The use of machine learning in hydrology, and the advent of deep learning in hydrology in the last several years, has gained substantial research interest with numerous works exploring how and why deep learning offers new and exciting ideas both in addition to and in complement to traditional hydrological models (in particular, see Shen 2018 and Shen et al. 2018). As we refer to numerous studies that:

- use deep learning which outperform regional-scale traditional hydrological models (e.g. Kratzert et al. 2018, Kratzert et al. 2019a, and in the revised submission, Gauch et al. 2021);
- use deep learning in the geosciences more broadly (e.g. Vandal et al. 2017, Ham et al. 2019, Gange et al. 2019, McGovern et al. 2019); and
- use non-deep learning in Canadian hydrometeorology (e.g. Cannon 2011, Cannon 2018, Lima et al. 2016 and 2017, Shrestha et al. 2021, Snauffer 2018),

we do not think it is necessary to further explore or elaborate in more detail why DL is an attractive complement to traditional modelling approaches as this content has been addressed in the cited literature.

We hope the referee agrees we have made it clearer in the revised submission that DL models have outperformed numerous process-based models, motivating their use and further investigation, but at the same time the most successful DL models follow a lumped hydrological modelling approach. Therefore there is an opportunity to explore DL models which can take advantage of spatially discretized forcing datasets.

**This is an important question as the application of DL methods may be limited to predicting within the range of training datasets. Additionally, while the authors outlined the development and evaluation of DL method for streamflow simulation as their objectives, it is not stated how the DL method could be used for real world applications beyond the proof of concept type approach presented in this paper.**

We now more clearly state the advantages of DL for real world applications. One of the key aspects in real world applications is whether one can trust the model predictions. The application of DL models is limited to making predictions during periods when we can trust the model's predictions. This "period of trust" is the training period only if we cannot trust the model beyond when it was trained. We can build trust in model predictions outside of this period in different ways:

**Trusting models for near-term forecasting:** In our study, we evaluate the model performance on the testing set (2011 – 2015), which follows the training/validation period (up to 2010). This provides evidence that the models have learned enough from the training period to successfully extrapolate to the near-term.

**Trusting models for long-term forecasting:** Traditional modelling approaches are used for long-term forecasting (e.g. climate change projections) under the assumption that the representation of the underlying physics will not change between the model training/validation period and the future. In other words, we may trust traditional models to work in the future (or at the very least, know the limits of our trust in these models in the future) because we understand why they work. To trust DL models for future projections, we need to understand what they are learning. Our study makes progress on this front, and in the revised submission, we emphasize more why this is important and what sorts of applications become available when we better trust and understand DL models in hydrology.

We address these points, and potential future applications and areas of study arising from the success of the CNN-LSTM approach and the use of globally available climate reanalysis data, in the following points added to the updated submission:

*Line 106*: "Fleming et al. (2021) discuss the importance of model interpretability in the context of operational hydrological forecasting where model predictions may be used for potentially high-stakes decision making. The end user may need to communicate why models make a certain prediction in order to answer clients' questions or to satisfy legal requirements. We may begin to build trust in a model's ability to forecast in the near-term by evaluating model performance on a testing dataset that is separate in time from the training and validation datasets. This approach, however, does not offer much insight into the physical relationships that the models are relying on for decision making. Additionally, without an understanding of what models have learned, it is challenging to trust a DL model for predictions in periods or places where observational datasets do not exist (e.g. for reconstructing missing historical streamflow, for predicting streamflow at ungauged basins, or for long-term forecasting of streamflow under climate change scenarios). By interpreting what a DL model has learned, we can better understand where and when a DL model can be trusted and the tasks for which it can be applied."

*Line 1004*: "Considering that ERA5 climate reanalysis has global spatial coverage and is temporally complete to 1979, there are many opportunities to investigate the transferability of this approach to different regions, the use of different predictor variables, and the use of different spatial and temporal resolution of both input and target data."

Beyond the applications in the above text (e.g. reconstructing missing historical streamflow, investigating transferability of this approach to additional regions/scales), we noted in the original manuscript (and clarified in the resubmission) that the modelling setup may be conducive for transfer learning, which is another avenue of application which could help overcome challenges arising from data limitations:

*Line 947*: "In order for the model to learn the mapping between the meteorological forcing and streamflow, a sufficiently long data record is necessary for training. The CNN-LSTM architecture presented here predicts streamflow at multiple stations simultaneously. For a model which predicts at $N$ stations simultaneously, one target observation is $N$ station-days of streamflow. For a model which predicts at a single station (e.g. an LSTM with a single output neuron), one target observation is a single station-day of streamflow. For a given training dataset with $M$ station-days of streamflow observations, the CNN-LSTM with $N$ output neurons would have $M/N$ observations for training, while the model with a single output neuron would have $M$ observations for training. That the number of observations for training has been reduced is potentially detrimental to the model's performance. A potential solution to this problem could be to use transfer learning with a CNN-LSTM model pre-trained in a region with a sufficiently long streamflow record and then transferred to the new region of interest."

**The clustering method divided the study region into six clusters based on seasonal streamflow, latitude and longitude variables in order to fine-tune the model training. However, there are a number of studies in the region which describe the spatial heterogeneity of the region. For instance, streamflow responses in the lee- and windward side of coast and rocky mountains, as well as mountainous and interior plains are know to be quite different (e.g., Moore 1991; Shrestha et al. 2012). Therefore, I would think including variables like slope and aspect will be able to better characterize the spatial heterogeneity and provide clusters that better capture the variability in the streamflow response. Better clustering can potentially improve the model fine-tuning and model performance in several regions, especially in the Eastern slopes of the Rocky Mountains where the model performed relatively poorly.**

This is a valid point raised by the referee and is one we have considered during the study design.

The clustering is not used to just find stations which are most hydrologically similar; if this were the sole goal of clustering, then we agree with the referee and should use a greater number of relevant variables such as slope and aspect (and drainage area, elevation, glacier coverage, etc.) which better capture the heterogeneity of the region. A key product of clustering is to find subsets of stations which can be predicted by an *interpretable* model. In our case, it is desirable to identify clusters which are in large part determined by geographic location because one of our main goals is to determine where in space the model is learning to focus when predicting streamflow at each cluster.

If multiple clusters overlap in space, it is less easily detectable if the model is learning to focus on different physically relevant areas, or if it is learning to map similarly large areas of the input through to the output. To illustrate this point, we here cluster only the seasonal streamflow (i.e. without the geographical information). In this case, clusters 2, 4, and 5 span similar large regions of the input space, and also overlap considerably with cluster 0. For clusters 2, 4, and 5 in particular, we would expect the model to be sensitive to perturbation throughout most of British Columbia, and so the sensitivity heat maps would be very similar. We would then not be able to tell if the model is really learning different things for clusters 2, 4, and 5. By emphasizing geographic location in clustering, as we do in the paper, we can set ourselves up to train models

that can be better interpreted because of their sensitivity to different geographic areas in the domain.

[Figure]

Another example to justify our point: consider two stations that are nearby one another, but have different characteristics such as slope, aspect, and percent glaciation in the watershed. In order to predict streamflow at each station, it should still be most important that the model focuses on areas near and within the two watersheds, respectively. For each station, the mapping through to streamflow from this 'most relevant information', then, may be different, but the sensitive areas should be similar. So while clustering in the space of hydrologic variables other than geographic location may lead to small improvements in performance as measured by NSE, it makes it more difficult to understand what the model is learning and why it is making decisions.

To add additional context about the region's hydrology, we revised the following text:

*Line 230*: "Streamflow throughout the study region varies strongly in space and time and reflects the varied topographic and climatic conditions in British Columbia and Alberta. Here we provide a brief, high-level overview of streamflow characteristics, and while it is not a complete summary of the full range of hydrologic conditions throughout the study region, we aim to highlight that streamflow through the region is heterogeneous in space and time. Streamflow at low-elevation coastal stations is primarily driven by rainfall, with monthly discharge maximized in November or December. In contrast, streamflow at stations that are at higher elevation, further north, or further inland transition to a snowmelt-dominated regime, with monthly discharge maximised in spring or early summer. Numerous glaciers exist in high elevation alpine areas throughout both the Coast Mountains along the west coast of British Columbia and the Rocky Mountains along the border between British Columbia and Alberta, and glacier runoff contributes to streamflow through late summer once the seasonal snowpack has melted (Eaton and Moore, 2010). East of the Rocky Mountains, the Prairie region in eastern Alberta is uniquely characterized by relatively flat topography with small surface depressions (LaBaugh et al., 1998). Water can pond and be

stored in these depressions, leading to intermittent connectivity throughout many basins and drainage areas which may vary in time (e.g. Shook and Pomeroy, 2011)."

Revisions that discuss prior work:

*Line 251*: "Previous studies have used a range of techniques to cluster or summarize the diversity of spatiotemporal streamflow characteristics in the study region (e.g. Halverson and Fleming (2015) use complex networks to represent similarity between streamflow timeseries in the Coastal Mountains, while Anderson and Radić (2020) use principal component analysis and Self-Organizing Maps to characterize summer streamflow through Alberta). In this study we use a relatively simple clustering approach, only considering seasonal streamflow, station latitude, and station longitude."

To comment on why we use this simpler clustering approach:

*Line 274*: "Our clustering approach does not explicitly consider input features such as land use, glacier coverage, drainage area, or elevation, but rather implicitly considers the expressions of these features in the seasonal hydrograph. The goal of this type of clustering is to define subsets of stream gauge stations that are nearby in space and share similar hydrographs. We prioritize proximity in space over an explicit representation of other important features (e.g. drainage area, elevation, glacier coverage) because a key goal of the study is to interpret where in space the DL models have learned to focus when predicting streamflow. As discussed in Sect. 4.3.1 and Sect. 4.5.1, having clusters of stream gauge stations which are nearby in space allows us to visualize if the trained models are learning to focus on the subregion of the input domain which overlaps with the watersheds where streamflow is being predicted."

**It is surprising to see that the study used 0.75° x 0.75° resolution ERA5 reanalysis data, especially given that the authors stated finer resolution climate data may improve model performance (L637). I wonder why the authors did not used to the finer resolution data readily available for the region (e.g., Werner et al. 2019)?**

We recognize that there are multiple high quality datasets available for our study region. One key reason to use ERA5 data specifically, and not other regional datasets which are available for this region, is because of ERA5's global spatial coverage. By training on ERA5 data, models can more easily be adapted for applications such as hindcasting or for transfer learning across regions. We did not pursue these applications here, but we comment on their potential for future work.

There are also practical computational reasons to use coarser resolution climate data. There is a balance between "how much information there is to learn from" (related to the size/dimension of individual observations for training, i.e. the amount of information in a single weather video) and "how much learning can be done during training" (related to the number of observations available for training); generally speaking, the more information there is to learn from, the more learning that needs to be done. When predicting streamflow at multiple stations simultaneously,

the number of observations for training is reduced, and so it is beneficial to also reduce the size/dimension of individual training samples. The simplest way to do this is by using coarser resolution climate data, which we find to be sufficient for meeting our goals.

It is common in hydrological modelling to first perform downscaling of the input data to the model resolution. In other words, there is a mapping from coarser resolution climate data, to finer resolution climate data, to streamflow (e.g. there is a transfer function from coarse resolution climate data to fine resolution climate data, and then another transfer function from fine resolution data to streamflow). Here, the intention is that the effect of downscaling is learned in the mapping from coarse resolution climate data to streamflow directly (e.g. a single transfer function from coarse resolution climate data to streamflow). That CNNs have been used to map coarse-resolution climate data to high-resolution climate data suggests that key information of high-resolution climate data is present within coarse resolution data (e.g. Vandal et al., 2017), and so it could be possible that an 'implicit downscaling' is learned during the mapping from coarse climate data to streamflow. We now include this point in the text:

*Line 940*: "Our model uses forcing data at relatively coarse spatial resolution (0.75° x 0.75°, or ~75 km resolution)  as compared to studies identified in Table 2 (e.g. 0.0625° x 0.0.0625° in Shrestha et al. (2012); 10 km resolution in Eum et al. (2017)). Studies that employ a climate downscaling step first map coarse resolution climate data to fine resolution climate data, and then map the downscaled fine resolution climate data to streamflow. Here, the CNN-LSTM is effectively representing a single transfer function that maps coarse resolution climate data directly to streamflow, and it is possible that an effective downscaling of climate data is learned by the model. This indirect downscaling is plausible since statistical methods are often used for climate downscaling, including CNNs (Vandal et al., 2017)."

**The authors described the DL methods as if the study is on image/video processing. While the methods may be same as image/video processing, there is a need to rephrase section 4 in terms of hydro-climatic modelling.**

The benefit of using terminology from video/image processing (e.g. 'video', 'frame', 'channel', 'pixel') is that it offers a succinct way to describe the input data and its structure while maintaining accuracy (e.g. it is simpler to refer to a 'frame' rather than 'one day of three weather predictors'). However, we agree that the connection between the image processing and hydro-climatic terminology should be improved. To improve clarity, we add the following description in the first paragraph of Sect. 4.3:

*Line 381*: "To ensure consistency between terminology in both image processing and hydro-climatic modelling, a 'weather video' refers to 365 days of the three weather predictors, a 'frame' or 'image' in a weather video refers to one day of the three weather predictors, a 'channel' in a 'frame' or 'image' refers to one day of one weather predictor, and a 'pixel' refers to one grid cell."

**Specific comments**

**L95-110: The objective and novelty need to be revised by clearly describing what the DL method bring to the table compared to the process-based models, and how the DL method could be used for real-world application.**

See response to the first two major comments.

**L135: What are the range of basin areas for the selected stations?**

The basin drainage areas span approximately 5 orders of magnitude (minimum area: ~6 km^2, maximum area: 133,000 km^2). Stream gauge elevation and drainage area are now visualized in Figure A3 and referred to in Sect. 3.2 (Streamflow clusters):

*Line 265*: "The elevation and drainage area of stations for each cluster is shown in Fig. A3."

[Figure]

**Figure A3: Elevation and drainage area of stations within each of the identified clusters.** Station elevation is calculated from a digital elevation model from the Shuttle Radar Topography Mission (SRTM) at 90 m resolution (Farr et al., 2007). Drainage area is taken from the Environment Canada HYDAT database (Environment and Climate Change Canada, 2018). The coastal cluster is at the lowest elevation and with the smallest drainage areas. Clusters in mainland British Columbia (central, southern, and north-western) span wide ranges of elevation and drainage area, while clusters in Alberta (eastern and north-eastern) span narrower ranges of elevation and drainage area.

**L138: Naturalized flow generally means regulated flow adjusted with regulation/abstraction removed. Correct term is natural flow.**

We have fixed this typo and note for clarity:

*Line 206*: "HYDAT classifies stream gauge stations as either "regulated" (downstream of regulating structures such as a dam) or "natural" (upstream of regulating features). We use stations that are classified as natural and that are currently active."

**L140-145: 40% missing data can lead to challenges in model setup. Wondering if the model performance was inferior for basins with missing data than basins with complete data sets?**

One challenge is that we need temporally complete datasets since the number of output neurons is constant through training. We recognize that the threshold of 40% missing data may lead to challenges, especially if the data is missing during periods of dynamic streamflow (in other words, the model would be missing how to learn when streamflow should substantially change due to meteorological forcing); however, this is not the case. A vast majority of data is missing between November and February, when temperatures are coldest and streamflow is more inhibited as compared to spring and summer. These data are typically missing at stations which record data seasonally, rather than continuously, and the 40% threshold allows us to include seasonal stations which do not record in winter months. The following figure demonstrates how the most missing data occurs in the low-flow period.

It is true that the two worst performing clusters (eastern and north-eastern clusters; Figure 5) are also those with the most missing data (below). However, the missing data occurs during low-flow periods and as such is not likely to be driving the poor performance, since NSE is largely determined by predictions in spring and summer.

[Figure]

**L158-170: As stated earlier including slope and aspect may improve the cluster selection and model performance.**

See above response in 'Major Comments'.

**Figure 2: State in figure caption how the discharge values are normalized. Similarly, the authors need to provide more details in all Figure captions.**

In Figure 2, the following was added to the caption:

*Line 297*: "Seasonal discharge of each station is normalized to have a mean of zero and unity variance".

Figure captions were expanded as addressed prior to the "Major Comments" above.

**L189: It is not clear how the gridded weather data is mapped to the streamflow stations, are the nearest grid cells or mean values from several grid cells taken?**

All values of the gridded weather data ("weather video") are mapped to all stream gauge stations (see: Figure 3). The specific mapping from weather-to-streamflow for each stream gauge is learned through training. We clarify this point in Sect. 4.3 when the weather videos are first described in detail (new text in italics):

*Line 390*: "One year-long weather video is used as an input to predict the next day of streamflow at the 226 stream gauge stations; *in other words, all grid cells of temperature and precipitation are mapped to streamflow at all stream gauge stations.*"

**L315-316: Since previous 365 days of data are required, is Jan. 1 1980 is the first day used for streamflow training?**

The referee is correct that Jan 1 1980 is the first day of streamflow used. We add:

*Line 414*: "Since 365 days of previous temperature and precipitation are used to predict streamflow, and since the ERA5 data begins on December 1, 1979, the first day of streamflow used is January 1, 1980."

To clarify the dates for predictors/predictands and to further justify our decisions when creating the training/validation/testing sets:

*Line 416*: "In other words, the training period is defined by daily streamflow from January 1, 1980 to December 31, 2000, with forcing data ranging from January 1, 1979 to December 30, 2000. The validation period uses streamflow data from January 1, 2001 to December 31, 2010, with forcing data ranging from January 1, 2000 to December 30, 2010. The testing period uses streamflow data from January 1, 2010 to December 31, 2015, with forcing data ranging from January 1, 2009 to December 30, 2015. We choose to separate the training/validation/testing datasets into non-overlapping time periods of streamflow so that model performance can be evaluated on out-of-sample streamflow examples. We choose to use a full decade for validation

because we want to encourage the model to perform well across a range of conditions and not for one particular year or climate state, since oscillations in the climate system such as the El-Nino Southern Oscillation, the Pacific Decadal Oscillation, and the Pacific-North American atmospheric teleconnection influence streamflow through modifications to temperature, precipitation, and snow accumulation through the study region (e.g. Fleming and Whitfield, 2010; Hsieh et al., 2003; Hsieh and Tang, 2001; Whitfield et al., 2010). We also choose to use multiple years for testing so as to not bias our conclusions towards the conditions of a single year. Furthermore, we partition the training, validation, and testing data by year rather than by percentage of observations (i.e. the testing subset is chosen as 5 years, not 10% of observations) so that we do not bias our results by including or excluding parts of the year when the model performs better or worse than average. Overall, the training-validation-testing data split is approximately 59% - 27% - 14% of the total streamflow dataset. The input data are normalized so that each variable (maximum temperature, minimum temperature, precipitation) has a mean of zero and unity variance over the training period. The target data from each of the 226 stations are normalized so that each station's streamflow has a mean of zero and unity variance over the training period."

**L364: The spatial perturbation section is hard to follow, how was the amplitude of 1 used in perturbation of climate fields?**

To improve the clarity of this section and to show how the amplitude of 1 is used in the perturbation of the climate fields, we added equations to define perturbed daily temperature and precipitation fields ($T_{max,p}$, $T_{min,p}$, and $P_p$) as:

*Line 512*:

$$p(x,y) = \beta * e^{\frac{-1}{2}\left[\frac{(x-x_p)^2}{\sigma_x^2} + \frac{(y-y_p)^2}{\sigma_x^2}\right]}$$
$$T_{max,p}(x,y,t) = T_{max}(x,y,t) + p(x,y)$$
$$T_{min,p}(x,y,t) = T_{min}(x,y,t) + p(x,y)$$
$$P_p(x,y,t) = P(x,y,t) + p(x,y)$$

We also write that:

*Line 516*: "…$T_{max}$, $T_{min}$, and $P$ are the unperturbed normalized daily maximum temperature, minimum temperature, and precipitation, respectively."

*Line 517*: "The amplitude of the perturbation was chosen to be 1 since the climate variables are normalized to have unity variance across the training period. This way each climate variable is perturbed by a maximum of a single standard deviation."

**L411: Also say temperature perturbations are constant throughout the time period.**

We have edited the text:

*Line 567*: "To test the hypothesis, we add a spatially *and temporally* uniform temperature perturbation…".

**L425: on what basis/reference was the criteria for freshet timing defined?**

One challenge in choosing a definition of "freshet timing/onset" is the number of definitions which are used in the literature, such as:

*Zhang et al., 2001*: The date when the increase in daily streamflow across 4 days is greater than the average from January to July

*Woo and Thorne, 2003*: The first day flow is more than double than the flow of the prior day

*Burn et al., 2004*: The first day flow exceeds 1.5 times the average of the previous 16 days

*Vincent et al., 2015*: The date when the cumulative sum of the difference between daily mean streamflow and its climatology reaches a minimum in the hydrological year

We found our definition to be robust across the snowmelt-dominated cluster (e.g. cluster-ensemble-mean streamflow fluctuations prior to the spring rising limb were not large enough to be mis-identified as the freshet onset) and, in our view, easier to visually and conceptually understand as meaning "When has spring snowmelt substantially increased flow?". We now include two additional references (Woo and Thorne, 2003; Burn et al., 2004) when we introduce our definition to emphasize the diversity of freshet definitions used previously in literature (Line 580).

**The results in Figure 4b have not been adequately described in the text.**

We have added the following description in the text:

*Line 738*: "The central, coastal, and north-western stations have smaller sensitive areas, while the southern, central, and north-eastern stations have larger sensitive areas (Fig. 4b and Fig. 5c). … Notably, the clusters which are sensitive to the smallest areas of the input (central, coastal, and north-western, Fig. 4b) all experience a substantial decrease in A through fine-tuning (Fig. A4b and Fig. 5c). This indicates that fine-tuning may be necessary for the model to focus on small areas of the input space."

**Figure 6: name the basins for which these example results are presented.**

The names and station IDs are now included in the title of Figure 6. Additionally, for all stations used we include station names, numbers, latitude, longitude, and if they are part of the Reference Hydrometric Basin Network in Table S1.

**L492: Given that streamflow at a hydrometric station is response to precipitation and temperature over the entire drainage basin, it is to be expected that there are higher sensitivity in response by including areas near and within the station. This need to be clarified, in the context of how big the drainage basins are, and whether the inclusion of precipitation and temperature variables from a wider region improved the model performance.**

The referee is correct that a greater sensitivity in response to perturbations is expected near/within the basin; however, this should only be true if the model has learned that the grid cells near/within the basin are more important to streamflow at that station as compared to grid cells which are further away.  Since all grid cells are mapped to all stream gauge stations (clarified in response to a point above), the model needs to learn which grid cells are most relevant.  The model automatically learns that grid cells near/within basins are more important (i.e. higher sensitivity), which indicates the model's ability to learn physically relevant and interpretable information.

**L562: How is the intensity of freshet calculated?**

The intensity of the freshet was calculated as the peak spring flow, as defined after Equation 16 in the original submission.  However, upon reflection, we have edited the text and now refer to this quantity as the "freshet peak flow", still defined after Equation 16, rather than "intensity".

**L562-580: The results in Figures 9 and 10 seem to be consistent with previous climate change impact studies in the region. This is quite promising and is perhaps one of results the authors can highlight further. I suggest expanding the discussion in this section by linking with previous climate impacts studies.**

We have added the following paragraph:

*Line 907*: "When the input temperature series is made warmer (cooler), the freshet onset timing and peak flow advances (delays) and decreases (increases) (Fig. 9 and Fig. 10).  This finding is consistent with previous studies of climate change impacts in the region.  For example, Shrestha et al. (2012) used the macro-scale Variable Infiltration Capacity (VIC) hydrological model forced by a suite of global climate models in the Fraser River Basin (which spans the central cluster in our study), finding that spring peak flows occur earlier in the year and with lower magnitude under a warmer future climate (Shrestha et al., 2012).  Schnorbus et al. (2014) used the VIC model to project streamflow in the Peace, Campbell, and Columbia River watershed in British Columbia (primarily in the north-western, coastal, central, and southern clusters in our study) under a range of climate change scenarios (Schnorbus et al., 2014).  They found greater spring flows and lower summer flows in the snowmelt dominated locations, while the coastal location was projected to experience enhanced winter flows and depressed summer flows.  It is promising that not only does the CNN-LSTM model perform well in the historical period (e.g. the test period of 2011 - 2015), but produces conceptually similar projections for a warmer climate as compared to existing physically-based models."

**L589: Rephrase the sentence, it appears as if previous studies also used deep learning.**

The referee's interpretation of that sentence is correct. The studies discussed in that paragraph also used deep learning, but in different regions and with different model architectures.

**L626-627: While it is true that the non-contributing areas may have played a part in DL results in parts of eastern cluster, the cited studies are outside of the study region and not directly comparable. Also non-contributing areas may not be a factor for the entire region. There are maps available which outline the extent of non-contributing areas.**

We have adjusted our language in this section to emphasize that the effect of non-contributing areas is not the only possible explanation, but is at least a possible explanation to some degree within the eastern cluster.

It is a good point raised by the referee that non-contributing areas may not be a factor for all stations. We further investigate this by calculating the fraction of non-contributing areas for all stations in the eastern cluster using data from the 'Areas of Non-Contributing Drainage within Total Gross Drainage Areas of the AAFC Watersheds Project – 2013' (found at this link, with citation below: https://open.canada.ca/data/en/dataset/adb2e613-f193-42e2-987e-2cc9d90d2b7a). We find that across the eastern cluster:

- Minimum fraction of non-contributing areas across all stations: 0 (only 4 of 34 basins have no non-contributing areas)
- Minimum fraction of non-contributing areas across the 30 basins with non-zero non-contributing area: 0.01
- Mean fraction of non-contributing areas: 0.20
- Maximum fraction of non-contributing areas: 0.79

We add:

*Line 882*: "In the eastern cluster, 30 out of 34 basins have non-contributing areas, ranging from 1% to 79% of the total basin area with a mean of 20% of the total basin area not contributing to streamflow on average (Government of Canada, 2020)."

Furthermore, we provide code which reproduces this step of the analysis on Github (https://github.com/andersonsam/cnn_lstm_era/blob/master/non_contributing_areas.ipynb).

**Table 2 heading: State the period of test set used. Also it should be clarified in the heading that various validation periods were used in reference models.**

We have added the period of our test set (2011 – 2015) and that various validation periods were used in the reference models (Line 1010).

**Discussion and Conclusions: the changes suggested above also applies to the results and discussion section.**

The changes made to the discussion and conclusions sections have been listed above.

References:

[revised manuscript text omitted]

---

## Author Comment (AC2)

**General comments:**

**This is an intriguing study that combines two distinct deep-learning technologies (the convolutional neural network, CNN, and long short-term memory neural network, LSTM) to create a new method for regional daily streamflow prediction that integrates complex spatiotemporal structures and dependencies. The method is applied to streamflow data from the southern portion of Canada's two westernmost provinces, which is a geophysically complex and interesting region. Some effort is also made to address physical interpretation and meaningfulness of the technique. It is a promising study with widely relevant results that has strong potential for publication in a top-tier hydrology journal like HESS.**

We thank the referee for their comment and are glad they agree the results are widely relevant and the study is promising.

**That said, the submission as it currently stands appears to have some substantial issues that need to be addressed before it can be considered for publication. The overall feel of how the manuscript is written is one of technical naivete and oversimplification, undermining the credibility of the study. For example, the text of the paper and possibly some of the analytical steps suggest a superficial understanding of the physical hydrology of western Canadian rivers and their associated datasets; and overall, the literature review around machine learning and its hydrologic applications is wholly inadequate and does not provide the reader with accurate and meaningful context to the study. Additionally, several basic elements one normally expects of a machine learning paper today seem to be missing, like clear descriptions of training vs. testing vs. validation data subsets, or the use of informative benchmark models to evaluate the new model against.**

We have addressed these concerns in the revised manuscript. Our responses to each individual comment are given further below.

**The study is also not reproducible based on the limited information provided in the paper.**

We have now improved the information on the data used and added more details in the methods. We also added Table S1 in the Supplementary Information which contains station names, numbers, latitude, longitude, and RHBN status for all stations used in this study. For easier use, we also include these data as 'station_table.csv' on Github.

We note that the first submission included all code used on Github, including detailed steps how to download, access, and structure all data required. The file 'main_publish.ipynb' goes step by step to reproduce the figures used in the paper. In addition, we have now added a notebook 'mini.ipynb' which does not require readers to download any data themselves. Instead, we provide enough preprocessed climate/flow data to create 1 year of input/target and all trained bulk/fine models. From there, users can cluster the stream gauge stations, generate model predictions, evaluate model performance, make sensitivity heat maps, and perturb temperature

and measure the models' responses; essentially, all results from the paper, but with a single year of data instead of the >3 decades of data used in the full 'main_publish.ipynb'.

**My recommendation is to accept the paper for publication in HESS pending major revisions. I hope the detailed comments provided below, as well as the references section that follows those detailed comments, will be helpful to the authors as they revise their manuscript.**

We appreciate the detailed comments to help improve the manuscript. We have gone through and responded to the individual comments below.

**Detailed comments:**

**\* Line 30: Should also cite Hsu et al. (1995) here, as to my knowledge it was the first peer-reviewed journal paper to present the use of machine learning for rainfall-runoff modeling. (Full literature citations are provided below.)**

We have now included this citation.

**\* Lines 34-37: This feels like an overstatement/misstatement of both the limitations of conventional machine learning and the advantages of deep learning in a hydrologic prediction context. For one thing, a basic result in AI, dating back to the late 1980s or so, is that non-deep ANNs (in particular, multilayer perceptrons having a single hidden layer) are theoretically capable of learning any continuous relationship. Another issue: contrary to what is implied in the passage, non-deep ANNs are not the only kind of non-deep machine learning – there are several other major classes (random forests, support vector machines, and so forth). There also continues to be intense research in non-deep ML to create new kinds of AI, including news kinds of neural networks, having certain useful characteristics that have been successfully applied to river prediction; online sequential learning is an obvious example (e.g., Lima et al., 2015, 2016, 2017). Indeed, new kinds of non-deep machine learning algorithms are being developed specifically for hydrometeorological analysis and prediction tasks (e.g., Cannon, 2010, 2011, 2018; Fleming et al., 2015, 2019, 2021). On the other hand, deep learning applications in hydrology are currently in vogue and seem to be very promising in certain circumstances, but the body of work on the subject – particularly around streamflow prediction – remains exceedingly small, and the ultimate suitability of deep learning to this task, including capabilities and limitations, remains unclear at this point. A more mature way of looking at deep learning in hydrologic prediction is that work to date suggests it is a promising research direction that could potentially offer an alternative or complementary approach to non-deep machine learning for certain tasks.**

We have edited this section (and much of the introduction). New text is in **blue** if paragraphs have text from both the first submission and the edited manuscript, while new text is in black if the responses are entirely new text:

*Line 38*: "Early applications of machine learning in hydrology date back to the 1990s, with artificial neural network (ANN) models used for rainfall-runoff modelling (e.g. **Hsu et al., 1995; Maier and Dandy, 1996; Zealand et al., 1999) and a range of other hydrometeorological analysis such as flood forecasting (Fleming et al., 2015), improving gridded snow-water equivalent data products (Snauffer et al., 2018), and predicting total April-August streamflow (Hsieh et al., 2003).**"

*Line 45*: "In addition to ANNs, which have received particular attention in hydrology (Maier et al., 2010; Maier and Dandy, 2000), numerous types of non-deep machine learning applications have also been developed for hydrometeorological analyses, and in particular, many have been developed for applications in Western Canada.  For example: Bayesian neural networks, support vector regression, and Gaussian processes have been used for streamflow prediction at a single basin (Rasouli et al., 2012); quantile regression neural networks have been used for precipitation downscaling in British Columbia (Cannon, 2011) and estimation of rainfall intensity-duration-frequency curves across Canada (Cannon, 2018); online sequential extreme learning machines have been used for streamflow prediction in two basins (Lima et al., 2016, 2017); and random forest models have been used to identify temperature controls on maximum snow-water equivalence in Western North America (Shrestha et al., 2021).  While ANNs and other non-deep machine learning architectures have a long history and continue to find useful applications in hydrology, DL has more recently become a promising area of investigation due to several key characteristics (Shen, 2018): DL models can automatically extract abstract features from large, raw datasets (Bengio et al., 2013), in contrast to labour-intensive manual feature extraction often required for non-deep models; and the existence of DL model architectures which are explicitly designed to learn complex spatial and/or temporal information, in particular convolutional neural networks (LeCun et al., 1990) and long short-term memory neural networks (Hochreiter and Schmidhuber, 1997)."

*Line 156*: "Deep learning in hydrology has shown promise for streamflow prediction tasks, but knowledge gaps exist surrounding the development of architectures which explicitly incorporate both space and time, the interpretation of model learning, and the limitations of such modelling approaches."

**\* Lines 49-50: The use of point observations (of weather, presumably) does not necessarily imply that a model is spatially lumped.  It is very common in process-based hydrologic modeling, including semi-distributed and fully distributed models, to spatially interpolate measurements from point data sources.  In fact, some process-based models even integrate that spatial interpolation step into the software platform, along with adjustments for adiabatic lapse rates, etc., etc.**

The term "point observations" has been removed; while we meant that the LSTM approach had been used as a lumped hydrological model with point-observations as input, we agree with the referee that this was not necessarily clear.

**\* Lines 67-70: Explainability is an issue for all machine learning models, not just deep learning models; it feels like this passage is conflating ML generally with DL specifically.  For a recent example of a new non-deep ML technique specifically introduced to improve interpretability of a practical hydrologic prediction model, see Fleming et al. (2021), which also provides a much better explanation of exactly why geophysical explainability is a key requirement for practical applications of machine learning in hydrologic prediction.**

We add the following:

*Line 106*: "Fleming et al. (2021) discuss the importance of model interpretability in the context of operational hydrological forecasting where model predictions may be used for potentially high-stakes decision making.  The end user may need to communicate why models make a certain prediction in order to answer clients' questions or to satisfy legal requirements.  We may begin to build trust in a model's ability to forecast in the near-term by evaluating model performance on a testing dataset that is separate in time from the training and validation datasets.  This approach, however, does not offer much insight into the physical relationships that the models are relying on for decision making.  Additionally, without an understanding of what models have learned, it is challenging to trust a DL model for predictions in periods or places where observational datasets do not exist (e.g. for reconstructing missing historical streamflow, for predicting streamflow at ungauged basins, or for long-term forecasting of streamflow under climate change scenarios).  By interpreting what a DL model has learned, we can better understand where and when a DL model can be trusted and the tasks for which it can be applied."

**\* Lines 78-79: the authors are not using the terms white-box and (in particular) black-box in the way they are usually used.  Most working in hydrology, in particular, would regard any physically explainable ML as being white-box in some sense.  The term "black-box" is normally reserved for machine learning algorithms that do not offer any physical interpretability, which is to say, most of them.**

These lines have been rephrased:

*Line 129*: "In contrast to the above approaches which interpret the model through explicit use of the model parameters, alternative methods exist which do not use internal network states for interpretation."

**\* Lines 85-86: would be useful to note the similarities and differences between recurrent and LSTM neural networks here for a general readership.  The text seems to be haphazardly switching between the two, which are related but not identical; LSTM is essentially a specific and advanced form of recurrent ANN.  This applies to the title of the paper too; why "recurrent" instead of "long short-term memory"?**

The original intent behind using "convolutional-recurrent" phrasing rather than "convolutional long short-term memory" phrasing was to keep the wording more succinct, which came at the cost of precision (since LSTM is a type of recurrent network).  Upon reflection we have decided

to change this to "convolutional long short-term memory" to be more precise, both in the title and throughout the text.  We also include:

*Line 60*: "Long short-term memory (LSTM) neural networks are designed to learn sequential relationships on a range of scales (Hochreiter and Schmidhuber, 1997).  LSTMs are a type of recurrent neural network (RNN).  Traditional RNNs include a feedback loop between the network output and input in order to learn temporal dependency within the data (Rumelhart et al., 1985); however, they struggle to learn long-term dependencies greater than around 10 time steps (Bengio et al., 1994).  LSTMs overcome this limitation through the inclusion of an internal memory state or cell state which can store information, and learning is achieved by including internal gates through which information can flow and interact with the cell state."

**\* Lines 104-105 are a bit off as well.  There seems to be an implication here that more complex models are better models, and that in contrast this study is aiming for parsimonious models. That's an odd way of looking at the desirability of different modeling approaches and structures.  Most modelers view a parsimonious model as being fundamentally better, holding all else equal, i.e., so-called Accom's razor.**

We did not mean to imply or convey that more complex models are necessarily better models; rather, we were considering that it may be possible to achieve better model performance in this instance through increased complexity (e.g. more layers / convolutional filters / LSTM units / etc). Here we simply meant that our goal is not to necessarily achieve better performance by optimizing all hyperparameters and architecture (e.g. we use our few-layer model which works well rather than aiming for a deeper model with more parameters which may work a bit better). This is now a point that we make later, and as such, we remove the sentence that was originally at lines 104-105.  The point we now make later is:

*Line 459*: "It is possible that a better performing architecture or training scheme could be constructed by optimizing hyperparameters with an out-of-sample subset; however, we show our model setup and design is sufficient for achieving the goals of this study."

**\* In addition to the various other papers referenced in this review that should be cited in the paper but were not, the authors may also wish to read and cite the review articles by Reichstein et al. (2019) and McGovern et al. (2019).  Citing prior applications of machine learning to hydrologic and related modeling in the study area would also be appropriate.  Some examples that come to mind include Rasouli et al. (2012), Lima et al. (2015, 2016, 2017), Snauffer et al. (2018), Fleming et al. (2015), Hsieh et al. (2003), and Shrestha et al (2021).**

We thank the referee for the suggested references. We now note (text in **blue** indicates new text in an old sentence; all other points are entirely new text):

*Line 24*: "The use of deep learning (DL) has gained traction in geophysical disciplines as an active field of exploration in efforts to maximize the use of growing in situ and remote sensing datasets (Bergen et al., 2019; **Reichstein et al., 2019**; Shen, 2018)."

*Line 98*: "Notably, the CNN-LSTM architecture has been identified as being an architecture of potential or emergent interest for geoscientific applications involving spatiotemporal phenomena (Reichstein et al., 2019)."

*Line 89*: "In the geosciences, CNNs have gained popularity more recently with applications including long-term El-Nino forecasting (Ham et al., 2019), precipitation downscaling (Vandal et al., 2017), **hail prediction (Gagne et al., 2019),** and urban water flow forecasting (Assem et al., 2017)."

*Line 122*: "Here we introduce select concepts and methods which can be used to interpret DL models; further details for machine learning and deep learning interpretation in a geoscientific context can be found in McGovern et al. (2019)."

As previously noted in response to an earlier comment:

*Line 38*: "Early applications of machine learning in hydrology date back to the 1990s, with artificial neural network (ANN) models used for rainfall-runoff modelling (e.g. **Hsu et al., 1995; Maier and Dandy, 1996; Zealand et al., 1999) and a range of other hydrometeorological analysis such as flood forecasting (Fleming et al., 2015), improving gridded snow-water equivalent data products (Snauffer et al., 2018), and predicting total April-August streamflow (Hsieh et al., 2003).**"

*Line 45*: "In addition to ANNs, which have received particular attention in hydrology (Maier et al., 2010; Maier and Dandy, 2000), numerous types of non-deep machine learning applications have also been developed for hydrometeorological analyses, and in particular, many have been developed for applications in Western Canada.  For example: Bayesian neural networks, support vector regression, and Gaussian processes have been used for streamflow prediction at a single basin (Rasouli et al., 2012); quantile regression neural networks have been used for precipitation downscaling in British Columbia (Cannon, 2011) and estimation of rainfall intensity-duration-frequency curves across Canada (Cannon, 2018); online sequential extreme learning machines have been used for streamflow prediction in two basins (Lima et al., 2016, 2017); and random forest models have been used to identify temperature controls on maximum snow-water equivalence in Western North America (Shrestha et al., 2021).  While ANNs and other non-deep machine learning architectures have a long history and continue to find useful applications in hydrology, DL has more recently become a promising area of investigation due to several key characteristics (Shen, 2018): DL models can automatically extract abstract features from large, raw datasets (Bengio et al., 2013), in contrast to labour-intensive manual feature extraction often required for non-deep models; and the existence of DL model architectures which are explicitly designed to learn complex spatial and/or temporal information, in particular convolutional neural networks (LeCun et al., 1990) and long short-term memory neural networks (Hochreiter and Schmidhuber, 1997)."

**\* Figure 1 would be much better, especially for an international readership that is unlikely to be strongly familiar with the study area, if it was a multi-panel figure that additionally illustrated topography, mean annual temperature, mean annual precipitation, and perhaps mean April 1 snow water equivalent.**

We have updated this figure and its caption to include panels of elevation, mean annual temperature, and mean annual precipitation:

[Figure]

**\* Lines 137-139: perhaps this passage merely is poorly written, but as it stands, the text implies a disturbing lack of understanding of the streamflow data being modeled.  Naturalized flow data are flow data that have been adjusted for upstream water management activities – diversions, withdrawals, reservoir operations, etc.  Data for stations upstream of dams are not necessarily naturalized, contrary to what is implied in this passage of the paper, and certainly in datasets like the HYDAT database used here, that step has not been undertaken and in many cases is unnecessary.  Similarly, dams are not the only disturbance that result in non-natural streamflow data that would in principle require naturalization prior to use in a hydrologic modeling study of the sort done here; another obvious example is land use change.**

The word "naturalized" was an unfortunate typo, and it should have been "natural" flow ("natural" in the sense that the HYDAT system classifies stream gauges as either "natural" or "regulated").  We clarify this in the text:

*Line 206*: "HYDAT classifies stream gauge stations as either "regulated" (downstream of regulating structures such as a dam) or "natural" (upstream of regulating features).  We use stations which are classified as natural and which are currently active."

**Why not use the Reference Hydrometric Basin Network (RHBN) stations or something similar? There is no mention here at all of the RHBN station network, which has been very widely used for decades for hydrological analysis and modeling studies in Canada.**

We now include the following on the RHBN:

*Line 224*: "The Reference Hydrometric Basin Network (RHBN) is a subset of the national stream gauge network which have long records and minimal human impacts that have been identified for use in climate change studies.  Of the 226 stations used in our study, 213 are within the RHBN.  The remaining 13 stations have long observational records and are not modified by regulating structures but may have more than minimal human impacts through other disturbances to the natural system such as land use changes.  We provide station names, station numbers, and if they are a part of the RHBN network (Table S1)."

**Also, I think quite a few hydrologists would raise their eyebrows at the specific data selection and processing procedures described in the first paragraph of section 3.1.**

Unfortunately, this comment does not provide us with any information as to how we could improve our data selection and processing procedures.  We will here comment on the steps we took.

One challenge is that we need temporally complete datasets, as the number of output neurons is constant through training.  We recognize that the threshold of 40% missing data may lead to challenges, if the data is missing during periods of dynamic streamflow (in other words, the model would be missing how to learn when streamflow should substantially change due to meteorological forcing); however, this is not the case.  A vast majority of data is missing between November and February, when temperatures are coldest and streamflow is more inhibited as compared to spring.  These data are typically missing at stations which record data seasonally, rather than continuously, and the 40% threshold allows us to "forgive" seasonal stations which do not record in winter months.  The following figure demonstrates how the most missing data occurs in the low-flow period.

[Figure]

**\* The second paragraph of section 3.1 is also muddled. All that's needed here is a concise statement that hydrometric network density is much higher in southern than northern Canada, and so, for the purposes of this study, the authors focused on the former.**

We edit this point:

*Line 220*: "We further restrict the study region to stations south of 56° N because stream gauge density is greater below this latitude."

**\* While the approach described on lines 159-170 is interesting and perhaps sufficient for the purposes of this study, overall it appears to be a naïve representation of spatiotemporal pattern formation in streamflow regimes in this study area. At an absolute minimum, some acknowledgement of prior work, and some caveats about the simple method and assumptions used here for regime classification, are needed. See in particular Halverson and Fleming (2015) and references cited therein. A particularly notable omission is that glacier-fed rivers are not identified as a distinct regime, whereas glacial cover is well-known to be a major control of streamflow dynamics in several areas within this region; see Moore et al. (2009), Fleming et al. (2016), Jost et al. (2012), and Bidlack et al. (2021).**

We recognize that we take a relatively simple approach in clustering stations into subdomains based only on seasonal hydrograph, latitude, and longitude. However, we use this clustering step not with the sole goal of finding stations which have the most similar physical and hydrological conditions (e.g. glacier cover, aspect, land use); rather, a key product of clustering is to find subsets of stations for which the model's learning can be more easily interpreted. It is desirable to identify clusters which are in large part determined by geographic location because one goal is to visualize where in space the model is learning to focus when predicting streamflow for each

cluster. When stations are nearby each other in space and the model is most sensitive in that small region, then we can better understand that the model is looking in the right place. When stations are spread over a larger area and clusters overlap more in space (e.g. if the importance of latitude and longitude are "watered down" by using other predictors in the clustering algorithm), the model may be sensitive over a large overlapping area for multiple clusters, and it becomes harder to interpret. Is the model focused on the watershed regions? Or is it just using the entire domain?

As noted to reviewer 1 who also had a similar comment on clustering: Consider two stations which are nearby one another, but have different characteristics such as drainage area, elevation, slope, aspect, and glaciation. In order to predict streamflow at each station, it should still be most important that the model focuses on areas near and within the two watersheds, respectively. For each station, the mapping through to streamflow from this 'most relevant information', then, may be different, but the sensitive areas should be similar. So, while clustering in the space of hydrologic variables other than geographic location may lead to small improvements in performance as measured by NSE by allowing the fine-tuned model to 'focus in' on more common details, it may make it more difficult to understand what the model is learning to do.

While we choose to not change our clustering method, we have added more context about the region's hydrology:

*Line 230*: "Streamflow throughout the study region varies strongly in space and time and reflects the varied topographic and climatic conditions in British Columbia and Alberta. Here we provide a brief, high-level overview of streamflow characteristics, and while it is not a complete summary of the full range of hydrologic conditions throughout the study region, we aim to highlight that streamflow through the region is heterogeneous in space and time. Streamflow at low-elevation coastal stations is primarily driven by rainfall, with monthly discharge maximized in November or December. In contrast, streamflow at stations that are at higher elevation, further north, or further inland transition to a snowmelt-dominated regime, with monthly discharge maximised in spring or early summer. Numerous glaciers exist in high elevation alpine areas throughout both the Coast Mountains along the west coast of British Columbia and the Rocky Mountains along the border between British Columbia and Alberta, and glacier runoff contributes to streamflow through late summer once the seasonal snowpack has melted (Eaton and Moore, 2010). East of the Rocky Mountains, the Prairie region in eastern Alberta is uniquely characterized by relatively flat topography with small surface depressions (LaBaugh et al., 1998). Water can pond and be stored in these depressions, leading to intermittent connectivity throughout many basins and drainage areas which may vary in time (e.g. Shook and Pomeroy, 2011)."

To comment on prior work that used clustering in the region:

*Line 251*: "Previous studies have used a range of techniques to cluster or summarize the diversity of spatiotemporal streamflow characteristics in the study region (e.g. Halverson and Fleming (2015) use complex networks to represent similarity between streamflow timeseries in the

Coastal Mountains, while Anderson and Radić (2020) use principal component analysis and Self-Organizing Maps to characterize summer streamflow through Alberta).  In this study we use a relatively simple clustering approach, only considering seasonal streamflow, station latitude, and station longitude."

To comment on why we use this simpler clustering approach:

*Line 274*: "Our clustering approach does not explicitly consider input features such as land use, glacier coverage, drainage area, or elevation, but rather implicitly considers the expressions of these features in the seasonal hydrograph.  The goal of this type of clustering is to define subsets of stream gauge stations that are nearby in space and share similar hydrographs.  We prioritize proximity in space over an explicit representation of other important features (e.g. drainage area, elevation, glacier coverage) because a key goal of the study is to interpret where in space the DL models have learned to focus when predicting streamflow.  As discussed in Sect. 4.3.1 and Sect. 4.5.1, having clusters of stream gauge stations which are nearby in space allows us to visualize if the trained models are learning to focus on the subregion of the input domain which overlaps with the watersheds where streamflow is being predicted."

**\* Section 3.1: I think reproducibility requires that the hydrometric station list used here be shown to readers.  A table in an appendix or supplementary materials would be fine.**

A table of station names, numbers, latitude, longitude, and if they are part of the RHBN network has been included in supplementary information (Table S1) and as 'station_table.csv' on Github.

**\* Section 3.5: provide information about the latency of the ERA5 reanalysis product – is it available in near-real time?  Some reanalysis products are, and some aren't.  It's a crucial question if one were interested in operationalizing a hydrologic prediction system like this for actual use in flood forecasting or another similar practical hydrologic prediction application.  If ERA5 products are not available in near-real time, then briefly but clearly state that limitation and its implications for wider use of the modeling framework introduced here.**

We include the following:

*Line 317*: "ERA5 data are available as a preliminary product 5 days behind real time, and as a final product 2 – 3 months behind real time (Hersbach et al., 2020).  This latency has implications for model applications, as it may not be possible to use ERA5 data for real-time forecasting with the model in this study."

**\* "data" = plural**

Having double checked all uses of "data", we found two which were incorrectly singular and these have been corrected.

**\* Somewhere in Section 3 or 4 there needs to be an explicit and clear description of what the training vs. testing vs. validation datasets are. There is a very brief mention of training vs validation but it is inadequate. The reader is not provided with information about how the training vs validation split is made, nor whether another subset is reserved for out-of-sample hyperparameter selection. These are standard practices in machine learning, and information about them is needed for transparency, reproducibility, and credibility of the study.**

We edit and include the following text:

*In original manuscript:* Since 365 days of previous temperature and precipitation are used to predict streamflow, and since the ERA5 data begin on December 1, 1979, the first day of streamflow predicted is January 1, 1980. For all models, we use 1980 – 2000 for training, 2001 – 2010 for validation, and 2011 – 2015 for testing.

*Added in updated manuscript, Line 416:* "In other words, the training period is defined by daily streamflow from January 1, 1980 to December 31, 2000, with forcing data ranging from January 1, 1979 to December 30, 2000. The validation period uses streamflow data from January 1, 2001 to December 31, 2010, with forcing data ranging from January 1, 2000 to December 30, 2010. The testing period uses streamflow data from January 1, 2010 to December 31, 2015, with forcing data ranging from January 1, 2009 to December 30, 2015. We choose to separate the training/validation/testing datasets into non-overlapping time periods of streamflow so that model performance can be evaluated on out-of-sample streamflow examples. We choose to use a full decade for validation because we want to encourage the model to perform well across a range of conditions and not for one particular year or climate state, since oscillations in the climate system such as the El-Nino Southern Oscillation, the Pacific Decadal Oscillation, and the Pacific-North American atmospheric teleconnection influence streamflow through modifications to temperature, precipitation, and snow accumulation through the study region (e.g. Fleming and Whitfield, 2010; Hsieh et al., 2003; Hsieh and Tang, 2001; Whitfield et al., 2010). We also choose to use multiple years for testing so as to not bias our conclusions towards the conditions of a single year. Furthermore, we partition the training, validation, and testing data by year rather than by percentage of observations (i.e. the testing subset is chosen as 5 years, not 10% of observations) so that we do not bias our results by including or excluding parts of the year when the model performs better or worse than average. Overall, the training-validation-testing data split is approximately 59% - 27% - 14% of the total streamflow dataset. The input data are normalized so that each variable (maximum temperature, minimum temperature, precipitation) has a mean of zero and unity variance over the training period. The target data from each of the 226 stations are normalized so that each station's streamflow has a mean of zero and unity variance over the training period."

*Line 459*: "It is possible that a better performing architecture or training scheme could be constructed by optimizing hyperparameters with an out-of-sample subset; however, we show our model setup and design is sufficient for achieving the goals of this study."

**\* A modern paper on machine learning applications to hydrologic prediction requires, in general, a performance comparison against some relevant benchmark model. Linear regression using precisely the same input dataset as the deep learning method introduced here is an obvious starting point and can provide a meaningful assessment of how much nonlinearity, interactions, etc contribute to the (presumably better) performance of the new technique. A conventional ANN and an LSTM would also be useful, if more ambitious, points of comparison.**

We agree with the reviewer and have now included a comparison of our CNN-LSTM model with an ensemble of linear models. The revisions read as following:

*Line 862*: "We compare our fine-tuned CNN-LSTM models against linear models to evaluate the extent to which the nonlinearities introduced by the CNN-LSTM approach improve streamflow predictions. We create an ensemble of 10 linear models for each cluster of stream gauge stations. Each linear model is a fully-connected ANN with an input layer, an output layer, and linear activation functions. We use the same training, validation, and testing data as in the CNN-LSTM approach. However, instead of structuring the input data as a video, each input observation is flattened and all values are input into the ANN. The target output is the next day of streamflow at all stations in the cluster. Therefore, for each model for cluster $i$, there are 420,480 input neurons (since each original observation is structured as a $365 \times 12 \times 32 \times 3$ video) and $N$ output neurons (where $N$ is the number of stations in cluster $i$). This approach was chosen in order to keep as much similarity as possible between the CNN-LSTM and linear model setup. The two approaches use the same input data, the same target data, and the same number of ensemble members, while the key difference is the nonlinearity and architecture of the CNN-LSTM model. We find that the CNN-LSTM model outperforms this simple linear benchmark, achieving a greater NSE at 222 out of 226 stations. The linear model has a minimum NSE of -13.33, a median NSE of 0.35, and a maximum NSE of 0.76, while the CNN-LSTM model has a minimum NSE of -0.7, a median NSE of 0.68, and a maximum NSE of 0.96."

**The only significant attempt the paper makes at this is Table 2, which scours the peer-reviewed journal literature for examples of hydrologic models that have been developed previously for a few of the locations considered in this study. That comparison is interesting and probably worth including in the paper, but it also has limited meaningfulness as different date ranges etc were used in the studies. Moreover, Table 2 relies on a small handful of academic studies and misses a lot of existing models within the study area operated by pragmatic water-management organizations like a large government-owned hydroelectric utility (BC Hydro), a provincial ministry (BC River Forecast Center), regional water management authorities (e.g., the MIKE-SHE model operated in the Okanagan Basin), and so forth. Moreover, given that even the simplest machine learning architecture outperforms process-based models in most cases, the somewhat mixed results in Table 2 are a little surprising.**

We recognize that this comparison may have limited meaningfulness as different date ranges were used in the study, which is why we emphasized in the text that it is not a direct comparison.

However, it is still valuable to at the very least comment on the performance of existing models in the peer-reviewed literature.

**In Section 5 there is also a very brief verbal comparison against the LSTM-based work of Kratzert et al. (2018) but that study used a completely different set of basins and data, so again, the comparison is extremely approximate.**

Yes, we are aware that Kratzert et al. (2018) use a different set of basins and data, which we note in the text. This does not mean that there is nothing to learn from prior regional-DL models. The purpose of this discussion in Section 5 is to say that the LSTM approach was improved by including temporally static catchment characteristics in the input, and noting that there would be ways to extend the CNN-LSTM approach to do this as well, outlining a potential avenue for future work.

**I get that the purpose of this study is more around demonstrating a new technology, and perhaps delving a little into the question of explainability, but I suspect most readers would like to see more meaningful inter-model performance comparisons here.**

We agree that inter-model comparison is important here, and we thank the referee for the idea to include a comparison to the linear benchmark (outlined above).

**\* Estimating predictive uncertainty is a key element of a hydrologic prediction system. Figure 6 and its caption suggests that predictive uncertainty is quantitatively estimated here but is vague about the method. It appears that an ensemble of 10 different models is formed, and twice the standard deviation of the predictions from those 10 models on a given day is used as the de facto prediction bound for that day. This is a reasonable first-cut approach, I think. However, the method needs to be described in the methods section, and some capabilities and limitations need to be mentioned; I suspect that because weather uncertainty is not factored in (as far as I can tell from the manuscript as submitted) the ensemble spread will be substantially under-dispersive.**

These uncertainty bounds reflect the range of streamflow predictions due to the randomness in the initialization of weights of the network and through training, and do not reflect uncertainty in meteorological drivers. This point has been made clearer by including the following:

*Line 475*: "We compute NSE using the mean predictions across the ensemble members, and we quantify an uncertainty in the streamflow prediction as being twice the standard deviation across ensemble members. This uncertainty is due to randomness from the initialized parameters and through training. It is a measure of how different streamflow predictions may be even when using the same architecture and data, and it is not a measure of uncertainty in meteorological forcing. When and where this uncertainty is small (large) indicates that the models in the ensemble predict similar (different) streamflow values for that day. We evaluate performance from an ensemble mean rather than a single model's prediction, and so this uncertainty gives an indication of the magnitude of scatter around the ensemble mean."

**\* The bar for explainability does not seem to be set very high here. The sensitivity analyses included in the paper are very useful, but they really amount to more of a plausibility test than an interpretability test. In particular, the paper demonstrates, though observing the CNN-LSTM responses to perturbations in the meteorological driving data, that its streamflow predictions (a) are most sensitive to weather in and near the basin as opposed to further away, and (b) are sensitive to temperature regimes, in particular, demonstrate hydrograph timing shifts corresponding to changes in snow accumulation and melt driven by temperature perturbations.**

Yes, the paper demonstrates the points (a) and (b), but it also goes further than that. We also demonstrate that the process of fine-tuning strongly influences the model's decision making by allowing it to (c) focus on smaller areas of the input (smaller A through fine-tuning), and (d) become more sensitive to perturbation near/within the watersheds being predicted and less so to areas further away (larger D through fine-tuning). "Interpretability" is more than revealing physical explanations of the input-output relationships, but is also building an understanding of the role of training steps. One question a person could ask themselves when training an ML model is: "When am I finished training?". One might compare NSE between a bulk and fine-tuned model and find them to be very close (as they are in this study, and others e.g. Kratzert et al. 2018). Only through (c) and (d) might we interpret how and if fine-tuning is improving model performance – perhaps not by making streamflow predictions which are more similar to observed values, but by better focusing in on the watershed areas.

**Those results suggest the CNN-LSTM model is capturing key geophysical processes more-or-less correctly, but it does not clearly reveal physical explanations of the input-output relationships – only that the behaviors are consistent with some basic physical expectations. I think the paper is publishable without diving further into explainability, but the authors ought to phrase their outcomes a little more precisely around the question of interpretability and may wish to consider some additional sleuthing to demonstrate that the CNN-LSTM reveals physical processes. There is some precedent for this in machine learing-based streamflow modeling, and looking closely at those precedents may be useful to the authors; examples include Fleming (2007), Kratzert et al. (2018), and Fleming et al. (2021). Looking even more broadly across the literature than this would likely lead to even more suggestions of how to examine the geophysical relationships the model is capturing.**

While it is not uncommon to center "interpretability" around the question of "which pixels are most important / relevant / sensitive for the model's decision making?" in the geophysical deep learning literature (e.g. Toms et al. (2020)), we have now added an additional analysis. We demonstrate that in glacier-fed rivers, August temperature perturbations are positively related to August mean streamflow (e.g. hotter temperatures lead to more flow), while this is not the case for non-glacier-fed rivers. Additionally, the strength of this relationship is positively (and non-linearly) related to the watershed glacier cover (greater percentage glaciation leads to flow being more sensitive to August temperature perturbations). This evidence supports the hypothesis that the model is learning physical processes (e.g. glacier-runoff contributions to streamflow, where melt is positively related to temperature) and is elaborated in the text:

[revised manuscript text omitted]
" – well, in practice the most widely applied hydrologic models tend to use only these two types of forcing because that's all that is usually available, so I guess this point might be worth mentioning here but it's not particularly "notable" to most streamflow modelers.**

We rephrase this line:

*Line 938*: "It is notable that the CNN-LSTM model achieves good streamflow simulation with only coarse resolution climate forcing data and localized streamflow data, with no knowledge of features such as basin characteristics, topography, or land use, and no explicit climate downscaling steps."

**\* Lines 635-638: is it possible that, through its empirical and complex meteorological input-hydrologic output mappings – effectively, a transfer function linking the meteorological data to the point streamflow observations – the CNN-LSTM effectively downscaled the reanalysis data, at least to some degree?  May be worth talking about here.**

Yes, this is possible, especially considering that CNNs have been used to map coarse resolution climate data to fine resolution climate data, indicating that sufficient information of high-resolution climate data is present within coarse resolution climate data (Vandal et al. (2017)).  We now make the following point in the text:

*Line 940*: "Our model uses forcing data at relatively coarse spatial resolution (0.75° x 0.75°, or ~75 km resolution)  as compared to studies identified in Table 2 (e.g. 0.0625° x 0.0.0625° in Shrestha et al. (2012); 10 km resolution in Eum et al. (2017)).  Studies that employ a climate downscaling step first map coarse resolution climate data to fine resolution climate data, and then map the downscaled fine resolution climate data to streamflow.  Here, the CNN-LSTM is effectively representing a single transfer function that maps coarse resolution climate data directly to streamflow, and it is possible that an effective downscaling of climate data is learned by the model.  This indirect downscaling is plausible since statistical methods are often used for climate downscaling, including CNNs (Vandal et al., 2017)."

**\* Lines 646-653: are the authors sure their method requires less data than an LSTM, as claimed here?  Doesn't the CNN-LSTM still ultimately need data for all N basins?  This passage needs further explanation/clarification.**

It is not that our method requires less data, but that our method leads to having fewer observations for training.  Consider a scenario where an LSTM is being used to predict streamflow at 10 stations individually (1 station-day of streamflow per observation), compared with a CNN-LSTM which is being used to predict streamflow at all 10 stations simultaneously (10 station-days of streamflow per observation).  Suppose that each station has 20 years of observations for training, meaning that there are (365 days / year) * (20 years) * (10 stations) station-days of streamflow in this dataset.  The LSTM approach converts 1 station-day to 1 observation for training.  The CNN-LSTM approach converts 10 station-days to 1 observation for training.  This reduces the number of observations for training by the CNN-LSTM approach by a factor of 10 (i.e. reducing the number of observations for training, but not reducing the total data requirements).  We do not wish to frame this as a "good" or "bad" thing, but rather it is something to consider when designing a model and how to train it.

We rephrase and include the following to improve clarity on this point:

*Line 947*: "In order for the model to learn the mapping between the meteorological forcing and streamflow, a sufficiently long data record is necessary for training. The CNN-LSTM architecture presented here predicts streamflow at multiple stations simultaneously. For a model which predicts at $N$ stations simultaneously, one target observation is $N$ station-days of streamflow. For a model which predicts at a single station (e.g. an LSTM with a single output neuron), one target observation is a single station-day of streamflow. For a given training dataset with $M$ station-days of streamflow observations, the CNN-LSTM with $N$ output neurons would have $M/N$ observations for training, while the model with a single output neuron would have $M$ observations for training. That the number of observations for training has been reduced is potentially detrimental to the model's performance. A potential solution to this problem could be to use transfer learning with a CNN-LSTM model pre-trained in a region with a sufficiently long streamflow record and then transferred to the new region of interest."

[revised manuscript text omitted]

---

## Author Response (AR2)

**Response to Referee 1**

**General comments:**

**The idea of this study is to develop a regional streamflow model using a convolutional long short-term memory artificial neural network, which is the merger of two distinct deep learning (DL) techniques. This and several other innovations presented in the paper are quite impressive, and the overall performance of the model seems good. The revised paper is also in much better shape than the original submission, with considerably more detail given, much better figures, and some significant new analysis to shore up some weak points in the original study, such as including a linear benchmark model for comparison and additional sensitivity analyses demonstrating physically reasonable responses to perturbations in temperature fields, which (to some degree) ties into broader goals like explainable machine learning.**

We thank the referee for their detailed review and are pleased they find the revised manuscript to be a substantial improvement.

**Unfortunately, the quality of the writing and explanations remains somewhat inadequate. The general impression one receives when reading this manuscript (which may or may not be true, but it is the impression one gets from the writing) is that the authors have some background with areas of geophysical science adjacent to watershed hydrology, but not watershed hydrology itself, and certainly not any aspect of operational hydrology or streamflow modeling. Similarly, treatments of machine learning in the manuscript seem to suggest familiarity with a very narrow range of sophisticated techniques but not a great awareness of the overall field of machine learning and, in particular, prior work on its application to streamflow modeling. Sadly, this may only serve to reinforce negative impressions among the water resource community as a whole about the general usefulness and credibility of machine learning – impressions that have been crippling in some important ways to the advancement of the field of hydrology.**

We have addressed the detailed comments below to improve the quality of writing and explanations. Additionally, we have had Dr. William Hsieh review the manuscript with the goal of identifying shortcomings related to the quality of writing and explanations, and have implemented his suggestions for improvement.

**Additionally, the primary technical innovation presented here – taking the long short-term memory (LSTM) neural network from time series analysis, which has recently seen several high-profile research applications to streamflow modeling, and adding in a convolutional neural network (CNN) from image analysis – involves an incremental advance over the existing LSTM approach, yet the existing LSTM approach is never implemented here. As a result, the paper cannot provide information on how much of an advantage, if any, the addition of a CNN architecture provides. Perhaps this is not strictly needed for publication, but it is an obvious limitation of the study that may compromise the adoption of this novel streamflow modeling technique by others.**

**Overall, this manuscript contains many clever and potentially powerful ideas but seems to be poorly executed, and it feels like an appropriate recommendation is for publication pending major revisions.**

We are glad the referee agrees that this study is built around many clever and potentially powerful ideas.  We address all comments below, with new text in the manuscript identified in **bold blue text**.

**Detailed comments:**

**Line 10, delete "the region of"**

Deleted.

**Lines 31-32, not clear what the authors mean by a spatially distributed DL model. Use of the concepts of lumped, semi-distributed, and fully distributed hydrologic models has been pretty much exclusive to process-based models and it's not clear how it can be extended to ML-based hydrologic models. Reading the rest of the paper, I can make an educated guess as to what the authors trying to get at here, but the reader should not have to do this and it's still not clear that the terms really apply as such to machine learning models in hydrology. Are they referring to multiple forecast points (corresponding to gages)? If so, that's captured in the concept of a regional hydrology model, which is not necessarily the same thing as a fully distributed hydrology model (as many fully distributed models make predictions at only a single location for example). Are they referring to fully spatially distributed (e.g. gridded) inputs? If so, that was successfully tackled decades ago in ML-based streamflow modeling by Hsieh et al. (2003). Moreover, the regional DL models with many input data and output prediction locations introduced by Kratzert et al. (2018, 2019a, 2019b) feel like they may be just as spatially distributed as the DL model introduced in this submission. The authors need to be much clearer and more specific on what they mean here and consider whether the confusion created by this mixing-and-matching of terminology is really beneficial to their ultimate purpose and the clarity and credibility of this submission. I suspect this is another case (the problem was widespread in the original submission) of slightly misusing standard hydrologic nomenclature. That said, see comment below re: line 137, where the manuscript handles all this much better.**

We have edited this section as follows:

Line 30: "These recent DL-based studies have emphasized the development of lumped hydrological models **with inputs that are aggregated to the basin-level**.  However, **fewer DL-based studies have explored the use of spatially discretized forcing and geophysical data** (Gauch and Lin, 2020).  In contrast, traditional process-based approaches have made

substantial progress towards distributed hydrological models **which are driven by spatially discretized inputs** (Freeze and Harlan, 1969; Marsh et al., 2020; Pomeroy et al., 2007)."

Furthermore, throughout the manuscript we reserve "distributed" to describe distributed process-based models, and refer to DL-based models forced by spatially discretized inputs as a "**DL analogue of a distributed hydrological model**" or a "**DL model that is driven by spatially discretized forcing data**".

**Line 41: replace "total April-August streamflow" with "seasonal water supply", which was the point of the exercise and is a major, mainstream task in water resource forecasting and management.**

Changed to "**seasonal water supply**".

**Lines 45-48: this basic description of ML in hydrology is clunky and imprecise. It could be easily read to imply the authors think that a Bayesian neural network is not an ANN, or that they think ANNs aren't non-deep (in truth, traditional feedforward-error backpropagation ANNs of the sort being referred to here may by deep or non-deep depending on how many hidden layers they have), etc. Mistakes like this up-front in the introductory section may immediately draw the paper's basic credibility into immediate question, no matter how innovative and correct the actual technical work subsequently presented in the paper may be.**

We have rephrased this passage to more precisely introduce machine learning applications in Western Canada:

Previously: "In addition to ANNs, which have received particular attention in hydrology (Maier et al., 2010; Maier and Dandy, 2000), numerous types of non-deep machine learning applications have also been developed for hydrometeorological analyses, and in particular, many have been developed for applications in Western Canada.

Updated, Line 46: "**In particular, numerous types of machine learning applications have been developed for hydrometeorological analyses and applications in Western Canada.**"

**Line 53: here the authors are implying that ANNs are non-deep, whereas that may or may not be the case (see preceding comment). This error is just sloppy writing and is totally avoidable. Again, the overall impression one gets from these passages is that the authors are not very familiar or comfortable with the field of ML in general, which is not a helpful image to present to the reader.**

We edit this sentence to remove this implication:

Previously: "While ANNs and other non-deep machine learning architectures have a long history and continue to find useful applications in hydrology…"

Updated, Line 53: "While **such** machine learning architectures have a long history and continue to find useful applications in hydrology…"

**Lines 56-57, comment about advantages of deep learning relative to "labour-intensive manual feature extraction often required for non-deep models" - essentially true but also substantially exaggerated, which again undermines the credibility of the manuscript. Automated predictor selection and feature creation techniques have been used in statistical modeling for decades and have appeared in non-deep machine learning too. A recent example is Fleming et al. (2021b).**

The clause "in contrast to labour-intensive manual feature extraction often required for non-deep models" is removed.

**Lines 70-75: good description, but might want to consider mentioning here that Kratzert et al. (2019a) additionally used spatially heterogeneous physical basin characteristics as predictors in regional LSTM models. I believe this may be mentioned later in the manuscript but ought to be briefly pointed out here.**

We have edited the passage as follows:

Line 70: "LSTM models trained on many basins have been shown to outperform standard hydrological models for prediction at ungauged basins, **and the inclusion of physical basin characteristics as predictors further improved the LSTM model performance**, demonstrating the potential for LSTM models to be used as regional hydrological models (Kratzert et al., 2019a).  However, while addressing the need to learn complex sequential information, the LSTM approach does not explicitly **learn from spatially discretized information**, and as such has been primarily used for lumped hydrological modelling."

**Line 80: beach state classification in coastal geomorphology is another example; see Ellenson et al. (2020).**

Line 81: "… **and beach state classification (Ellenson et al., 2020).**"

**The introductory section's discussion of explainability in machine learning is inadequate and under-referenced, especially from the viewpoint of socially relevant hydrologic model applications, i.e., things such as actual flood and water supply forecasting at government agencies and the like. At a minimum, on line 100, after the sentence ending with "making", add the following: "Practical methods are beginning to appear that allow users to easily identify and geophysically interpret, in detail, spatiotemporal patterns or input-output relationships identified by, respectively, new unsupervised learning (e.g., Fleming et al., 2021a) and supervised learning (e.g., Fleming et al., 2021b) algorithms designed for applied operational hydrological modelling environments where interpretability is key. However, there is still much work to be done on developing new and better ways to further the goal of**

**explainable machine learning for hydrology, in both deep and non-deep contexts and both operations and research settings.”** Both of these cited manuscripts describe new but non-deep ML methods that are far more focused on, and successful at, providing extensive and complete geophysical interpretations than the method introduced in this submission or other deep learning work in hydrology so far.

The suggested text has been added.

**Line 107: add “or practical” after “research” and before “questions”**

The suggested text has been added.

**Line 137: yes, nicely done! Compared to lines 31-32 (see comment above), this is a much better description of what's being meant by a “distributed” model in the context of DL in this paper, though it's still not clear that using the term to described the CNN-LSTM application is particularly helpful.**

We thank the referee for the positive feedback. We are more careful through the text to refer to the CNN-LSTM model not as being distributed, but rather as being a DL model that uses spatially discretized forcing data as input.

**Point 2 on line 140, the authors should modify the text slightly to be explicit that they're referring to the spatially distributed input data**

This point has been edited as:

Line 145: “2) investigate if the model has learned to focus on the areas **of the spatially distributed input data** that are within or near the watersheds where streamflow is being predicted”

**Line 153: this region is never referred to by anyone strong familiar with it as “the south-central domain of western Canada” and this description doesn't even make geographic sense (what's “central” about it?). Ironically, statements like this are likely to undermine the credibility of the work specifically with hydrologists working in the paper's study area. Maybe try just calling it what it is: “southwestern Canada” and/or “the southern portions of the western Canadian provinces of British Columbia and Alberta”.**

Line 161: Rephrased as “**southwestern Canada**”.

**Line 187, after “acceptable” insert “for the purposes of this study”**

Line 193: We have added “**for the purposes of this study**”.

**Lines 205 and 207, "maximized" is not quite the right word here; the term suggests optimization**

Line 213: We have changed "maximized" to "**highest**", consistent with Eaton and Moore (2010).

**Line 210, is "uniquely" the right word here? As described here, this is not unique to this region, as similar processes happen in other parts of the Canadian Prairies and presumably elsewhere as well. It's the only part of the study region with that characteristic, though, which maybe is what the authors are trying to say?**

The word "uniquely" has been removed.

**Line 224: the standard statistical nomenclature is "unit variance" not "unity variance"**

This change has been made throughout the text.

**Line 331: insert "(from which CNN technologies primarily originated)" after "image processing"**

This text has been added.

**Line 331: I'm not confident hydroclimatologists would view this type of work as "hydro-climatic modelling"**

We have edited this sentence:

Line 346: "To ensure consistency between terminology in both image processing **(from which CNN technologies primarily originated)** and **this study**…"

**Line 332: after "the three weather predictors", should "at all grid cells" be added to further clarify? I think that's what's meant here, but it needs to be entirely clear given the analogies being drawn here between video processing and spatiotemporal climate fields**

Line 349: We have added "**at all grid cells**" for clarity.

**Lines 370-375: this is mostly sound logic, except perhaps for the PDO, which is generally thought to typically remain in one state for a couple decades or so between regime shifts**

The reference to the PDO has been removed.

**Section 4.3.1: this is the first place in the manuscript that ensemble modeling is introduced, and strangely, no effort is made to explain here or anywhere else in the paper how that ensemble is formed. Between this treatment (or lack thereof) in the manuscript on the one hand and their rebuttal letter on the other hand, it's not clear the authors are aware that**

**there is more than one way to create an ensemble of ML models, and they do need to provide a brief explanation of how they did it (one sentence will suffice).**

We clarify how we generate this ensemble:

Line 416: "1. Bulk training: **a CNN-LSTM model is initialized with random weights and is then** trained on all 226 stream gauge stations in the region."

Removed: "In total, we train an ensemble of 10 bulk models and further fine-tune each one, yielding an ensemble of 10 bulk models and an ensemble of 10 fine-tuned models per cluster."

Line 427: Replaced with: "**We initialize 10 bulk models with 10 different sets of random weights.  Each bulk model is trained and then fine-tuned on each cluster of stream gauge stations, creating 10 fine-tuned CNN-LSTM models per each of the six clusters of stream gauge stations.  We use this ensemble of 10 bulk models and 10 fine-tuned models (per cluster) for our analysis**."

**Line 460: "Gaussian distribution" not just "gaussian" which is informal slang**

This edit has been made where needed.

**Lines 505-507: temperature degree-day models go a lot further back than this; provide more several more references, including references specific to the use of this type of snow sub-models within standard watershed hydrology models**

We have updated this section as follows:

Line 529: "While the assumption is a simplification of processes dictated by the surface energy balance, the use of positive temperatures as successful indicators for the warming and melting of snow is a common assumption of positive-degree-day models in simulating snow and glacier melt **and was first used by Finsterwalder and Schunk (1887).  Such positive-degree-day models have since been widely applied for modelling snow and glacier melt across multiple spatial scales (e.g. Hoinkesand and Steinacker, 1975; Braithwaite, 1995; Hock, 2003; Radic et al., 2014), and have been used in watershed hydrology models such as the UBC watershed model (Quick and Pipes, 1977) and the HBV-model (Bergström, 1976).**"

**Figure 6: it is interesting that the overall test-phase NSE reported here for the Englishman River is substantially lower than that for the ensemble non-deep ANN for this same river described by Fleming et al. (2015). This may suggest that the advantage of simultaneous regional modeling across a large domain by the CNN-LSTM network introduced in this paper is accompanied by the disadvantage of weaker performance on a single given river of particular interest, which is often what water resource professionals are primarily concerned with – a particular river with socially destructive flooding events, for example, or a tributary to a reservoir that requires inflow predictions. This result might be traceable, at least in part,**

**to the benefits of making specific choices, on the basis of general expertise in physical hydrology and familiarity with the particular watershed in question, that can be easily made when modeling a single watershed but are more cumbersome in a regional model. For example, the Englishman River-specific ensemble non-deep ANNs of Fleming et al. (2015) included snow pillow and antecedent streamflow inputs as inputs. That does not imply that the submitted study has done anything wrong – on the contrary, the result likely reflects an expected trade-off between scale and detail in a modeling system. The paper should explicitly acknowledge this point using the Englishman River, and the comparison to previous AI work in that basin, as what appears to be a clear example. For these reasons, the Fleming et al. (2015) study should also obviously be added to Table 2, giving an additional point of comparison for the Englishman River, which is already included in that table but only for a very old study presenting a lower-performing process-based hydrologic model.**

This is a good point raised by the referee, and we now explicitly discuss this comparison between the results for the Englishman River as in Fleming et al. (2015) and in our study:

Line 806: "**Prior studies have modelled daily streamflow at the Englishman River near Parksville (08HB002), one of the locations in our study; for example, Fleming et al. (2015) use an ensemble of ANNs to forecast streamflow and achieve NSE values in the range of 0.7 – 0.8, while Lima et al. (2016) use nonlinear extreme learning machines and achieve NSE > 0.8. These examples outperform the NSE value of 0.59 achieved by our CNN-LSTM. Their success could be in part due to the inclusion of more locally-specific input data (e.g. Fleming et al. (2015) include snow pillow and antecedent streamflow data, while Lima et al. (2016) include predictors such as sea level pressure, wind speed, and humidity, among others), a decision which can be more easily implemented for modelling at a single stream gauge station as compared to a regional scale model. These examples highlight what may be a trade-off between scale and detail in the modelling approach, where the advantage of simultaneous streamflow modelling at multiple stream gauge stations across a region as done by the CNN-LSTM network may be met by the disadvantage of weaker performance on one particular river of interest.**"

It is not obvious to us that Fleming et al. (2015), which uses an ANN, should be included in Table 2, which compares only with process-based models. As such we reserve this discussion for the main text.

**Drop the term "heat map" from the manuscript. It's a standard term in graphics production, but in the context of a manuscript dealing with various geophysical quantities including temperature, it's unnecessarily ambiguous.**

We have removed the term "heat map" and refer to this quantity throughout the manuscript as a "sensitivity map" or "$S(x,y)$" as defined in Equation 12.

**Lines 624-626: can the authors offer a specific hypothesis or two why the eastern and northeastern clusters show such a strong sensitivity to coastal conditions? Could it perhaps**

**reflect some meteorological setup, e.g., jet stream position, storm tracks, etc.? It's a very prominent feature of the results.**

We add:

Line 654: "Another possible explanation is that there could be temporal patterns of sensitivity. For example, the eastern and north-eastern regions may be sensitive to coastal conditions when storms travel from west to east. Alternatively, the sensitivity maps may be most sensitive to coastal conditions during winter, when the model could be tracking above-freezing temperatures. Future work should investigate these links further to evaluate their meaning and implications for CNN-LSTM performance."

We agree that this is a prominent feature of the results and it is the subject of our future work.

**A major point of the article is that the resulting CNN-LSTM neural network provides results that are physically explainable in the sense that perturbations to driving fields yield the streamflow responses one would expect on the basis of physical hydrologic knowledge. That's great, but it should be made very clear that this is not necessarily a unique attribute of deep learning – that has not been at all demonstrated here, and much the same might be expected from non-deep machine learning or even statistical models in the same application, provided they are built correctly.**

We agree with the referee on this point and we do not claim that the physically explainable perturbation responses are necessarily unique to deep learning models. We add:

Line 152: "We explore several ways that perturbations to the input temperature and precipitation fields result in streamflow responses that are expected on the basis of physical hydrologic knowledge. While this is not necessarily a unique property of DL and may be found when using non-deep machine learning or other empirical models applied to the same task, our findings are encouraging given the recent use of DL for streamflow prediction tasks."

**Line 777: this is a grossly inadequate explanation**

We have provided the following clarification:

Line 820: "The CNN-LSTM is designed to receive an input structured as a weather video, while in comparison, ANNs are designed to receive an input structured as a single vector. The input neurons in the ANN correspond to each variable at each grid point and each day in a single weather video, meaning that there are 420,480 input neurons. For example, the input to predict flow on September 30, 2011 is daily maximum temperature, minimum temperature, and precipitation from September 30, 2010 through September 29, 2011, at each grid point in the study region. For the CNN-LSTM, these data are structured as a weather video with shape $365 \times 12 \times 32 \times 3$ (e.g. day $\times$ latitude $\times$ longitude $\times$ variable), but for the ANN, these data are structured as a vector with length 420,480."

**Line 800: "seasonal-scale input time series" – really? Decades-long time series with a daily sampling interval were used in this study, were they not? So, what are the authors trying to express here?**

We have clarified this section to reflect that a single year of temperature and precipitation is used to predict the next day of streamflow, from which there could be insufficient information to know the state of depressional storage (wetting or drying) and thus the correct streamflow response:

Line 845: "Storage in ponds can vary on both seasonal and decadal timescales (Hayashi et al., 2016; Shaw et al., 2012), but only a single year of daily temperature and precipitation is used to predict the next day of streamflow. It could be that the CNN-LSTM model cannot accurately predict the streamflow response in eastern basins because one year of temperature and precipitation is insufficient information to know the state of depressional storage (e.g. seasonal and decadal fluctuations in wetting or drying)."

**Lines 813-816: this supposition is inconsistent with the basics of physical hydrology. Interactions between streamflow and geology (aquifers, soil moisture storage, etc) directly and nonlinearly affect the temporal dynamics of streamflow responses to forcing meteorology. The passage is also inconsistent with the work of Kratzert et al. (2019a), who demonstrated that including static catchment characteristics as predictors in a LSTM streamflow model substantially improves performance, and Kratzert et al. (2019b), who demonstrated that a new variant they developed of the LSTM can extract features corresponding to static basin characteristics that capture geological and other watershed properties.**

This passage has been removed.

**Line 880, "a single year of temperature and precipitation alone" – elsewhere the manuscript states that the data were over 1980-2015 (or 1979-2015, the paper is inconsistent on that point, e.g., between the abstract and conclusions). Even after subsetting the data into training and testing datasets, that still leaves several years, so where does the "single year" comment come from?**

The "single year" referred to the input time series for the model (e.g. one year of daily temperature and precipitation is input to the model to predict one day of streamflow). In this case, we have removed "a single year" to comment only on processes which cannot be learned from temperature and precipitation:

Line 915: "The poor performance in the Prairie region may be due to the importance of processes which are underrepresented or not represented in the training data, such as processes occurring over longer than annual timescales, or at smaller spatial scales, or which are not able to be described from temperature and precipitation alone."

Temperature and precipitation data begin in 1979 while streamflow data begin in 1980 (since the prior year of temperature and precipitation is used to predict the next day of streamflow). We have ensured consistency on this point in the text (e.g. we refer to streamflow predictions between 1980 – 2015, and only refer to temperature and precipitation data in 1979). We also note specifically:

Line 383: "Since 365 days of previous temperature and precipitation are used to predict streamflow, and since the ERA5 data begin on December 1, 1979, the first day of streamflow predicted is January 1, 1980."

We have edited the conclusion so that it only refers to the date range for which streamflow predictions are made (1980 – 2015), so that it is consistent with the abstract:

Prior: "We focused on using a relatively simple deep learning model, with the input data represented by temperature and precipitation reanalysis for the period 1979 – 2015 given on relatively coarse spatial resolution (0.75° x 0.75°)."

Now, on line 904: "We focused on using a relatively simple deep learning model, with the input data represented by temperature and precipitation reanalysis given on relatively coarse spatial resolution (0.75° x 0.75°). **The deep learning model is used to predict daily streamflow between 1980 – 2015 at 226 stream gauge stations.**"

**I believe the terms "validation" and "testing" are used inconsistently across the manuscript.**

We have reviewed the use of "validation" and "testing" and have not found their use to be inconsistent. One potential source of confusion may be that the process-based models (e.g. in Table 2) report their performance on their "validation period", while we report our CNN-LSTM performance on our "testing period". We have further clarified this difference:

Line 797: "**We note a difference in terminology between the process-based model results and our CNN-LSTM results. Both evaluate models on 'unseen data' that were not used to determine the model parameters; however, the process-based models refer to this dataset as 'validation data' while we refer to this dataset as 'testing data'.**"

**Moreover, for the benefit of readers less familiar with machine learning, the manuscript should clearly explain the difference between training, validation, and testing datasets in the context of the CNN-LSTM network used here.**

We also clarify the difference between the training, validation, and testing datasets in the context of the CNN-LSTM model:

Line 378: "**We divide our data into three subsets referred to as training, validation, and testing datasets, as is common practice in DL model development (e.g. Goodfellow et al.,**

2016).  **The training data are used to iteratively update the model parameters such that the error between the model's predictions and known observations is reduced across the training set; the validation data are used to determine when to stop updating the model parameters to prevent the model from overfitting to the training data; and the testing data are used to evaluate the final model's performance.**"

**References:**

Ellenson A, Simmons JA, Wilson GW, Hesser TJ, Splinter KD. 2020. Beach state recognition using Argus imagery and convolutional neural networks. Remote Sensing, 12, 3953.

Fleming SW, Bourdin DR, Campbell D, Stull RB, Gardner T. 2015. Development and operational testing of a super-ensemble artificial intelligence flood-forecast model for a Pacific Northwest river. Journal of the American Water Resources Association, 51, 502-512.

Fleming SW, Garen DC, Goodbody AG, McCarthy CS, Landers LC. 2021b. Assessing the new Natural Resources Conservation Service water supply forecast model for the American West: a challenging test of explainable, automated, ensemble artificial intelligence. Journal of Hydrology, 602, 126782.

Fleming SW, Vesselinov VV, Goodbody AG. 2021a. Augmenting geophysical interpretation of data-driven operational water supply forecast modeling for a western US river using a hybrid machine learning approach. Journal of Hydrology, 597, 126327.

Hsieh WW, Yuval, Li J, Shabbar A, Smith S. 2003. Seasonal prediction with error estimation of Columbia River streamflow in British Columbia. Journal of Water Resource Planning and Management, 129, 146-149.

Kratzert F, Klotz D, Brenner C, Schulz K, Herrnegger M. 2018. Rainfall-runoff modelling using long short-term memory (LSTM) networks. Hydrology and Earth System Sciences, 22, 6005-6022.

Kratzert F, Klotz D, Herrnegger M, Sampson AK, Hochreiter S, Nearing GS. 2019a. Toward improved predictions in ungauged basins: exploiting the power of machine learning. Water Resources Research, 55, 11344-11354.

Kratzert F, Klotz D, Shalev G, Klambauer G, Hochreiter S, Nearing G. 2019b. Towards learning universal, regional, and local hydrological behaviors via machine learning applied to large-sample datasets. Hydrology and Earth System Sciences, 23, 5089-5110.

References:

Bergström, S.: Development and Application of a Conceptual Runoff Model for Scandinavian Catchments, 1976.

Braithwaite, R. J.: Positive degree-day factors for ablation on the Greenland ice sheet studied by energy-balance modelling, 41, 153–160, https://doi.org/10.1017/S0022143000017846, 1995.

Eaton, B. and Moore, R. D.: Regional Hydrology, in: Compendium of forest hydrology and geomorphology in British Columbia, edited by: Pike, R. G., Redding, T. E., Moore, R. D., Winkler, R. D., and Bladon, K. D., B.C. Ministry of Forests and Range, Victoria, British Columbia, 85–110, 2010.

Ellenson, A. N., Simmons, J. A., Wilson, G. W., Hesser, T. J., and Splinter, K. D.: Beach State Recognition Using Argus Imagery and Convolutional Neural Networks, 12, 3953, https://doi.org/10.3390/rs12233953, 2020.

Finsterwalder, S. and Schunk, H.: Der suldenferner, 18, 72–89, 1887.

Fleming, S. W., Bourdin, D. R., Campbell, D., Stull, R. B., and Gardner, T.: Development and Operational Testing of a Super-Ensemble Artificial Intelligence Flood-Forecast Model for a Pacific Northwest River, 51, 502–512, https://doi.org/https://doi.org/10.1111/jawr.12259, 2015.

Freeze, R. A. and Harlan, R. L.: Blueprint for a physically-based, digitally-simulated hydrologic response model, https://doi.org/10.1016/0022-1694(69)90020-1, 1969.

Gauch, M. and Lin, J.: A Data Scientist's Guide to Streamflow Prediction, 2020

Goodfellow, I., Bengio, Y., and Courville, A.: Deep Learning, MIT Press, 2016.

Hayashi, M., van der Kamp, G., and Rosenberry, D. O.: Hydrology of Prairie Wetlands: Understanding the Integrated Surface-Water and Groundwater Processes, 36, 237–254, https://doi.org/10.1007/s13157-016-0797-9, 2016.

Hock, R.: Temperature index melt modelling in mountain areas, 282, 104–115, https://doi.org/https://doi.org/10.1016/S0022-1694(03)00257-9, 2003.

Hoinkesand, H. and Steinacker, R.: Hydrometeorological implications of the mass balance of Hintereisferner, 1952-53 to 1968-69, 1975.

Kratzert, F., Klotz, D., Herrnegger, M., Sampson, A. K., Hochreiter, S., and Nearing, G. S.: Toward Improved Predictions in Ungauged Basins: Exploiting the Power of Machine Learning, 55, 11344–11354, https://doi.org/10.1029/2019WR026065, 2019a.

Maier, H. R. and Dandy, G. C.: Neural networks for the prediction and forecasting of water resources variables: a review of modelling issues and applications, 15, 101–124, https://doi.org/https://doi.org/10.1016/S1364-8152(99)00007-9, 2000.

Maier, H. R., Jain, A., Dandy, G. C., and Sudheer, K. P.: Methods used for the development of neural networks for the prediction of water resource variables in river systems: Current status and future directions, 25, 891–909, https://doi.org/https://doi.org/10.1016/j.envsoft.2010.02.003, 2010.

Marsh, C. B., Pomeroy, J. W., and Wheater, H. S.: The Canadian Hydrological Model (CHM) v1.0: a multi-scale, multi-extent, variable-complexity hydrological model -- design and overview, 13, 225–247, https://doi.org/10.5194/gmd-13-225-2020, 2020.

Pomeroy, J. W., Gray, D. M., Brown, T., Hedstrom, N. R., Quinton, W. L., Granger, R. J., and Carey, S. K.: The cold regions hydrological model: A platform for basing process representation and model structure on physical evidence, in: Hydrological Processes, 2650–2667, https://doi.org/10.1002/hyp.6787, 2007.

Quick, M. C. and Pipes, A.: U.B.C. WATERSHED MODEL / Le modèle du bassin versant U.C.B, 22, 153–161, 1977.

Radic, V., Bliss, A., Beedlow, A. C., Hock, R., Miles, E., and Cogley, J. G.: Regional and global projections of twenty-first century glacier mass changes in response to climate scenarios from global climate models, 42, 37–58, https://doi.org/http://dx.doi.org/10.1007/s00382-013-1719-7, 2014.

Shaw, D. A., Vanderkamp, G., Conly, F. M., Pietroniro, A., and Martz, L.: The Fill–Spill Hydrology of Prairie Wetland Complexes during Drought and Deluge, 26, 3147–3156, https://doi.org/https://doi.org/10.1002/hyp.8390, 2012.

**Response to Referee 2**

**I thank the authors for carefully considering my comments and providing detailed and appropriate responses. The revised manuscript is a major improvement in terms of presentation, and highlighting key findings and novelty. I have a few minor points that the authors need to clarify.**

We are glad that the referee considers the revised manuscript to be a major improvement. We thank the referee for their points, which we address below (new text in **blue and bold**).

**I am not convinced with the argument that ERA5 data has global coverage and appropriate for this study because the DL methods could be transferred to other regions. Just because ERA5 driven DL model performed well in this region does not mean that it is appropriate for other regions. The methods seem to be transferable irrespective of the input data, and 'best' available input data should be used in each study region. This needs to be clarified.**

We have updated the following text:

Line 271: "ERA5 reanalysis is globally available from 1979 through the present (once complete it will be available from 1950 onwards) and has been shown to compare well against other reanalysis products (Hersbach et al., 2020). **ERA5 reanalysis was preceded by the ERA-Interim reanalysis, which has been evaluated for use across British Columbia. It was found that daily minimum and daily maximum temperatures are well represented across British Columbia (Odon et al., 2018). Additionally, daily precipitation was found to be well represented, with the caveat that extreme precipitation is less successfully represented (Odon et al., 2019). ERA5 reanalysis better represents precipitation as compared to ERA-Interim reanalysis at the global scale (Hersbach et al., 2020).** Importantly, precipitation from ERA5 has been found to typically outperform ERA-Interim reanalysis in the northern Great Plains region, which experiences a similar climate to the Prairie region in our study area (Xu et al., 2019). **For these reasons we consider the ERA5 reanalysis to be suitable for our study.**"

**I am doubtful that the "day when the 30-day running mean of modelled streamflow rises to be halfway between the winter minimum flow (Qmin) and spring maximum flow (Qmax)" is a robust estimation of freshet timing. This is similar to "centre of volume" not being a robust measure of snowmelt timing (Whitfield 2013). Given that this is not a main focus of this paper, I suggested qualifying it as an 'indicator of freshet timing', with a caveat that this may not reflect actual freshet timing.**

We qualify this as an "indicator of freshet timing" as suggested:

Line 552: "For each cluster and temperature perturbation, we define **an indicator of freshet timing** ($t_{freshet}$) as the day when the 30-day running mean of modelled streamflow rises to be halfway between the winter minimum flow ($Q_{min}$) and spring maximum flow ($Q_{max}$)"

Throughout the text we now refer to $t_{freshet}$ as an "indicator of freshet timing" or "freshet timing indicator" rather than "freshet timing".

**Whitfield, P. H., 2013: Is 'Centre of Volume' a robust indicator of changes in snowmelt timing? Hydrol. Process., 27, 2691–2698, https://doi.org/10.1002/hyp.9817.**

---

## Author Response (AR3)

The submitted manuscript is the finalized version of the prior submission (no tracked changes).